# Gaussian Process regression model for dynamically calibrating and surveilling a wireless low-cost particulate matter sensor network in Delhi

Tongshu Zheng[1], Michael H. Bergin[1], Ronak Sutaria[2], Sachchida N. Tripathi[3], Robert Caldow[4], David E. Carlson[1,5]

[1]Department of Civil and Environmental Engineering, Duke University, Durham, NC 27708, USA
[2]Respirer Living Sciences Pvt. Ltd, 7, Maheshwar Nivas, Tilak Road, Santacruz (W), Mumbai 400054, India
[3]Department of Civil Engineering, Indian Institute of Technology Kanpur, Kanpur, Uttar Pradesh 208016, India
[4]TSI Inc., 500 Cardigan Road, Shoreview, MN 55126, USA
[5]Department of Biostatistics and Bioinformatics, Duke University, Durham, NC 27708, USA

*Correspondence to*: Tongshu Zheng (tongshu.zheng@duke.edu)

**Abstract.** Wireless low-cost particulate matter sensor networks (WLPMSNs) are transforming air quality monitoring by providing PM information at finer spatial and temporal resolutions; however, large-scale WLPMSN calibration and maintenance remain a challenge because the manual labor involved in initial calibration by collocation and routine recalibration is intensive, the transferability of the calibration models determined from initial collocation to new deployment sites is questionable, as calibration factors typically vary with urban heterogeneity of operating conditions and aerosol optical properties, and the stability of low-cost sensors can drift or degrade over time. This study presents a simultaneous Gaussian Process regression (GPR) and simple linear regression pipeline to calibrate and monitor dense WLPMSNs on the fly by leveraging all available reference monitors across an area without resorting to pre-deployment collocation calibration. We evaluated our method for Delhi, where the $PM_{2.5}$ measurements of all 22 regulatory reference and 10 low-cost nodes were available for 59 days from January 1, 2018 to March 31, 2018 ($PM_{2.5}$ averaged $138 \pm 31$ μg m$^{-3}$ among 22 reference stations), using a leave-one-out cross-validation (CV) over the 22 reference nodes. We showed that our approach can achieve an overall 30 % prediction error (RMSE: 33 μg m$^{-3}$) at a 24 h scale and is robust as underscored by the small variability in the GPR model parameters and in the model-produced calibration factors for the low-cost nodes among the 22-fold CV. Of the 22 reference stations, high-quality predictions were observed for those stations whose $PM_{2.5}$ means were close to the Delhi-wide mean (i.e., $138 \pm 31$ μg m$^{-3}$) and relatively poor predictions for those nodes whose means differed substantially from the Delhi-wide mean (particularly on the lower end). We also observed washed-out local variability in $PM_{2.5}$ across the 10 low-cost sites after calibration using our approach, which stands in marked contrast to the true wide variability across the reference sites. These observations revealed that our proposed technique (and more generally the geostatistical technique) requires high spatial homogeneity in the pollutant concentrations to be fully effective. We further demonstrated that our algorithm performance is insensitive to training window size as the mean prediction error rate and the standard error of the mean (SEM) for the 22 reference stations remained consistent at ~30 % and ~3–4 % when an increment of 2 days' data were included in the model training. The markedly low requirement of our algorithm for training data enables the models to

always be nearly most updated in the field, thus realizing the algorithm's full potential for dynamically surveilling large-scale WLPMSNs by detecting malfunctioning low-cost nodes and tracking the drift with little latency. Our algorithm presented similarly stable 26–34 % mean prediction errors and ~3–7 % SEMs over the sampling period when pre-trained on the current week's data and predicting 1 week ahead, therefore suitable for online calibration. Simulations conducted using our algorithm suggest that in addition to dynamic calibration, the algorithm can also be adapted for automated monitoring of large-scale WLPMSNs. In these simulations, the algorithm was able to differentiate malfunctioning low-cost nodes (due to either hardware failure or under heavy influence of local sources) within a network by identifying aberrant model-generated calibration factors (i.e., slopes close to zero and intercepts close to the Delhi-wide mean of true $PM_{2.5}$). The algorithm was also able to track the drift of low-cost nodes accurately within 4 % error for all the simulation scenarios. The simulation results showed that ~20 reference stations are optimum for our solution in Delhi and confirmed that low-cost nodes can extend the spatial precision of a network by decreasing the extent of pure interpolation among only reference stations. Our solution has substantial implications in reducing the amount of manual labor for the calibration and surveillance of extensive WLPMSNs, improving the spatial comprehensiveness of PM evaluation, and enhancing the accuracy of WLPMSNs.

## 1 Introduction

Low-cost air quality (AQ) sensors that report high time resolution data (e.g., ≤ 1 h) in near real time offer excellent potential for supplementing existing regulatory AQ monitoring networks by providing enhanced estimates of the spatial and temporal variabilities of air pollutants (Snyder et al., 2013). Certain low-cost particulate matter (PM) sensors demonstrated satisfactory performance benchmarked against Federal Equivalent Methods (FEMs) or research-grade instruments in some previous field studies (Holstius et al., 2014; Gao et al., 2015; SCAQMD, 2015a–b; Jiao et al., 2016; Kelly et al., 2017; Mukherjee et al., 2017; SCAQMD, 2017a–c; Crilley et al., 2018; Feinberg et al., 2018; Johnson et al., 2018; Zheng et al., 2018). Application-wise, low-cost PM sensors have had success in identifying urban fine particle ($PM_{2.5}$, with a diameter of 2.5 μm and smaller) hotspots in Xi'an, China (Gao et al., 2015), mapping urban air quality with additional dispersion model information in Oslo, Norway (Schneider et al., 2017), monitoring smoke from prescribed fire in Colorado, US (Kelleher et al., 2018), measuring a traveler's exposure to $PM_{2.5}$ in various microenvironments in Southeast Asia (Ozler et al., 2018), and building up a detailed city-wide temporal and spatial indoor $PM_{2.5}$ exposure profile in Beijing, China (Zuo et al., 2018).

On the down side, researchers have been plagued by calibration-related issues since the emergence of low-cost AQ sensors. One common brute force solution is initial calibration by collocation with reference analyzers before field deployment and follow-up with routine recalibration. Yet, the transferability of these pre-determined calibrations at collocation sites to new deployment sites is questionable as calibration factors typically vary with operating conditions such as PM mass concentrations, relative humidity (RH), temperature, and aerosol optical properties (Holstius et al., 2014; Austin et al., 2015; Wang et al., 2015; Lewis and Edwards, 2016; Crilley et al., 2018; Jayaratne et al., 2018; Zheng et al., 2018). Complicating

this further, the pre-generated calibration curves may only apply for a short term as the stability of low-cost sensors can develop drift or degrade over time (Lewis and Edwards, 2016; Jiao et al., 2016; Hagler et al., 2018). Routine recalibrations which require frequent transit of the deployed sensors between the field and the reference sites are not only too labor intensive for a large-scale network but also still cannot address the impact of urban heterogeneity of ambient conditions on calibration models (Kizel et al., 2018).

As such, calibrating sensors on-the-fly while they are deployed in the field is highly desirable. Takruri et al. (2009) showed that the Interacting Multiple Model (IMM) algorithm combined with the Support Vector Regression (SVR)-Unscented Kalman Filter (UKF) can automatically and successfully detect and correct low-cost sensor measurement errors in the field; however, the implementation of this algorithm still requires pre-deployment calibrations. Fishbain and Moreno-Centeno (2016) designed a self-calibration strategy for low-cost nodes with no need for collocation by exploiting the raw signal differences between all possible pairs of nodes. The learned calibrated measurements are the vectors whose pairwise differences are closest in normalized projected Cook-Kress (NPCK) distance to the corresponding pairwise raw signal differences given all possible pairs over all time steps. However, this strategy did not include reference measurements in the self-calibration procedure, and therefore the tuned measurements were still essentially raw signals (although instrument noise was dampened). An alternative calibration method involves chain calibration of the low-cost nodes in the field with only the first node calibrated by collocation with reference analyzers and the remaining nodes calibrated sequentially by their respective previous node along the chain (Kizel et al., 2018). While this node-to-node calibration procedure proved its merits in reducing collocation burden and data loss during calibration/relocation/recalibration and accommodating the influence of urban heterogeneity on calibration models, it is only suitable for relatively small networks because calibration errors propagate through chains and can inflate toward the end of a long chain (Kizel et al., 2018).

In this paper, we introduce a simultaneous Gaussian Process regression (GPR) and simple linear regression pipeline to calibrate $PM_{2.5}$ readings of any number of low-cost PM sensors on the fly in the field without resorting to pre-deployment collocation calibration by leveraging all available reference monitors across an area (e.g., Delhi, India). The proposed strategy is theoretically sound since the GPR (also known as Kriging) can capture the spatial covariance inherent in the data and has been widely used for spatial data interpolation (e.g., Holdaway, 1996; Di et al., 2016; Schneider et al., 2017) and the simple linear regression calibration can adjust for disagreements between low-cost sensor and reference instrument measurements and lead to more consistent spatial interpolation. This paper focuses on 1) quantifying experimentally the daily performance of our dynamic calibration model in Delhi during winter season based on model prediction accuracy on the holdout reference nodes during leave-one-out cross-validations (CV) and low-cost node calibration accuracy; 2) revealing the potential pitfalls of employing a dynamic calibration algorithm; 3) examining the sensitivity of our algorithm to the training data size and the feasibility of it for dynamic calibration; 4) demonstrating the ability of our algorithm to auto-detect faulty nodes and auto-correct the drift of nodes within a network via computational simulation, therefore the

practicality of adapting our algorithm for automated large-scale sensor network monitoring; and 5) studying computationally the optimal number of reference stations across Delhi to support our technique and the usefulness of low-cost sensors for extending the spatial precision of a sensor network. To the best of our knowledge, this is the first study to apply such a non-static calibration technique to a wireless low-cost PM sensor network in a heavily polluted region such as India and the first

to present methods of auto-monitoring dense AQ sensor networks.

## 2 Materials and methods

### 2.1 Low-cost node configuration

The low-cost packages used in the present study (dubbed "Atmos") shown in Fig. 1a were developed by Respirer Living Sciences (http://atmos.urbansciences.in/, last access: 30 November 2018) and cost ~ USD 300 per unit. The Atmos monitor

measures 20.3 cm L × 12.1 cm W × 7.6 cm H, weighs 500 g, and is housed in an IP65 (Ingress Protection rating 65) enclosure with a liquid crystal display (LCD) on the front showing real-time PM mass concentrations and various debugging messages. It includes a Plantower PMS7003 sensor (~ USD 25; dimension: 4.8 cm L × 3.7 cm W × 1.2 cm H) to measure $PM_1$, $PM_{2.5}$, and $PM_{10}$ mass concentrations, an Adafruit DHT22 sensor to measure temperature and relative humidity, and an ultra-compact Quectel L80 GPS model to retrieve accurate locations in real time. The operating principle and configuration

of PMS7003 are similar to its PMS1003, PMS3003, and PMS5003 counterparts and have been extensively discussed in previous studies (Kelly et al., 2017; Zheng et al., 2018; and Sayahi et al., 2018, respectively). The inlet and outlet of PMS7003 were aligned with two slots on the box to ensure unrestricted airflow into the sensor. The PM and meteorology data are read over the serial TTL interface every three seconds, aggregated every 1 min in memory on the device, before being transmitted by a Quectel M66 GPRS module through the mobile 2G cellular network to an online database. The Atmos

can also store the data on a local microSD card in case of transmission failure. Users have the option to configure the frequencies of data transfer and logging to 5, 10, 15, 30, and 60 minutes via a press key on the device and are able to view the settings on the LCD. All components of the Atmos monitors (key parts are labelled in Fig. 1b) are integrated to a custom-designed printed circuit board (PCB) which is controlled by a STMicroelectronics microcontroller (model STM32F051). Each Atmos was continuously powered up by a 5V 2A USB wall charger but also comes with a fail-safe 3.7V–2600 mAh

rechargeable Li-ion battery in case of power outage that can last up to 10 hours at a 1 min transmission frequency and 20 hours at a 5 min frequency.

The Atmos network's server architecture was also developed by Respirer Living Sciences and built on the following open-source components: KairosDB as the primary fast scalable time series database built on Apache Cassandra, custom-made

Java libraries for ingesting data and for providing XML/JSON/CSV-based access to aggregated time series data, HTML5/JavaScript for creating the front-end dashboard, and LeafletJS for visualizing Atmos networks on maps.

**2.2 Data description**

**2.2.1 Reference PM$_{2.5}$ data**

Hourly ground-level PM$_{2.5}$ concentrations from 21 monitoring stations operated by the Central Pollution Control Board (CPCB), the Delhi Pollution Control Committee (DPCC), the India Meteorological Department (IMD), and the Uttar Pradesh and Haryana States Pollution Control Boards (SPCBs) (https://app.cpcbccr.com/ccr/#/caaqm-dashboard/caaqm-landing, last access: 18 September 2018) and from one monitoring station operated by the U.S. Embassy in New Delhi (https://www.airnow.gov/index.cfm?action=airnow.global_summary#India$New_Delhi, last access: 18 September 2018) were available in our study domain of Delhi and its satellite cities including Gurgaon, Faridabad, Noida, and Ghaziabad from January 1, 2018 to March 31, 2018 (winter season) and were used as the reference measurements in our Delhi PM sensor network. The topographical, climatic, and air quality conditions of Delhi are well documented by Tiwari et al. (2012 and 2015) and Gorai et al. (2018). Briefly, Delhi experiences unusually high PM$_{2.5}$ concentrations over winter season due to a combination of increased biomass burning for heating, shallower boundary layer mixing height, diminished wet scavenging by precipitation, lower wind speed, and trapping of air pollutants by the Himalayan topology. Figure 2 visualizes the spatial distribution of these 22 reference monitors (triangle icons with *italic text*) and Table 1 lists their latitudes and longitudes. No station of the 22 reference monitors is known for regional background monitoring. The complex local built environment in Delhi arising from the densely and intensively mixed land use (Tiwari, 2002) and the significant contributions to air pollution from all vehicular, industrial (small scale industries and major power plants), commercial (diesel generators and tandoors), and residential (diesel generators and biomass burning) sectors (CPCB, 2009; Gorai et al., 2018) render the PM$_{2.5}$ concentrations relatively unconnected to the land-use patterns. We removed 104 1 h observations (labeled invalid and missing) from the U.S. Embassy dataset based on its reported QA/QC (quality assurance/quality control) remarks; however, the same procedure was not applied to the remaining 21 Indian government monitoring stations because neither the relevant Indian agencies provided QA/QC remarks or error flags in any of their regulatory monitoring stations' datasets nor can we obtain the QA/QC procedures (e.g., how and how often reference monitors are maintained and calibrated) for these reference monitors. Due to lack of relevant QA/QC information to exclude any measurement, all of the hourly PM$_{2.5}$ concentrations of the 21 monitoring stations operated by the Indian agencies were assumed to be correct. We would like to highlight this as a potential shortcoming of using the measurements from the Indian government monitoring stations. While mathematically the GPR model can operate without requiring data from all the stations to be non-missing on each day by relying on only each day's non-missing stations' covariance information to make inference, we practically required concurrent measurements of all the stations in this paper to drastically increase the speed of the algorithm (~10 mins to run a complete 22-fold leave-one-out CV, up to ~20 times faster) by avoiding the expensive computational cost of excessive amount of matrix inversions that can be incurred otherwise. We therefore linearly interpolated the 1 h PM$_{2.5}$ values for the hours with missing measurements for each station, after which we averaged the hourly data to daily resolution as the model inputs. We validate our

interpolation approach in Sect. 3.2.1 by showing that the model accuracies with and without interpolation are statistically the same.

### 2.2.2 Low-cost node PM$_{2.5}$ data

Hourly uncalibrated PM$_{2.5}$ measurements from 10 Atmos low-cost nodes across Delhi between January 1, 2018 and March 31, 2018 were downloaded from our low-cost sensor cloud platform. No correction or filter of any kind was applied to the raw signals of the low-cost nodes over the cloud platform before we downloaded the data. Figure 2 shows the sampling locations of these 10 low-cost nodes as circle icons and Table 1 specifies their latitudes and longitudes. In our current study, the factors governing the siting of these nodes consist of the ground contact personnel availability, the resource availability such as strong mobile network signal and 24/7 main power supply, the locations physical accessibility, and some other common criteria for sensor deployment (e.g., locations away from major pollution sources, situated in a place where free flow of air is available, and protected from vandalism and extreme weather). Similar to the preprocessing of the reference PM$_{2.5}$ data, we linearly interpolated the missing hourly PM$_{2.5}$ for each low-cost node and then aggregated the hourly data at a daily interval. The comparison of 1 h PM$_{2.5}$'s completeness before and after missing data imputation for both reference and low-cost nodes is detailed in Table 1 and the periods over which data were imputed for each site are illustrated in Fig. S1. There is no obvious pattern in the data missingness. To remove the prospective outliers such as erroneous surges/nadirs existing in the datasets of the 21 Indian government reference nodes and the 10 low-cost nodes or unreasonable interpolated measurements introduced during handling the missing data, we employed the Local Outlier Factor (LOF) algorithm with 20 neighbors considered (a number that works well in general) to remove a conservative ~10% of the 32-dimensional (22 reference + 10 low-cost nodes) 24 h PM$_{2.5}$ datasets. LOF is an unsupervised anomaly detection method that assigns each multi-dimensional data point an LOF score, defined as the ratio of the average local density of its k nearest neighboring data points (k = 20 in our study) to its own local density, to measure the relative degree of isolation of the given data point with respect to its neighbors (Breunig, et al., 2000). Normal observations tend to have LOF scores near 1 while outliers have scores significantly larger than 1. The LOF therefore identifies the outliers as those multi-dimensional observations with the top x% (x = 10 in our study) LOF scores. A total of 59 days' PM$_{2.5}$ measurements common to all 32 nodes in the network were left (see Fig. S1) and used for our model evaluation.

### 2.3 Simultaneous GPR and simple linear regression calibration model

The simultaneous GPR and simple linear regression calibration algorithm is introduced here as Algorithm 1. The critical steps of the algorithm are linked to sub-sections under which the respective details can be found. Complementing Algorithm 1, a flow diagram illustrating the algorithm is given in Figure 3.

**Algorithm 1: Algorithm of simultaneous GPR and simple linear regression**

**for** each reference node (denote: Ref$_k$) in the network **do**

    leave Ref$_k$ out as test sample (see Sect. 2.3.1 for details)

    **for** each low-cost node (denote: Low-cost$_i$) in the network **do**

        find Low-cost$_i$'s closest reference node (denote: Ref$_i$) (Sect. 2.3.2)

5        fit a simple linear regression model between Ref$_i$ and Low-cost$_i$'s PM$_{2.5}$: $\boldsymbol{Ref}_i = \alpha_i \cdot \boldsymbol{Low-cost}_i + \beta_i$ (Sect. 2.3.2)

        initialize the simple linear regression calibration factors to $\alpha_i$ (slope) and $\beta_i$ (intercept) for Low-cost$_i$ (Sect. 2.3.2)

        initialize the calibration of Low-cost$_i$ using $\alpha_i$ and $\beta_i$ (Sect. 2.3.2)

    **end for**

    initialize GPR hyperparameters $\boldsymbol{\Theta} = [\sigma_s^2, l, \sigma_n^2]$ to [0.1, 50, 0.01] (Sect. 2.3.3)

10    standardize the 10 calibrated low-cost and 21 reference nodes at once (Sect. 2.3.3)

    **while** convergence criteria not met **do**

        update/optimize GPR hyperparameters $\boldsymbol{\Theta}$ using the 31 standardized training nodes (Sect. 2.3.3 and .5)

        **for** each low-cost node (denote: Low-cost$_i$) in the network **do**

            **for** each day (denote: t) of the 59 days **do**

calculate Low-cost$_i$'s mean conditional on the remaining 30 nodes on day t (denote $\mu_{A|B}^{it}$) (Sect. 2.3.4 and .5)

            **end for**

        fit a linear regression between $\boldsymbol{\mu}_{A|B}^i \in \mathbb{R}^{59}$ and Low-cost$_i$: $\boldsymbol{\mu}_{A|B}^i = \alpha_i \cdot \boldsymbol{Low-cost}_i + \beta_i$ (Sect. 2.3.4 and .5)

        update calibration factors $\alpha_i$ and $\beta_i$ for Low-cost$_i$ (Sect. 2.3.4 and .5)

        update the calibration of Low-cost$_i$ using $\alpha_i$ and $\beta_i$ (Sect. 2.3.4 and .5)

**end for**

        check convergence criteria (Sect. 2.3.5)

    **end while**

    use the final GPR model to predict on Ref$_k$ (Sect. 2.3.6)

    transform the prediction back to original PM$_{2.5}$ scale (Sect. 2.3.6)

calculate RMSE and percent error (Sect. 2.3.6)

**end for**

### 2.3.1 Leave one reference node out

Because the true calibration factors for the low-cost nodes are not known beforehand, a leave-one-out CV approach (i.e., holding one of the 22 reference nodes out of modelling each run for model predictive performance evaluation) was adopted

as a surrogate to estimate our proposed model accuracy of calibrating the low-cost nodes. For each of the 22-fold CV, 31 node locations (denoted $\Gamma = \{\boldsymbol{x}_1, \dots, \boldsymbol{x}_{31}\}$) were available, where $\boldsymbol{x}_i$ is the latitude and longitude of node $i$. Let $y_{it}$ represent the daily PM$_{2.5}$ measurement of node $i$ on day $t$ and $\boldsymbol{y}_t \in \mathbb{R}^{31}$ denote the concatenation of the daily PM$_{2.5}$ measurements

recorded by the 31 nodes on day $t$. Given a finite number of node locations, a Gaussian Process (GP) becomes a Multivariate Gaussian Distribution over the nodes in the form of:

$$\boldsymbol{y}_t|\Gamma \sim N(\boldsymbol{\mu}, \boldsymbol{\Sigma}) \tag{1}$$

where $\boldsymbol{\mu} \in \mathbb{R}^{31}$ represents the mean function (assumed to be $\boldsymbol{0}$ in this study); $\boldsymbol{\Sigma} \in \mathbb{R}^{31 \times 31}$ with $\Sigma_{ij} = K(\boldsymbol{x}_i, \boldsymbol{x}_j; \boldsymbol{\Theta})$ represents

the covariance function/kernel function and $\boldsymbol{\Theta}$ is a vector of the GPR hyperparameters.

For simplicity's sake, the kernel function was set to a squared exponential (SE) covariance term to capture the spatially-correlated signals coupled with another component to constrain the independent noise:

$$K(\boldsymbol{x}_i, \boldsymbol{x}_j; \boldsymbol{\Theta}) = \sigma_s^2 \, exp\left(-\frac{\|x_i - x_j\|_2^2}{2l^2}\right) + \sigma_n^2 \boldsymbol{I} \quad \text{(Rasmussen and Williams, 2006)} \tag{2}$$

where $\sigma_s^2$, $l$, and $\sigma_n^2$ are the model hyperparameters (to be optimized) that control the signal magnitude, characteristic length-scale, and noise magnitude, respectively; $\boldsymbol{\Theta} \in \mathbb{R}^3$ is a vector of the GPR hyperparameters $\sigma_s^2$, $l$, and $\sigma_n^2$

### 2.3.2 Initialize low-cost nodes' (simple linear regression) calibrations

What separates our method from standard GP applications is the simultaneous incorporation of calibration for the low-cost nodes using a simple linear regression model into the spatial model. Linear regression has previously been shown to be

effective at calibrating PM sensors (Zheng et al., 2018). Linear regression was first used to initialize low-cost nodes' calibrations (step two in Fig. 3). In this step, each low-cost node $i$ was linearly calibrated to its closest reference node using Eq. 3, where the calibration factors $\alpha_i$ (slope) and $\beta_i$ (intercept) were determined by fitting a simple linear regression model to all available pairs of daily PM$_{2.5}$ mass concentrations from the uncalibrated low-cost node $i$ (independent variable) and its closest reference node (dependent variable). This step aims to bridge disagreements between low-cost and reference node

measurements, which can lead to a more consistent spatial interpolation and a faster convergence during the GPR model optimization.

$$\boldsymbol{r}_i = \begin{cases} \boldsymbol{y}_i, & \text{if reference node} \\ \alpha_i \cdot \boldsymbol{y}_i + \beta_i, & \text{if low} - \text{cost node} \end{cases} \tag{3}$$

where $\boldsymbol{y}_i$ is either a vector of all the daily PM$_{2.5}$ measurements of reference node $i$ or a vector of all the daily raw PM$_{2.5}$ signals of low-cost node $i$; $\boldsymbol{r}_i$ is either a vector of all the daily PM$_{2.5}$ measurements of reference node $i$ or a vector of all the

daily calibrated PM$_{2.5}$ measurements of low-cost node $i$; $\alpha_i$ and $\beta_i$ are the slope and intercept, respectively, determined from the fitted simple linear regression calibration equation with daily PM$_{2.5}$ mass concentrations of the uncalibrated low-cost node $i$ as independent variable and PM$_{2.5}$ mass concentrations of low-cost node $i$'s closest reference node as dependent variable.

### 2.3.3 Optimize GPR model (hyperparameters)

In the next step (step three in Fig. 3), a GPR model was fit to each day $t$'s 31 nodes (i.e., 10 initialized low-cost nodes and 21 reference nodes) as described in Eq. (4). Prior to the GPR model fitting, all the PM$_{2.5}$ measurements of the 31 nodes over 59 valid days used for GPR model hyperparameters training were standardized. The standardization was performed by first concatenating all these training PM$_{2.5}$ measurements (from the 31 nodes over 59 days), then subtracting their mean $\mu_{training}$ and dividing them by their standard deviation $s_{training}$ (i.e., transforming all the training PM$_{2.5}$ measurements to have a zero mean and unit variance). It is worth noting that assuming the mean function $\boldsymbol{\mu} \in \mathbb{R}^{31}$ to be $\boldsymbol{0}$ along with standardizing all the training PM$_{2.5}$ samples in this study is one of the common modelling formulations on the GPR model and the simplest one. More complex formulations including a station-specific mean function (lack of prior information for this project), a time-dependent mean function (computationally expensive), and a combination of both were not considered for this paper. After the standardization of training samples, the GPR was trained to maximize the log marginal likelihood over all 59 days using Eq. 5 and using an L-BFGS-B optimizer (Byrd et al., 1994). To avoid bad local minima, several random hyperparameter initializations were tried and the initialization that resulted in the largest log marginal likelihood after optimization was chosen (in this paper, $\boldsymbol{\Theta} = [\sigma_s^2, l, \sigma_n^2]$ was initialized to [0.1, 50, 0.01]).

$$\boldsymbol{r}_t | \Gamma \sim N(\boldsymbol{\mu}, \boldsymbol{\Sigma}) \tag{4}$$

where $t$ ranges from 1 (inclusive) to 59 (inclusive); $\boldsymbol{r_t} \in \mathbb{R}^{31}$ is a vector of all 31 nodes' PM$_{2.5}$ measurements (calibrated if low-cost nodes) on day $t$; $\Gamma = \{\boldsymbol{x_1}, \ldots, \boldsymbol{x_{31}}\}$ denotes 31 nodes' locations and $\boldsymbol{x_i} \in \mathbb{R}^2$ is a vector of the latitude and longitude of node $i$; $\boldsymbol{\mu} \in \mathbb{R}^{31}$ represents the mean function (assumed to be $\boldsymbol{0}$ in this study) and $\boldsymbol{\Sigma} \in \mathbb{R}^{31\times31}$ with $\Sigma_{ij} = K(\boldsymbol{x_i}, \boldsymbol{x_j}; \boldsymbol{\Theta})$ represents the covariance function/kernel function.

$$\arg\max_{\boldsymbol{\Theta}} L(\boldsymbol{\Theta}) = \arg\max_{\boldsymbol{\Theta}} \sum_{t=1}^{59} \log p(\boldsymbol{r_t}|\boldsymbol{\Theta}) = \arg\max_{\boldsymbol{\Theta}} \left(-0.5 \cdot 59 \cdot \log|\boldsymbol{\Sigma}_\theta| - 0.5 \sum_{t=1}^{59} \boldsymbol{r_t}^T \boldsymbol{\Sigma}_\theta^{-1} \boldsymbol{r_t}\right) \tag{5}$$

where $\boldsymbol{\Theta} \in \mathbb{R}^3$ is a vector of the GPR hyperparameters $\sigma_s^2$, $l$, and $\sigma_n^2$.

### 2.3.4 Update low-cost nodes' (simple linear regression) calibrations based on their conditional means

Once the optimum $\boldsymbol{\Theta}$ for the (initial) GPR was found, we used the learned covariance function to find the mean of each low-cost node $i$'s Gaussian Distribution conditional on the remaining 30 nodes within the network (i.e., $\mu_{A|B}^{it}$) on day $t$ as described mathematically in Eq. (6)–(8) and repeatedly did so until all 59 days' $\mu_{A|B}^{it}$ (i.e., $\boldsymbol{\mu_{A|B}^i} \in \mathbb{R}^{59}$) were found and then re-calibrated that low-cost node $i$ based on the $\boldsymbol{\mu_{A|B}^i}$. The re-calibration was done by first fitting a simple linear regression model to all 59 pairs of daily PM$_{2.5}$ mass concentrations from the uncalibrated low-cost node $i$ ($\boldsymbol{y_i}$, independent variable) and its conditional mean ($\boldsymbol{\mu_{A|B}^i}$, dependent variable) and then using the updated calibration factors (slope $\alpha_i$ and intercept $\beta_i$) obtained from this newly fitted simple linear regression calibration model to calibrate the low-cost node $i$ again (using Eq. 3). This procedure is summarized graphically in Fig. 3 step four and was performed iteratively for all low-cost nodes one at a

time. The reasoning behind this step is given in the supplement. A high-level interpretation of this step is that the target low-cost node is calibrated by being weighted over the remaining nodes within the network and the $\Sigma_{AB}^{it}\Sigma_{BB}^{it}{}^{-1}$ term computes the weights. In contrast to the inverse distance weighting interpolation which will weight the nodes used for calibration equally if they are equally distant from the target node, the GPR will value sparse information more and lower the importance of redundant information (suppose all the nodes are equally distant from the target node) as shown in Fig. S2.

$$p\left(\begin{bmatrix} r_A^{it} \\ \boldsymbol{r}_B^{it} \end{bmatrix}\right) = N\left(\begin{bmatrix} r_A^{it} \\ \boldsymbol{r}_B^{it} \end{bmatrix}; \begin{bmatrix} \mu_A^{it} \\ \boldsymbol{\mu}_B^{it} \end{bmatrix} \begin{bmatrix} \Sigma_{AA}^{it} & \Sigma_{AB}^{it} \\ \Sigma_{BA}^{it} & \Sigma_{BB}^{it} \end{bmatrix}\right) \tag{6}$$

$$r_A^{it}|\boldsymbol{r}_B^{it} \sim N(\mu_{A|B}^{it}, \Sigma_{A|B}^{it}) \tag{7}$$

$$\mu_{A|B}^{it} = \mu_A^{it} + \Sigma_{AB}^{it}\Sigma_{BB}^{it}{}^{-1}(\boldsymbol{r}_B^{it} - \boldsymbol{\mu}_B^{it}) \tag{8}$$

where $r_A^{it}$ and $\boldsymbol{r}_B^{it}$ are the daily PM$_{2.5}$ measurement(**s**) of the low-cost node $i$ and the remaining 30 nodes on day $t$; $\mu_A^{it}$, $\boldsymbol{\mu}_B^{it}$, and $\mu_{A|B}^{it}$ are the mean (**vector**) of the partitioned Multivariate Gaussian Distribution of the low-cost node $i$, the remaining 30 nodes, and the low-cost node $i$ conditional on the remaining 30 nodes, respectively, on day $t$; and $\Sigma_{AA}^{it}$, $\boldsymbol{\Sigma}_{AB}^{it}$, $\boldsymbol{\Sigma}_{BA}^{it}$, $\boldsymbol{\Sigma}_{BB}^{it}$, and $\Sigma_{A|B}^{it}$ are the covariance between the low-cost node $i$ and itself, the low-cost node $i$ and the remaining 30 nodes, the remaining 30 nodes and the low-cost node $i$, the remaining 30 nodes and themselves, and the low-cost node $i$ conditional on the remaining 30 nodes and itself, respectively, on day $t$.

## 2.3.5 Optimize alternately and iteratively and converge

Iterative optimizations alternated between the GPR hyperparameters and the low-cost node calibrations using the approaches described in Sect. 2.3.3 and 2.3.4, respectively (Fig. 3 steps five and six, respectively), until the GPR parameters **Θ** converged with the convergence criteria being the differences in all the GPR hyperparameters between the two adjacent runs below 0.01 (i.e., with $\Delta\sigma_s^2 \leq 0.01, \Delta l \leq 0.01, and \, \Delta\sigma_n^2 \leq 0.01$).

## 2.3.6 Predict on the holdout reference node and calculate accuracy metrics

The final GPR was used to predict the 59-day PM$_{2.5}$ measurements of the holdout reference node (Fig. 3 step seven) following the Cholesky decomposition algorithm (Rasmussen and Williams, 2006) with the standardized predictions being transformed back to the original PM$_{2.5}$ measurement scale at the end. The back transformation was done by multiplying the predictions by the standard deviation $s_{training}$ (the standard deviation of the training PM$_{2.5}$ measurements) and then adding back the mean $\mu_{training}$ (the mean of the training PM$_{2.5}$ measurements). Metrics including root mean square errors (RMSE, Eq. 9) and percent errors defined as RMSE normalized by the average of the true measurements of the holdout reference node in this study (Eq. 10) were calculated for each fold and further averaged over all 22 folds to assess the accuracy and sensitivity of our simultaneous GPR and simple linear regression calibration model.

$$\text{RMSE} = \sqrt{\frac{1}{59}\|\boldsymbol{y}_i - \widehat{\boldsymbol{y}}_i\|_2^2} \tag{9}$$

where $\boldsymbol{y}_i$ and $\hat{\boldsymbol{y}}_i$ are the true and model predicted 59 daily PM$_{2.5}$ measurements of the holdout reference node $i$.

$$\text{Percent error} = \frac{\text{RMSE}}{\text{avg. holdout reference PM}_{2.5}\text{ conc.}} \tag{10}$$

## 3 Results and discussion

### 3.1 Spatial variation of PM$_{2.5}$ across Delhi

Figure 4a presents the box plot of the daily averaged PM$_{2.5}$ at each available reference site across Delhi from January 1, 2018 to March 31, 2018. The Vasundhara and DTU sites were the most polluted stations with the PM$_{2.5}$ averaging 194 ± 104 µg m$^{-3}$ and 193 ± 90 µg m$^{-3}$, respectively. The Pusa and Sector 62 sites had the lowest mean PM$_{2.5}$, averaging 86 ± 40 µg m$^{-3}$ and 88 ± 36 µg m$^{-3}$, respectively. The Delhi-wide average of the 3-month mean PM$_{2.5}$ across the 22 reference stations was found to be 138 ± 31 µg m$^{-3}$. This pronounced spatial variation in mean PM$_{2.5}$ in Delhi (as reflected by the high SD of 31 µg m$^{-3}$) coupled with the stronger temporal variation for each station even at a 24 h scale (range: 35–104 µg m$^{-3}$, see Fig. 4a) caused nonuniform calibration performance of the GPR model across Delhi, as detailed in Sect. 3.2.

### 3.2 Assessment of GPR model performance

The optimum values of the GPR model parameters including the signal variance ($\sigma_s^2$), the characteristic length-scale ($l$), and the noise variance ($\sigma_n^2$) are shown in Fig. S3. The $\sigma_s^2$, $l$, and $\sigma_n^2$ from the 22-fold leave-one-out CV averaged 0.53 ± 0.02, 97.89 ± 5.47 km, and 0.47 ± 0.01, respectively. The small variability in all the parameters among all the folds indicates that the model is fairly robust to the different combinations of reference nodes. The learned length-scale can be interpreted as the modeled spatial pattern of PM$_{2.5}$ being relatively consistent within approximately 98 km, suggesting that the optimized model majorly captures a regional trend rather than fine-grained local variations in Delhi.

### 3.2.1 Accuracy of reference node prediction

We start by showing the accuracy of model prediction on the 22 reference nodes using leave-one-out CV (when the low-cost node measurements were included in our spatial prediction). Without any prior knowledge of the true calibration factors for the low-cost nodes, the holdout reference node prediction accuracy is a statistically sound proxy for estimating how well our technique can calibrate the low-cost nodes. The performance scores (including RMSE and percent error) for each reference station sorted by the 3-month mean PM$_{2.5}$ in descending order are listed in Table 2. An overall 30 % prediction error (equivalent to an RMSE of 33 µg m$^{-3}$) at a 24 h scale was achieved on the reference nodes following our calibration procedure. In this paper, we reported our algorithm's accuracy on the 24 h data only rather than on the 1 h data because real-time reference monitors that are certified as the Federal Equivalent Methods (FEMs) by the US Environmental Protection Agency (EPA) are required to provide results comparable to the Federal Reference Methods (FRMs) only for a 24 h but not

a 1 h sampling period. Our algorithm, which essentially relies on the accuracy of the reference measurements, can only calibrate/predict as well as the reference methods measure. Therefore, only the percent error based on the reliable 24 h reference measurements is a fair representation of our algorithm's true calibration/prediction ability. Although the technique is reasonably accurate, especially considering the minimal amount of field work involved, its calibration error is nearly 3 times higher than the one of the low-cost nodes that were well calibrated by collocation with an environmental β-attenuation monitors (E-BAM) in our previous study (error: 11 %; RMSE: 13 μg m$^{-3}$) under similar $PM_{2.5}$ concentrations at the same temporal resolution (Zheng et al., 2018). The suboptimal on-the-fly mapping accuracy is a result of the optimized model's ability to simulate only the regional trend well. From a different perspective, the GPR method would have modeled the spatial pattern of $PM_{2.5}$ in Delhi well had the natural spatial covariance among the nodes not been disturbed by the complex and prevalent local sources there. As a substantiation of the flawed local $PM_{2.5}$ variation modelling, the reference node mapping accuracy follows a pattern, with relatively high-quality prediction for those nodes whose means were close to the Delhi-wide mean (e.g., Delhi-wide mean ± SD as highlighted with shading in Table 2) and relatively poor prediction for those nodes whose means differed substantially from the Delhi-wide mean (particularly on the lower end).

In this paper, we interpolated the missing 1 h $PM_{2.5}$ values for all the reference and low-cost stations to fulfil our requirement of concurrent measurements of all the stations. This approach drastically increased the speed of the algorithm (up to ~20 times faster) by avoiding the expensive computational cost of excessive amount of matrix inversions that can be incurred from relying on only each day's non-missing stations' covariance information to make inference. Here we prove that the interpolation is an appropriate methodology for this paper by demonstrating that the model prediction percent errors for the 22 reference stations with and without interpolation are statistically the same. The comparison of the errors for each station can be found in Table S1. Table S1 shows that the percent errors for all the stations are essentially the same with only one exception of station Vasundhara whose error without interpolation is 10 % lower than that with interpolation. The Delhi-wide mean percent errors with (30 %) and without interpolation (29 %) are also essentially the same. We further used the Wilcoxon signed-rank test (Wilcoxon, 1945) to prove that the two related paired samples (i.e., the percent errors for the 22 reference stations with and without interpolation) are indeed statistically the same. The Wilcoxon signed-rank test is a non-parametric version of the parametric paired t-test (involving two related/matched samples/groups) that requires no specific distribution on the measurements (unlike the parametric paired t-test that assumes a normal distribution). We conducted a two-sided test which has the null hypothesis that the percent errors for the 22 reference stations with and without interpolation are the same (i.e., $H_0$: with = without) against the alternative that they are not the same (i.e., $H_1$: with ≠ without). The p-value of the test is 0.07. The level of statistical significance was chosen to be 0.05, which means that the null hypothesis (i.e., $H_0$: with = without) cannot be rejected when the p-value is 0.07, above 0.05. Therefore, interpolating missing 1 h $PM_{2.5}$ data for both reference and low-cost nodes is appropriate for this paper because the accuracies of model prediction on the 22 reference nodes with and without interpolation are not distinct based on the Wilcoxon signed-rank test result.

It is of particular interest to validate the value of establishing a relatively dense wireless sensor network in Delhi by examining if the addition of the low-cost nodes can truly lend a performance boost to the spatial interpolation among sensor locations. We juxtapose the interpolation performance using the full sensor network (including both the reference and low-cost nodes) with that using only the reference nodes in Fig. 5. In this context, the unnormalized RMSE is less representative than the percent error of the model interpolation performance because of the unequal numbers of overlapping 24 h observations for all the nodes (59 data points) and for only the reference nodes (87 data points). The comparison revealed that the inclusion of the 10 low-cost devices on top of the regulatory grade monitors can reduce mean and median interpolation error by roughly 2 %. While only a marginal improvement with 10 low-cost nodes in the network, the outcome hints that densely-deployed low-cost nodes can have great promise of significantly decreasing the amount of pure interpolation among sensor locations, therefore benefitting the spatial precision of a network. We will explore more about the significance of the low-cost nodes for the network performance in Sect. 3.3.3.

### 3.2.2 Accuracy of low-cost node calibration

Next we describe the technique's accuracy of low-cost node calibration. The model-produced calibration factors are shown in Fig. 6. The intercepts and slopes for each unique low-cost device varied little among all the 22 CV folds, reiterating the stability of the GPR model. The values of these calibration factors resemble those obtained in the previous field work, with slopes comparable to South Coast Air Quality Management District's evaluations on the Plantower PMS models (SCAQMD, 2017a–c) and intercepts comparable to our Kanpur, India post-monsoon study (Zheng et al., 2018).

Two low-cost nodes (i.e., MRU and IITD) were collocated with two E-BAMs throughout the entire study. This allows us to take their model-derived calibration factors and calibrate the corresponding raw values of the low-cost nodes before computing the calibration accuracy based on the ground truth (i.e., E-BAM measurements). Figures 7a and 7b show the scatterplots of the collocated E-BAM measurements against the model-calibrated low-cost nodes at the MRU and the IITD sites, respectively. The two sites had similarly large calibration errors (~50 %) because their concentrations were both near the lower end of $PM_{2.5}$ spectrum in Delhi. These high error rates echo the conditions found at the comparatively clean Pusa and Sector 62 reference sites. The scatterplots also reveal the reason why the technique especially has trouble calibrating low-concentration sites—the technique overpredicted the $PM_{2.5}$ concentrations at the low-concentration sites to match the levels as if subject to the natural spatial variation. The washed-out local variability after model calibration more obviously manifests in Fig. 4b, which stands in marked contrast to the true wide variability across the reference sites (Fig. 4a). In other words, the geostatistical techniques can calibrate the low-cost nodes dynamically, with the important caveat that it is effective only if the degree of urban homogeneity in $PM_{2.5}$ is high (e.g., the local contributions are as small a fraction of the regional ones as possible or the local contributions are prevalent but of similar magnitudes). Otherwise, quality predictions will only apply for those nodes whose means are close to the Delhi-wide mean. Gani et al. (2019) estimated that Delhi's

local contribution to the composition-based submicron particulate matter ($PM_1$) was ~30 to 50 % during winter and spring months. Clearly the huge amount of local influence in Delhi did not fully support our technique.

### 3.2.3 GPR model performance as a function of training window size

So far, the optimization of both GPR model hyperparameters and the linear regression calibration factors for the low-cost nodes has been carried out over the entire sampling period using all 59 available daily-averaged data points. It is of critical importance to examine the effect of time history on the algorithm, by analyzing how sensitive the model performance is to training window size. We tracked the model performance change when an increment of 2 days' data were included in the model training. The model performance was measured by the mean accuracy of model prediction on the 22 reference nodes (within the time period of the training window) using leave-one-out CV, as described in Sect. 3.2.1. Figure 8 illustrates that, throughout the 59 days, the error rate and the standard error of the mean (SEM) remained surprisingly consistent at ~30 % and ~3–4 %, respectively, regardless of how many 2-day increments were used as the training window size. The little influence of training window size on the GPR model performance is possibly a positive side effect of the algorithm's time-invariant mean assumption, strong spatial smoothing effect, and the additional averaging of the error rates of the 22 reference nodes. The markedly low requirement of our algorithm for training data is powerful in that it enables the GPR model hyperparameters and the linear regression calibration factors to always be nearly most updated in the field. This helps realize the algorithm's full potential for automatically surveilling large-scale networks by detecting malfunctioning low-cost nodes within a network (see Sect. 3.3.1) and tracking the drift of low-cost nodes (see Sect. 3.3.2) with as little latency as possible.

### 3.2.4 GPR model dynamic calibration performance

The stationary model performance in response to the increase of training data hints that using our method for dynamic calibration/prediction is feasible. We assessed the algorithm's 1 week-ahead prediction performance, by using simple linear regression calibration factors and GPR hyperparameters that were optimized from one week to calibrate the 10 low-cost nodes and predict each of the 22 reference nodes, respectively, in the next week. For example, the first/second/third/… week data were used as training data to build GPR models and simple linear regression models. These simple linear regression models were then used to calibrate the low-cost nodes in the second/third/fourth/… week, followed by the GPR models to predict each of the 22 reference nodes in that week. The performance was still measured by the mean accuracy of model prediction on the 22 reference nodes using leave-one-out CV, as described in Sect. 3.2.1. We found similarly stable 26–34 % dynamic calibration error rates and ~3–7 % SEMs throughout the weeks (see Figure S4).

### 3.2.5 RH adjustment to the algorithm

We attempted RH adjustment to the algorithm by incorporating an RH term in the linear regression models, where the RH values were the measurements from each corresponding low-cost sensor package's embedded Adafruit DHT22 RH and

temperature sensor. However, there was no improvement in the algorithm's accuracy after RH correction. A plausible explanation is regarding the infrequently high RH conditions during the winter months in Delhi and stronger smoothing effects at longer averaging time intervals (i.e., 24 h). Our previous work (Zheng et al., 2018) suggested that the PMS3003 $PM_{2.5}$ weights exponentially increased only when RH was above ~70%. The Delhi-wide average of the 3-month RH measured by the 10 low-cost sites was found to be 55 ± 15 %. Only 17 % and 6 % of these RH values were greater than 70 % and 80 %, respectively. The infrequently high RH conditions can cause the RH-induced biases insignificant. Additionally, our previous work found that even though major RH influences can be found in 1 min to 6 h $PM_{2.5}$ measurements, the influence significantly diminished in 12 h $PM_{2.5}$ measurements and was barely observable in 24 h measurements. Therefore, longer averaging time intervals can smooth out the RH biases.

Additionally, while our algorithm was analyzed over the 59 available days in this study, the daily-averaged temperature and RH measurements for the entire sampling period (i.e., from January 1 to March 31, 2018, 90 days) were statistically the same as those for the 59 days. To support this statement, we conducted the Wilcoxon rank-sum test, also called Mann-Whitney U test (Wilcoxon, 1945; Mann and Whitney, 1947) on the daily-averaged temperature and RH measurements from the Indira Gandhi International (IGI) Airport. The Wilcoxon rank-sum test is a non-parametric version of the parametric t-test (involving two independent samples/groups) that requires no specific distribution on the measurements (unlike the parametric t-test that assumes a normal distribution). We did not use a paired test here because the two groups had different sample sizes (i.e., 59 and 90, respectively). We conducted a two-sided test which has the null hypotheses that the daily-averaged temperature and RH measurements for the 90 days (19 ± 5 °C, 59 ± 14 %) and the 59 days (20 ± 5 °C, 59 ± 12 %) were the same (i.e., $H_0$: $Temperature_{59\ days}$ = $Temperature_{90\ days}$ / $RH_{59\ days}$ = $RH_{90\ days}$) against the alternatives that they were not the same (i.e., $H_1$: $Temperature_{59\ days}$ ≠ $Temperature_{90\ days}$ / $RH_{59\ days}$ ≠ $RH_{90\ days}$). The p-values for the temperature and RH comparisons are 0.28 and 0.59, respectively. The level of statistical significance was chosen to be 0.05, which means that the null hypotheses (i.e., $H_0$: $Temperature_{59\ days}$ = $Temperature_{90\ days}$ / $RH_{59\ days}$ = $RH_{90\ days}$) cannot be rejected when the p-values are both above 0.05. Therefore, the daily-averaged temperature and RH measurements from the IGI Airport for the entire sampling period and for the 59 days were not statistically distinct.

### 3.3 Simulation results

While the exact values of the calibration factors derived from the GPR model fell short of faithfully recovering the original picture of $PM_{2.5}$ spatiotemporal gradients in Delhi, these values of one low-cost node relative to another in the network (Sect. 3.3.1) or relative to itself over time (Sect. 3.3.2) turned out to be useful in facilitating automated large-scale sensor network monitoring.

### 3.3.1 Simulation of low-cost node failure or under heavy influence of local sources

One way to simulate the conditions of low-cost node failure or under heavy influence of local sources is to replace their true signals with values from random number generators so that the inherent spatial correlations are corrupted. In this study, we simulated how the model-produced calibration factors change when all (10), nine, seven, three, and one of the low-cost nodes within the network malfunction or are subject to strong local disturbance. We have three major observations from evaluating the simulation results (Fig. 9 and Fig. S5). First, the normal calibration factors are quite distinct from those of the low-cost nodes with random signals. Compared to the normal values (see Fig. 9 bottom right panel), the ones of the low-cost nodes with random signals have slopes close to 0 and intercepts close to the Delhi-wide mean of true $PM_{2.5}$ in Delhi (most clearly shown in Fig. 9 top left panel). Second, the calibration factors of the normal low-cost nodes are not affected by the aberrant nodes within the network (see Fig. 9 top right, middle left, middle right, and bottom left panels). These two observations indicate that the GPR model enables automated and streamlined process of instantly spotting any malfunctioning low-cost nodes (due to either hardware failure or under heavy influence of local sources) within a large-scale sensor network. Third, the performance of the GPR model seems to be rather uninfluenced by changing the true signals to random numbers (see Fig. S5, 33 % error rate when all low-cost nodes are random vs. baseline 30 % error rate). One possible explanation is that the prevalent and intricate air pollution sources in Delhi have already dramatically weakened the natural spatial correlations. This means that a significant degree of randomness has already been imposed on the low-cost nodes in Delhi prior to our complete randomness experiment. It is worth mentioning that flatlining is another commonly seen failure mode of our low-cost PM sensors in Delhi. The raw signals of such malfunctioning PM sensors were observed to flatline at the upper end of the sensor output values (typically thousands of $\mu g\ m^{-3}$). The very distinct signals of these flatlining low-cost PM nodes, however, make it rather easy to separate them from the rest of the nodes and filter them out at the early pre-processing stage before analyses, therefore without having to resort to our algorithm. Nevertheless, our not so accurate on-the-fly calibration model has created a useful algorithm for supervising large-scale sensor networks in real time as a by-product.

### 3.3.2 Simulation of low-cost node drift

We further investigated the feasibility of applying the GPR model to track the drift of low-cost nodes accurately over time. We simulated drift conditions by first setting random percentages of intercept and slope drift, respectively, for each individual low-cost node and for each simulation run. Next, we adjusted the signals of each low-cost node over the entire study period given these randomly selected percentages using Eq. (11). Then, we rebuilt a GPR model based on these drift-adjusted signals and evaluated if the new model-generated calibration factors matched our expected predetermined percentage drift relative to the true (baseline) calibration factors.

$$y_{i\_drift} = \frac{y_i}{(1 - \text{percentage slope drift}_i)} + \frac{\text{percentage intercept drift}_i \cdot \text{true intercept}_i}{(1 - \text{percentage slope drift}_i) \cdot \text{true slope}_i} \tag{11}$$

where $y_i$, true intercept$_i$, true slope$_i$, percentage intercept drift$_i$, percentage slope drift$_i$, and $y_{i\_drift}$ are a vector of the true signals, the standard model-derived intercept, the standard model-derived slope, the randomly generated percentage of intercept drift, the randomly generated percentage of slope drift, and a vector of the drift-adjusted signals, respectively, over the full study period for low-cost node $i$.

The performance of the model for predicting the drift was examined under a variety of scenarios including assuming that all (10), eight, six, four, and two of the low-cost nodes developed various degrees of drift such as significant (11 %–99 %), marginal (1 %–10 %), and a balanced mixture of significant and marginal. The testing results for 10, six, and two low-cost nodes are displayed in Table 3 and those for eight and four nodes are in Table S2. Overall, the model demonstrates excellent

drift predictive power with less than 4 % errors for all the simulation scenarios. The model proves to be most accurate (within 1 % error) when low-cost nodes only drifted marginally regardless of the number of nodes drift. In contrast, significant and particularly a mixture of significant and marginal drifts might lead to marginally larger errors. We also notice that the intercept drifts are slightly harder to accurately capture than the slope drifts. Similar to the simulation of low-cost node failure/under strong local impact as described in Sect 3.3.1, the performance of the model for predicting the

measurements of the 22 holdout reference nodes across the 22-fold leave-one-out CV was untouched by the drift conditions (see Fig. S6). This unaltered performance can be attributable to the fact that the drift simulations only involve simple linear transformations as shown in Eq. (11). The high-quality drift estimation has therefore presented another convincing case of how useful our original algorithm can be applied to dynamically monitoring dense sensor networks, as a by-product of calibrating low-cost nodes.

It should be noted that the mode of drift (linear or random drift) will not significantly affect our simulation results. As we demonstrated in Sect. 3.2.3, the performance of our algorithm is insensitive to the training data size. And we believe that models with a similar prediction accuracy should have a similar drift detection power. For example, if the prediction accuracy of the model trained on 59 days' data is virtually the same as the accuracy of the model trained on 2 days' data, and

if the model trained on 59 days is able to detect the simulated drift, then so should the model trained on 2 days. Then if we reasonably assume that the drift rate remains roughly unchanged within a 2-day window, then the drift mode (linear or random), which only dictates how the drift rate jumps (usually smoothly as well) between any adjacent discrete 2-day windows, does not matter anymore. All that matters is to track that one fixed drift rate reasonably well within those 2 days, which is virtually the same as what we already did with the entire 59 days' data.

**3.3.3 Optimal number of reference nodes**

Questions which remain unsolved are 1) what the optimum or minimum number of reference instruments is to sustain this technique and 2) if the inclusion of low-cost nodes can effectively assist in lowering the technique's calibration/mapping inaccuracy. It is interesting to note that optimizing the model's calibration accuracy can not only directly fulfill the

fundamental calibration task, but also help better the sensor network monitoring capability as an added bonus. To address these two outstanding issues, we randomly sampled subsets of all the 22 reference nodes within the network in increments of one node (i.e. from 1 to 21 nodes) and implemented our algorithm with and without incorporating the low-cost nodes, before finally computing the mean percent errors in predicting all the holdout reference nodes. To get the performance scores as close to truth as possible but without incurring excessive computational cost in the meantime, the sampling was repeated 100 times for each subset size. The calibration error in this section was defined as the mean percent errors in predicting all the holdout reference nodes further averaged over 100 simulation runs for each subset size.

Figure 10 describes the 24 h calibration percent error rate of the model as a function of the number of reference stations used for modelling with and without involving the low-cost nodes. The error rates generally decrease as the number of reference instruments increases (full network: from ~40 % with 1 node to ~29 % with 21 nodes; network excluding low-cost nodes: from ~43 % to ~30 %) but are somewhat locally variable and most pronounced when five, seven, and eight reference nodes are simulated. These bumps might simply be the result of five, seven, and eight reference nodes being relatively non-ideal (with regard to their neighboring numbers) for the technique, although the possibility of non-convergence due to the limited 100 simulation runs for each scenario cannot be ruled out. The 19 or 20 nodes emerge as the optimum numbers of reference nodes with the lowest errors of close to 28 %, while 17 to 21 nodes all yield comparably low inaccuracies (all below 30 %). The pattern discovered in our research shares certain similarities with Schneider et al. (2017) who studied the relationship between the accuracy of using colocation-calibrated low-cost nodes to map urban AQ and the number of simulated low-cost nodes for their urban-scale air pollution dispersion model and kriging-fueled data fusion technique in Oslo, Norway. Both studies indicate that at least roughly 20 nodes are essential to start producing acceptable degree of accuracy. Unlike Schneider et al. (2017) who further expanded the scope to 150 nodes by generating new synthetic stations from their established model and showed a "the more, the merrier" trend up to 50 stations, we restricted ourselves to only realistic data to investigate the relationship since we suspect that stations created from our model with approximately 30 % errors might introduce large noise which could misrepresent the true pattern. We agree with Schneider et al. (2017) that such relationships are location-specific and cannot be blindly transferred to other study sites.

At last, we used the Wilcoxon rank-sum test/Mann-Whitney U test again to prove that modelling with the 10 low-cost nodes can statistically significantly reduce the uncertainty of spatial interpolation of the reference node measurements in comparison to modelling without them, (at least) when the number of reference stations is optimum. We did not use a paired test here because the reference nodes for algorithm training for each simulation run were randomly chosen. Specifically in our study, for each number of reference stations, the two independent samples (100 replications per sample) are the 100 replications of the mean of the 24 h percent errors (in predicting all the holdout reference nodes) from the 100 repeated random simulations when modelling with and without the low-cost nodes, respectively. We conducted a one-sided test which has the null hypothesis that our model's mean 24 h prediction percent errors with and without including the low-cost nodes

are the same (i.e., $H_0$: with = without) against the alternative that the error with the low-cost nodes is smaller than the error without them (i.e., $H_1$: with < without). The p-values of the Wilcoxon rank-sum tests are superimposed on Fig. 10. The level of statistical significance was chosen to be 0.05, which means that the null hypothesis (i.e., $H_0$: with = without) can be rejected in favor of the alternative (i.e., $H_1$: with < without) when p-values are below 0.05. Figure 10 shows that the accuracy improvement when modelling with the 10 low-cost nodes is statistically significant when the optimum number of reference stations (i.e., 19 or 20) is used. Significant accuracy improvements were also observed for 17 and 18 reference stations that had comparably low prediction errors. Therefore, we conclude that when viewing the entire sensor network in Delhi as a whole system over the entire sampling period, modelling with the 10 low-cost nodes can decrease the extent of pure interpolation among only reference stations, (at least) when the number of reference stations is optimum. The accuracy gains are still relatively minor because of the suboptimal size of the low-cost node network (i.e., 10). We postulate that once the low-cost node network scales up to 100s, the model constructed using the full network information can be more accurate than the one with only the information of reference nodes by considerable margins.

## 4 Conclusions

This study introduced a simultaneous GPR and simple linear regression pipeline to calibrate wireless low-cost PM sensor networks (up to any scale) on the fly in the field by capitalizing on all available reference monitors across an area without the requirement of pre-deployment collocation calibration. We evaluated our method for Delhi, where 22 reference and 10 low-cost nodes were available from January 1, 2018 to March 31, 2018 (Delhi-wide average of the 3-month mean $PM_{2.5}$ among 22 reference stations: $138 \pm 31$ µg m$^{-3}$), using a leave-one-out CV over the 22 reference nodes. We demonstrated that our approach can achieve excellent robustness and reasonably high accuracy, as underscored by the low variability in the GPR model parameters and model-produced calibration factors for low-cost nodes and by an overall 30 % prediction error (equivalent to an RMSE of 33 µg m$^{-3}$) at a 24 h scale, respectively, among the 22-fold CV. We closely investigated into 1) the large model calibration errors (~50 %) at two low-cost sites (MRU and IITD with 3-month mean $PM_{2.5}$ of ~72 µg m$^{-3}$) where our E-BAMs were collocated; 2) the similarly large model prediction errors at the comparatively clean Pusa and Sector 62 reference sites; and 3) the washed-out local variability in the model calibrated low-cost sites. These observations revealed that the performance of our technique (and more generally the geostatistical techniques) can calibrate the low-cost nodes dynamically, but effective only if the degree of urban homogeneity in $PM_{2.5}$ is high. High urban homogeneity scenarios can be that the local contributions are as small a fraction of the regional ones as possible or the local contributions are prevalent but of similar magnitudes. Otherwise, quality predictions will only apply for those nodes whose means are close to the Delhi-wide mean. We showed that our algorithm performance is insensitive to training window size as the mean prediction error rate and the standard error of the mean (SEM) for the 22 reference stations remained consistent at ~30 % and ~3–4 % when an increment of 2 days' data were included in the model training. The markedly low requirement of our algorithm for training data enables the models to always be nearly most updated in the field, thus realizing the algorithm's

full potential for dynamically surveilling large-scale WLPMSNs by detecting malfunctioning low-cost nodes and tracking the drift with little latency. Our algorithm presented similarly stable 26–34 % mean prediction errors and ~3–7 % SEMs over the sampling period when pre-trained on the current week's data and predicting 1 week ahead, therefore suitable for dynamic calibration. Despite our algorithm's non-ideal calibration accuracy for Delhi, it holds the promise of being adapted for automated and streamlined large-scale wireless sensor network monitoring and of significantly reducing the amount of manual labor involved in the surveillance and maintenance. Simulations proved our algorithm's capability of differentiating malfunctioning low-cost nodes (due to either hardware failure or under heavy influence of local sources) within a network and of tracking the drift of low-cost nodes accurately with less than 4 % errors for all the simulation scenarios. Finally, our simulation results confirmed that the low-cost nodes are beneficial for the spatial precision of a sensor network by decreasing the extent of pure interpolation among only reference stations, highlighting the substantial significance of dense deployments of low-cost AQ devices for a new generation of AQ monitoring network.

Two directions are possible for our future work. The first one is to expand both the longitudinal and the cross-sectional scopes of field studies and examine how well our solution works for more extensive networks in a larger geographical area over longer periods of deployment (when sensors are expected to actually drift, degrade, or malfunction). This enables us to validate the practical use of our method for calibration and surveillance more confidently. The second is to explore the infusion of information about urban $PM_{2.5}$ spatial patterns such as high-spatial-resolution annual average concentration basemap from air pollution dispersion models (Schneider et al., 2017) into our current algorithm to further improve the on-the-fly calibration performance by correcting for the concentration range-specific biases.

**Data availability**

The data are available upon request to Tongshu Zheng (tongshu.zheng@duke.edu).

**Competing interests**

Author Ronak Sutaria is the founder of Respirer Living Sciences Pvt. Ltd, a startup based in Mumbai, India which is the developer of the Atmos low-cost AQ monitor. Ronak Sutaria was involved in developing and refining the hardware of Atmos and its server and dashboard and in deploying the sensors, but not in data analysis. Author Robert Caldow is the director of engineering at TSI and only responsible for the funding and technical support, but not for data analysis.

**Acknowledgments**

Sachchida N. Tripathi and Ronak Sutaria are supported under the Research Initiative for Real-time River Water and Air Quality Monitoring program funded by the Department of Science and Technology, Government of India and Intel

Corporation, and administered by the Indo-US Science and Technology Forum. The authors would like to thank CPCB, DPCC, IMD, SPCBs, and AirNow DOS (Department of State) for providing the Delhi 1 h reference PM$_{2.5}$ measurements used in the current study.

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

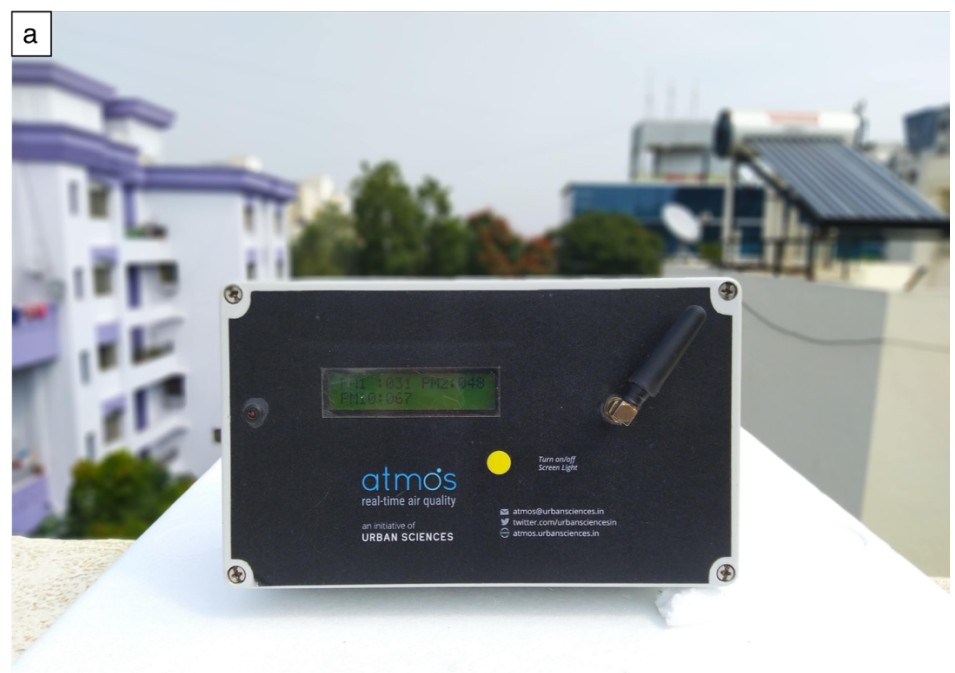

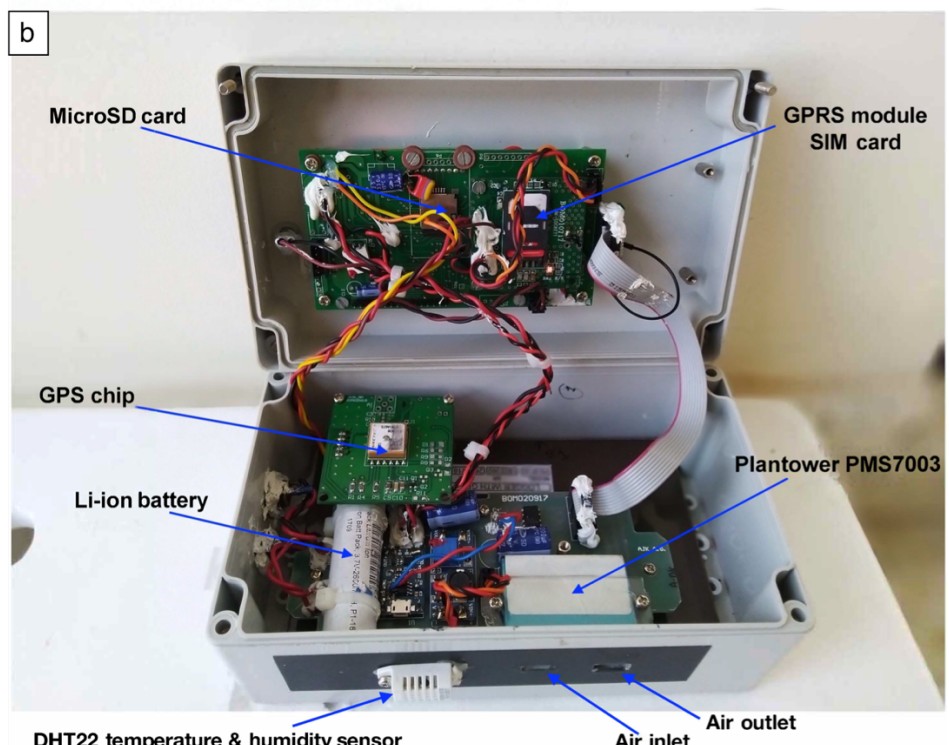

**Figure 1: (a) Front view of the low-cost node. (b) Key components of the low-cost node.**

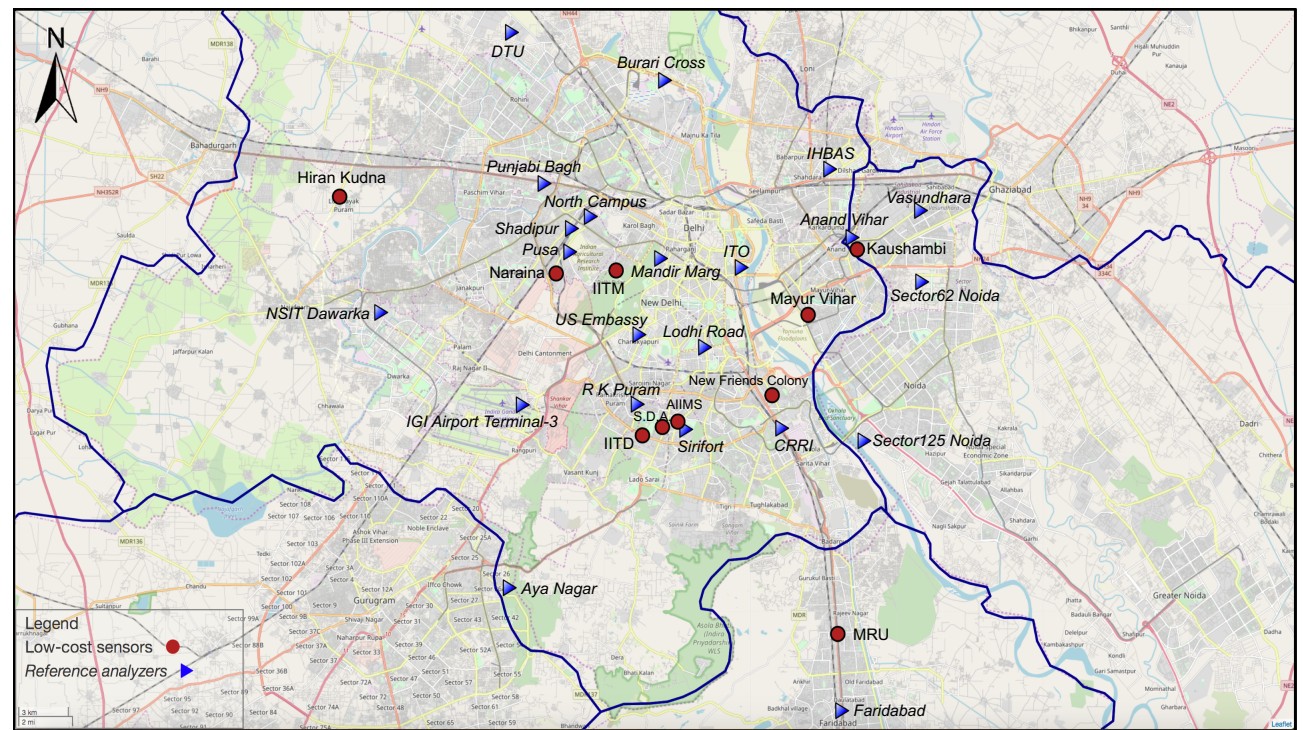

**Figure 2: Locations of the 22 reference nodes (triangle icons with *italic text*) and 10 low-cost nodes (circle icons) that form the Delhi PM sensor network.**

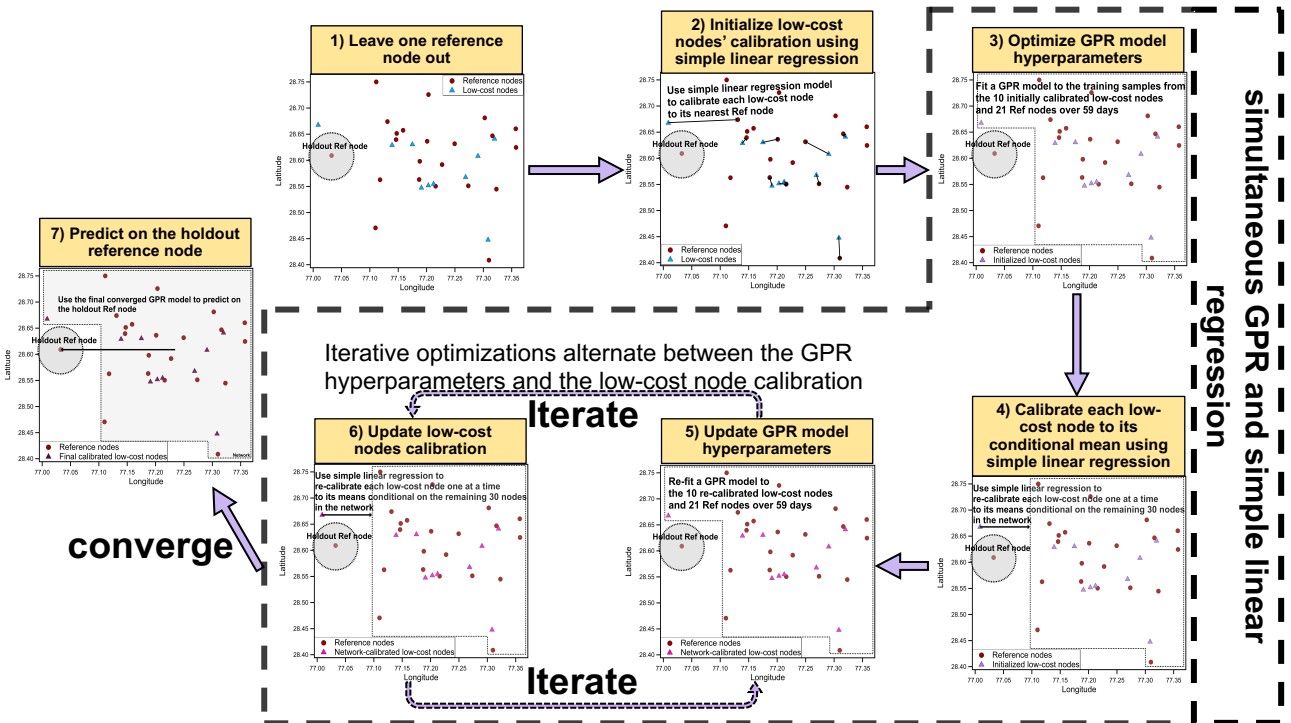

**Figure 3: The flow diagram illustrating the simultaneous GPR and simple linear regression calibration algorithm. In step one, for each of the 22-fold leave-one-out CVs, one of the 22 reference nodes is held out of modelling for the model predictive performance evaluation in step seven; in step two, fit a simple linear regression model between each low-cost node i and its closest reference node's PM$_{2.5}$, initialize low-cost node i's calibration model to this linear regression model, and calibrate the low-cost node i using this model; in step three, first initialize the GPR hyperparameters to [0.1, 50, 0.01] and then update/optimize the hyperparameters based on the training samples from the 10 initially calibrated low-cost nodes and 21 reference nodes over 59 days; in step four, first compute each low-cost node i's means conditional on the remaining 30 nodes given the optimized GPR hyperparameters, then fit a simple linear regression model between each low-cost node i and its conditional means, update low-cost node i's calibration model to this new linear regression model, and re-calibrate the low-cost node i using this new model; in step five and six, iterative optimizations alternate between the GPR hyperparameters and the low-cost node calibrations using the approaches described in step three and four, respectively, until the GPR hyperparameters converged; in step seven, predict the 59-day PM$_{2.5}$ measurements of the holdout reference node based on the finalized GPR hyperparameters and the low-cost node calibrations.**

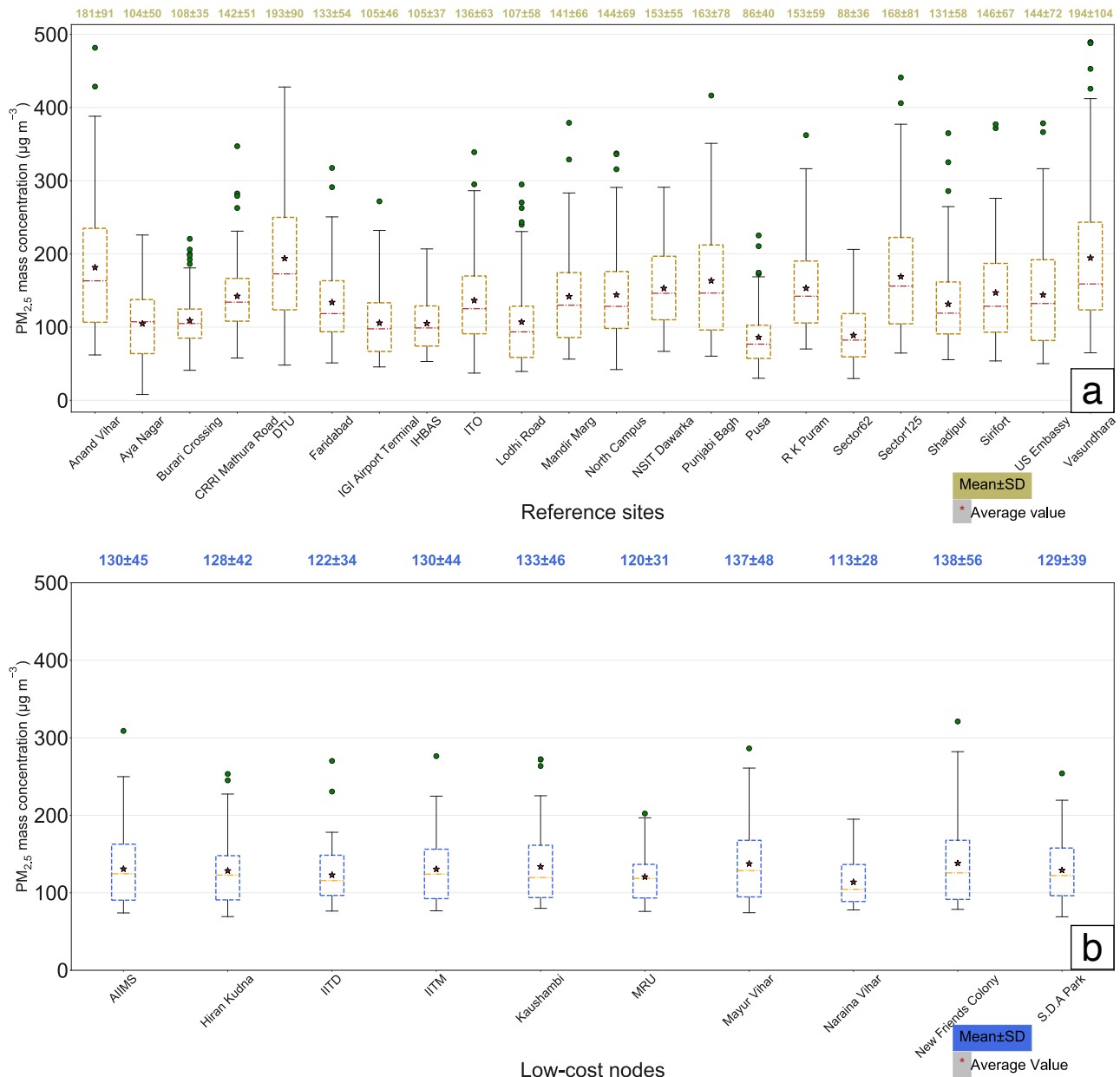

**Figure 4: a) Box plots of the 24 h aggregated true ambient PM$_{2.5}$ mass concentrations measured by the 22 government reference monitors across Delhi from January 1 to March 31, 2018. b) Box plots of the low-cost node 24 h aggregated PM$_{2.5}$ mass concentrations calibrated by the optimized GPR model. In both a) and b), mean and SD of the PM$_{2.5}$ mass concentrations for each individual site are superimposed on the box plots.**

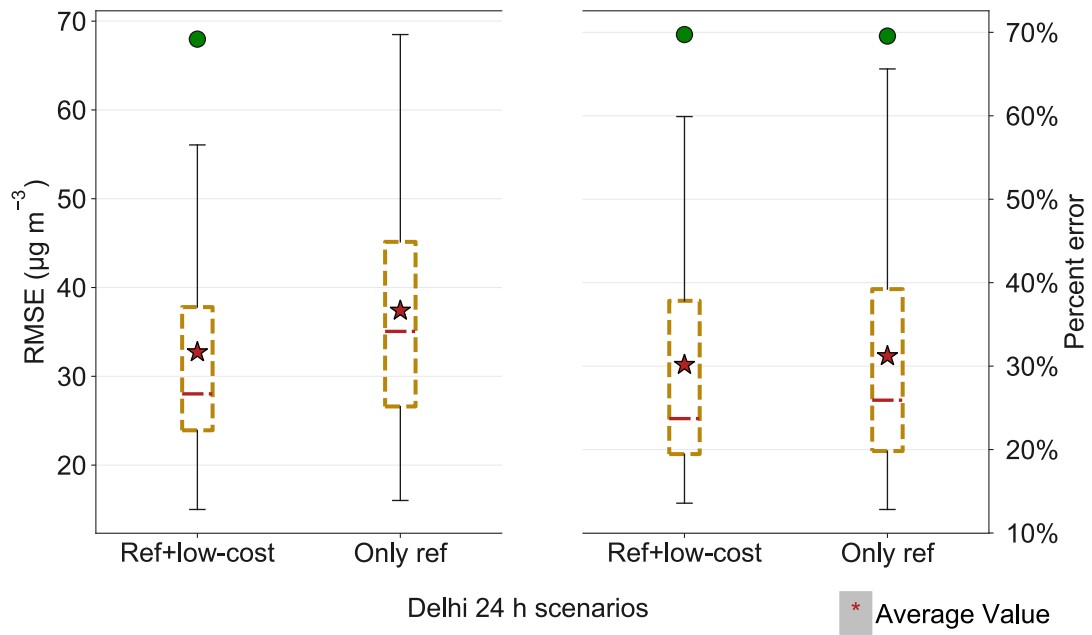

**Figure 5: Box plots of the GPR model 24 h performance scores (including RMSE and percent error) for predicting the measurements of the 22 holdout reference nodes across the 22-fold leave-one-out CV under two scenarios — using the full sensor network by including both reference and low-cost nodes and using only the reference nodes for the model construction. Note both scenarios were given the initial parameter values and bounds that maximize the model performance.**

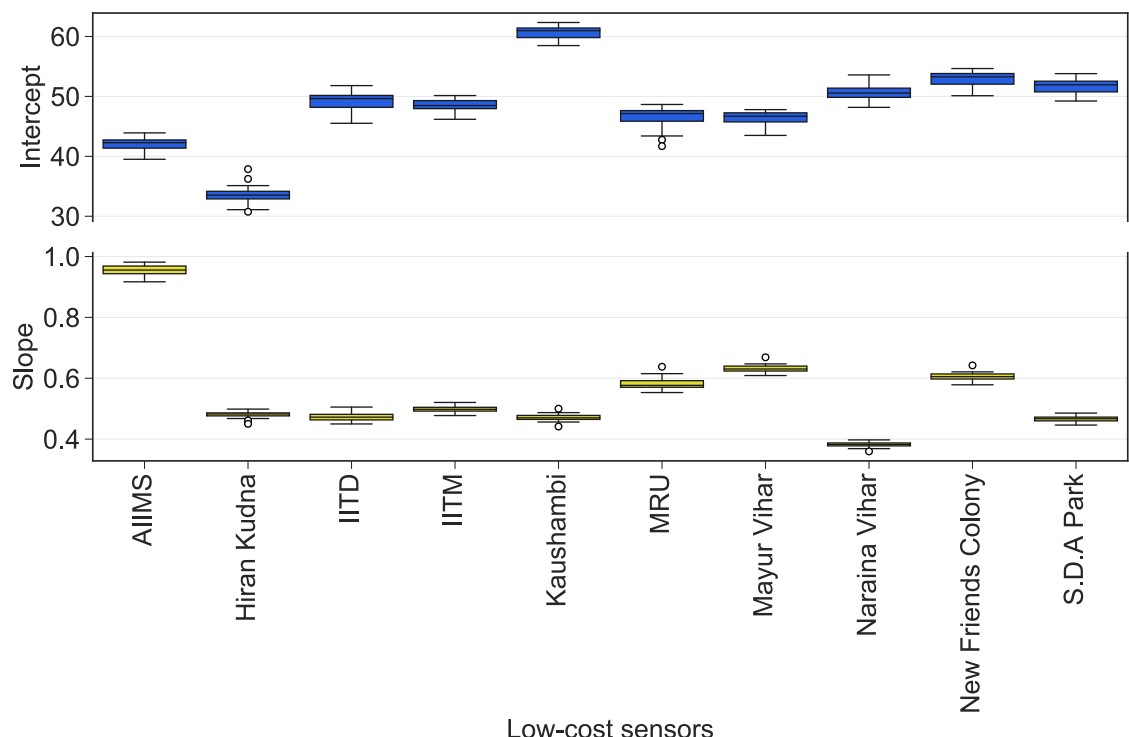

**Figure 6: Box plots of the learned calibration factors (i.e., intercept and slope) for each individual low-cost node from the 22 optimized GPR models across the 22-fold leave-one-out CV.**

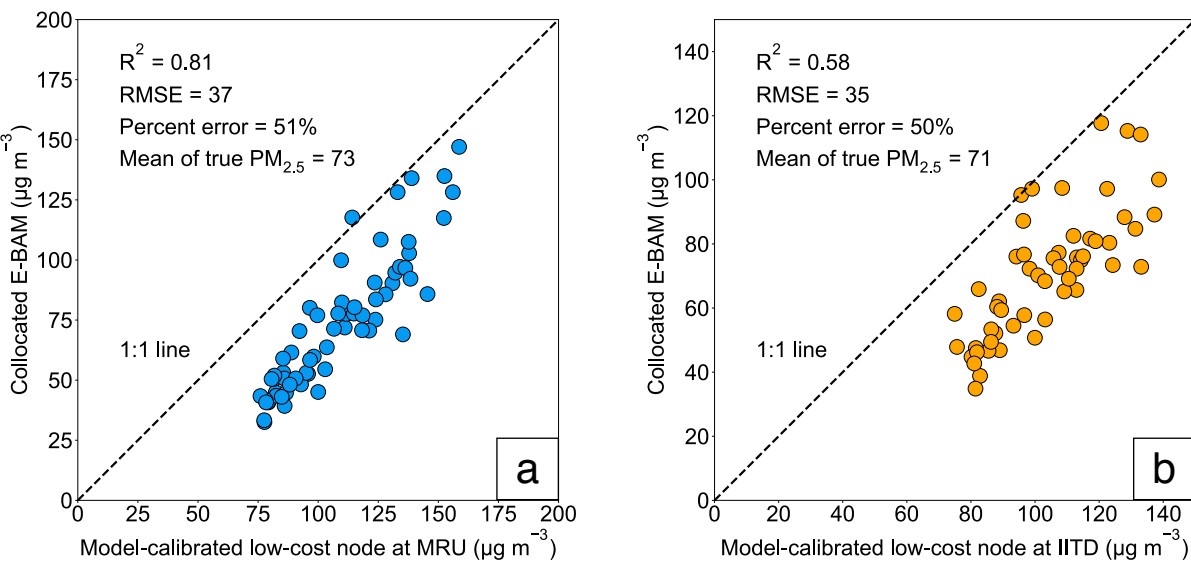

Figure 7: Correlation plots comparing the GPR model-calibrated low-cost node PM$_{2.5}$ mass concentrations to the collocated E-BAM measurements at a) MRU and b) IITD sites. In both a) and b), correlation of determination ($R^2$), RMSE, percent error, and mean of the true ambient PM$_{2.5}$ mass concentrations throughout the study (from January 1 to March 31, 2018) are superimposed on the correlation plots.

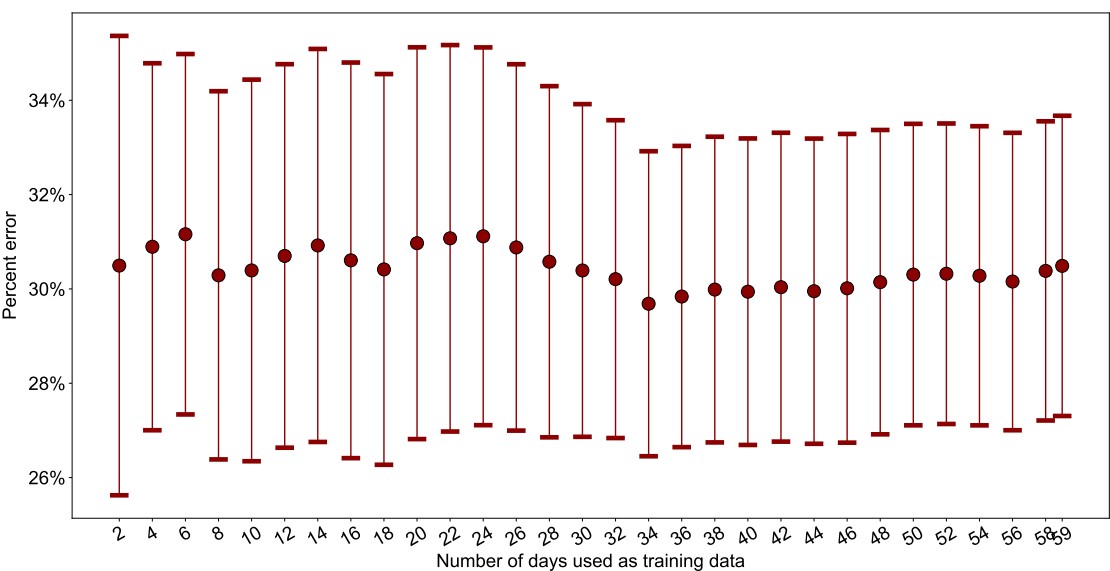

**Figure 8: The mean percent error rate of GPR model prediction on the 22 reference nodes using leave-one-out CV (see Sect. 3.2.1) as a function of training window size in an increment of 2 days. The error bars represent the standard error of the mean (SEM) of the GPR prediction errors of the 22 reference nodes.**

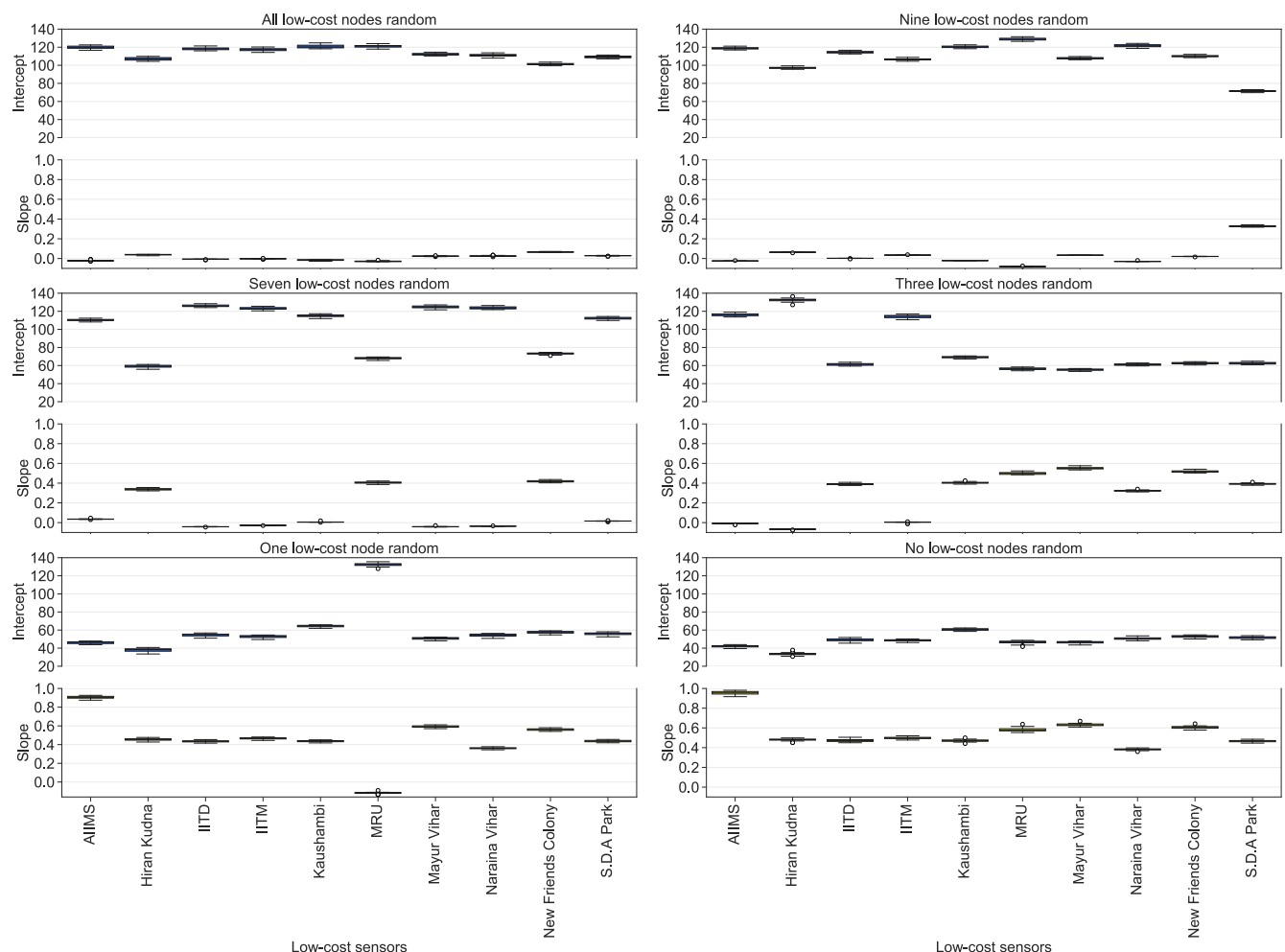

**Figure 9:** Learned calibration factors for each individual low-cost node from the optimized GPR models by replacing measurements of all (top left), nine (top right), seven (middle left), three (middle right), one (bottom left), and zero (bottom right) of the low-cost nodes with random integers bounded by the min and max of the true signals reported by the corresponding low-cost nodes. Note that the nine, seven, three, and one low-cost nodes (whose true signals are replaced with random integers) were randomly chosen.

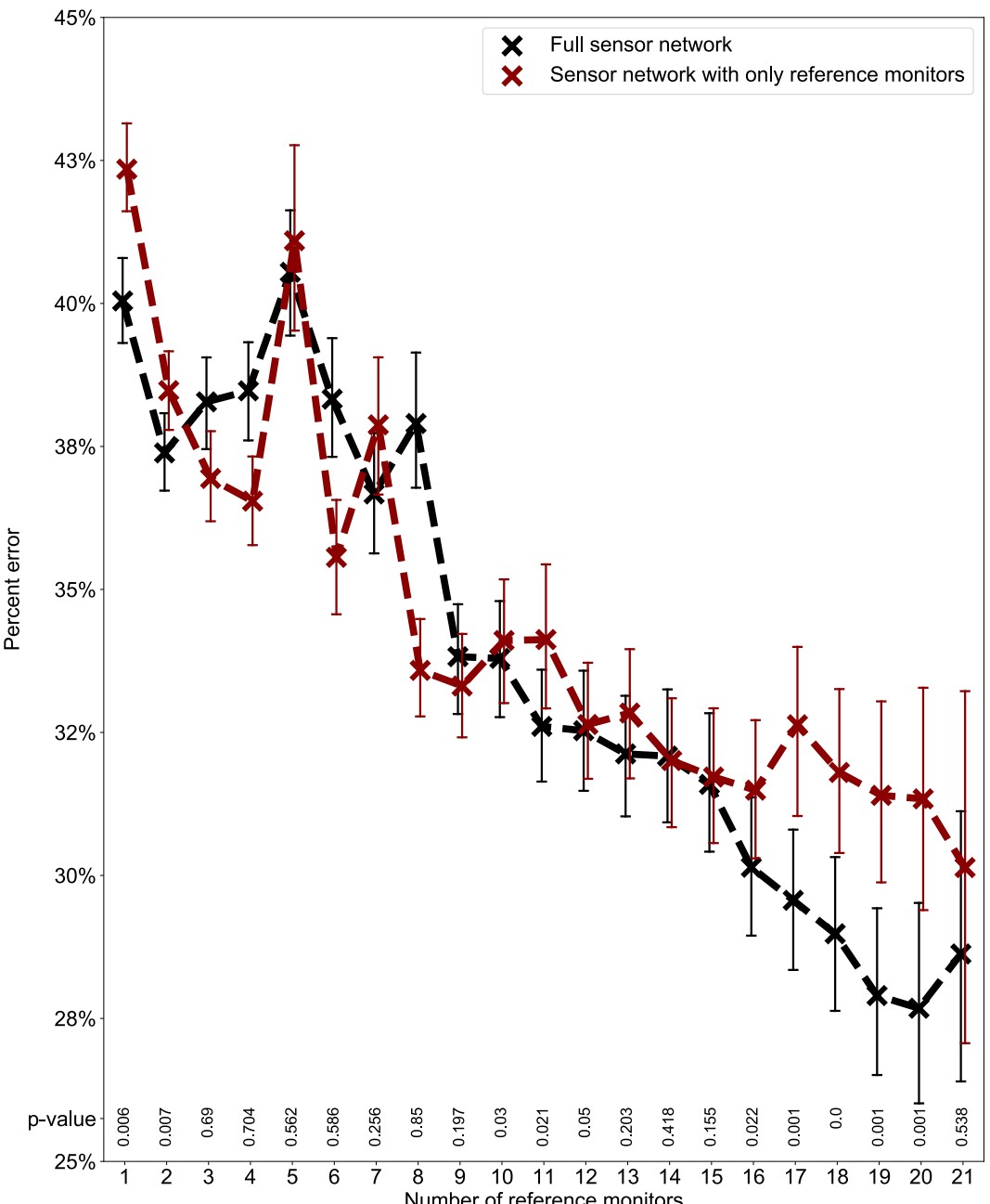

**Figure 10: Average 24 h percent errors of the GPR model for predicting the holdout reference nodes in the network as a function of the number of reference stations used for the model construction under two scenarios — using the full sensor network information by including both reference and low-cost nodes and using only the reference nodes for the model construction. Note each data point (mean value) is derived from 100 simulation runs. The error bars indicating 95 % CI of the means are based on 1000 bootstrap iterations. All scenarios were given the initial parameter values and bounds that maximize the model performance. The p-value of the Wilcoxon rank-sum test for each reference station number is superimposed, where p-value below 0.05 means that the error when modelling with the 10 low-cost nodes is smaller than the error without them for that reference station number.**

**Table 1: Delhi PM sensor network sites along with the 1 h percentage data completeness with respect to the entire sampling period (i.e., from January 1, 2018 00:00 to March 31, 2018 23:59, Indian Standard Time, IST; in total 90 days, 2160 hours) before and after 1 h missing-data imputation for each individual site. Note that a 10 % increase in the percentage data completeness after 1 h missing-data imputation is equivalent to ~216 hours of 1 h data being interpolated.**

| Category | Site names | Latitude | Longitude | Initial 1 h data completeness | 1 h data completeness after missing-data imputation |
|---|---|---|---|---|---|
| | Anand Vihar | N 28.6468350 | E 77.3160320 | 88 % | 100 % |
| | Aya Nagar | N 28.4706914 | E 77.1099364 | 97 % | 100 % |
| | Burari Cross | N 28.7258390 | E 77.2033350 | 98 % | 100 % |
| | CRRI Mathura Road | N 28.5512005 | E 77.2735737 | 98 % | 100 % |
| | Delhi Technological University (DTU) | N 28.7500499 | E 77.1112615 | 96 % | 100 % |
| | Faridabad | N 28.4088421 | E 77.3099081 | 98 % | 100 % |
| | IGI Airport Terminal-3 | N 28.5627763 | E 77.1180053 | 95 % | 100 % |
| | IHBAS, Dilshad Garden | N 28.6811736 | E 77.3025234 | 98 % | 100 % |
| | ITO Metro Station (ITO) | N 28.6316945 | E 77.2494387 | 98 % | 100 % |
| | Lodhi Road | N 28.5918245 | E 77.2273074 | 93 % | 100 % |
| Reference | Mandir Marg | N 28.6364290 | E 77.2010670 | 96 % | 100 % |
| | North Campus | N 28.6573814 | E 77.1585447 | 94 % | 100 % |
| | NSIT Dawarka | N 28.6090900 | E 77.0325413 | 95 % | 100 % |
| | Punjabi Bagh | N 28.6740450 | E 77.1310230 | 94 % | 100 % |
| | Pusa | N 28.6396450 | E 77.1462620 | 99 % | 100 % |
| | R K Puram | N 28.5632620 | E 77.1869370 | 95 % | 100 % |
| | Sector62 Noida | N 28.6245479 | E 77.3577104 | 93 % | 99 % |
| | Sector125 Noida | N 28.5447608 | E 77.3231257 | 90 % | 97 % |
| | Shadipur | N 28.6514781 | E 77.1473105 | 97 % | 100 % |
| | Sirifort | N 28.5504249 | E 77.2159377 | 78 % | 100 % |
| | US Embassy | N 28.5980970 | E 77.1880330 | 95 % | 100 % |
| | Vasundhara, Ghaziabad | N 28.6603346 | E 77.3572563 | 100 % | 100 % |
| | All India Institute of Medical Science (AIIMS) | N 28.5545006 | E 77.2124023 | 89 % | 100 % |
| | Hiran Kudna | N 28.6674995 | E 77.0089035 | 80 % | 97 % |
| | Indian Institute of Technology Delhi (IITD) | N 28.5473003 | E 77.1909027 | 88 % | 99 % |
| | Indian Institute of Tropical Meteorology (IITM) | N 28.6303400 | E 77.1750400 | 98 % | 100 % |
| | Kaushambi | N 28.6410008 | E 77.3199005 | 84 % | 100 % |
| Low-cost | Manav Rachna University (MRU) | N 28.4477005 | E 77.3084030 | 87 % | 100 % |
| | Mayur Vihar | N 28.6079998 | E 77.2906036 | 85 % | 93 % |
| | Naraina Vihar | N 28.6289005 | E 77.1391983 | 70 % | 79 % |
| | New Friends Colony | N 28.5676994 | E 77.2687988 | 99 % | 100 % |
| | S.D.A. Park | N 28.5517006 | E 77.2031021 | 66 % | 97 % |

**Table 2: Summary of the GPR model 24 h performance scores (including RMSE and percent error) for predicting the measurements of the 22 holdout reference nodes across the 22-fold leave-one-out CV when the full sensor network is used. The mean of the true ambient PM$_{2.5}$ mass concentrations throughout the study (from January 1 to March 31, 2018) for each individual reference node is provided. The reference nodes with the means of true PM$_{2.5}$ inside the range of [Delhi-wide mean ± SD, i.e., 138 ± 31] are indicated with shading.**

| Reference nodes | RMSE (µg m$^{-3}$) | Percent error | Mean of true PM$_{2.5}$ (µg m$^{-3}$) |
|---|---|---|---|
| Vasundhara, Ghaziabad | 68 | 44 % | 195 |
| DTU | 56 | 36 % | 194 |
| Anand Vihar | 47 | 32 % | 181 |
| Sector125 Noida | 31 | 23 % | 169 |
| Punjabi Bagh | 26 | 20 % | 163 |
| NSIT Dawrka | 25 | 19 % | 153 |
| R K Puram | 26 | 20 % | 153 |
| Sirifort | 22 | 18 % | 147 |
| US Embassy | 21 | 18 % | 144 |
| North Campus | 27 | 24 % | 144 |
| CRRI Mathura Road | 27 | 21 % | 142 |
| Mandir Marg | 16 | 14 % | 142 |
| ITO | 15 | 14 % | 136 |
| Faridabad | 21 | 18 % | 133 |
| Shadipur | 23 | 22 % | 132 |
| Burari Cross | 36 | 39 % | 109 |
| Lodhi Road | 34 | 41 % | 107 |
| IGI Airport Terminal–3 | 29 | 32 % | 106 |
| Aya Nagar | 34 | 38 % | 105 |
| IHBAS, Dilshad Garden | 38 | 41 % | 105 |
| Sector62 Noida | 47 | 60 % | 89 |
| Pusa | 48 | 70 % | 86 |
| Delhi-wide mean | 33 | 30 % | 138 |
| SD | 13 | 14 % | 31 |

**Table 3: Comparison of predetermined percentages of drift to those estimated from the GPR model for intercept and slope, respectively, for each individual low-cost node, assuming all (10), six, and two of the low-cost nodes developed various degrees of drift such as significant (11 %–99 %), marginal (1 %–10 %), and a balanced mixture of significant and marginal. Note the sensors that drifted, the percentages of drift, and which sensors drifted significantly or marginally are randomly chosen. The results reported under each scenario are based on averages of 10 simulation runs.**

| Drift category | Low-cost nodes | All low-cost nodes drift | | | | Six low-cost nodes drift | | | | Two low-cost nodes drift | | | |
|---|---|---|---|---|---|---|---|---|---|---|---|---|---|
| | | Intercept drift (%) | | Slope drift (%) | | Intercept drift (%) | | Slope drift (%) | | Intercept drift (%) | | Slope drift (%) | |
| | | True | Estimated | True | Estimated | True | Estimated | True | Estimated | True | Estimated | True | Estimated |
| Significant | AIIMS | 58 % | 57 % | 54 % | 54 % | 74 % | 71 % | 46 % | 47 % | 0 % | -1 % | 0 % | -1 % |
| | Hiran Kudna | 43 % | 30 % | 50 % | 52 % | 66 % | 61 % | 53 % | 53 % | 62 % | 64 % | 45 % | 44 % |
| | IITD | 51 % | 52 % | 52 % | 51 % | 0 % | -1 % | 0 % | -2 % | 0 % | 1 % | 0 % | -3 % |
| | IITM | 54 % | 53 % | 56 % | 55 % | 61 % | 58 % | 48 % | 48 % | 0 % | -1 % | 0 % | -2 % |
| | Kaushambi | 61 % | 62 % | 73 % | 72 % | 70 % | 70 % | 49 % | 48 % | 0 % | 0 % | 0 % | -2 % |
| | MRU | 55 % | 56 % | 56 % | 56 % | 58 % | 61 % | 41 % | 39 % | 0 % | -1 % | 0 % | -2 % |
| | Mayur Vihar | 60 % | 65 % | 48 % | 47 % | 0 % | 1 % | 0 % | -3 % | 0 % | 1 % | 0 % | -3 % |
| | Naraina Vihar | 56 % | 54 % | 76 % | 76 % | 0 % | -4 % | 0 % | 1 % | 0 % | -1 % | 0 % | -1 % |
| | New Friends Colony | 66 % | 68 % | 68 % | 67 % | 55 % | 55 % | 48 % | 47 % | 59 % | 61 % | 37 % | 36 % |
| | S.D.A. Park | 53 % | 47 % | 48 % | 50 % | 0 % | -4 % | 0 % | 2 % | 0 % | -1 % | 0 % | 0 % |
| | **Mean absolute difference** | **3 %** | | **1 %** | | **2 %** | | **1 %** | | **1 %** | | **2 %** | |
| 50 % significant and 50 % marginal | AIIMS | 4 % | 2 % | 5 % | 6 % | 0 % | -4 % | 0 % | 2 % | 0 % | 1 % | 0 % | -2 % |
| | Hiran Kudna | 51 % | 42 % | 51 % | 52 % | 50 % | 42 % | 50 % | 52 % | 0 % | 1 % | 0 % | -2 % |
| | IITD | 6 % | 4 % | 6 % | 6 % | 5 % | 2 % | 6 % | 8 % | 0 % | 0 % | 0 % | -2 % |
| | IITM | 56 % | 52 % | 40 % | 40 % | 64 % | 58 % | 47 % | 48 % | 0 % | 1 % | 0 % | -3 % |
| | Kaushambi | 60 % | 60 % | 42 % | 41 % | 5 % | 2 % | 5 % | 7 % | 0 % | 0 % | 0 % | -2 % |
| | MRU | 6 % | 5 % | 4 % | 3 % | 0 % | -6 % | 0 % | 3 % | 6 % | 3 % | 5 % | 5 % |
| | Mayur Vihar | 57 % | 59 % | 55 % | 55 % | 5 % | 2 % | 5 % | 6 % | 0 % | 1 % | 0 % | -2 % |
| | Naraina Vihar | 4 % | 0 % | 5 % | 7 % | 0 % | -4 % | 0 % | 2 % | 57 % | 65 % | 64 % | 63 % |
| | New Friends Colony | 6 % | 5 % | 6 % | 5 % | 0 % | -3 % | 0 % | 2 % | 0 % | -1 % | 0 % | -1 % |
| | S.D.A. Park | 53 % | 48 % | 61 % | 61 % | 59 % | 58 % | 64 % | 64 % | 0 % | 0 % | 0 % | -1 % |
| | **Mean absolute difference** | **3 %** | | **1 %** | | **4 %** | | **2 %** | | **2 %** | | **2 %** | |
| Marginal | AIIMS | 5 % | 5 % | 5 % | 4 % | 8 % | 8 % | 5 % | 5 % | 0 % | 0 % | 0 % | -1 % |
| | Hiran Kudna | 3 % | 4 % | 6 % | 5 % | 0 % | 0 % | 0 % | 0 % | 0 % | 0 % | 0 % | 0 % |
| | IITD | 5 % | 6 % | 7 % | 5 % | 7 % | 8 % | 5 % | 4 % | 6 % | 7 % | 5 % | 4 % |
| | IITM | 5 % | 5 % | 5 % | 5 % | 0 % | 0 % | 0 % | -1 % | 0 % | 0 % | 0 % | -1 % |
| | Kaushambi | 5 % | 5 % | 5 % | 4 % | 5 % | 6 % | 7 % | 6 % | 0 % | 0 % | 0 % | -1 % |
| | MRU | 5 % | 7 % | 4 % | 2 % | 6 % | 8 % | 5 % | 3 % | 5 % | 7 % | 6 % | 4 % |
| | Mayur Vihar | 7 % | 7 % | 5 % | 4 % | 0 % | 1 % | 0 % | -1 % | 0 % | 1 % | 0 % | -1 % |
| | Naraina Vihar | 6 % | 6 % | 7 % | 6 % | 7 % | 7 % | 6 % | 5 % | 0 % | 0 % | 0 % | -1 % |
| | New Friends Colony | 7 % | 8 % | 7 % | 5 % | 0 % | 1 % | 0 % | -2 % | 0 % | 1 % | 0 % | -1 % |
| | S.D.A. Park | 5 % | 5 % | 7 % | 6 % | 6 % | 6 % | 6 % | 6 % | 0 % | 0 % | 0 % | -1 % |
| | **Mean absolute difference** | **1 %** | | **1 %** | | **1 %** | | **1 %** | | **1 %** | | **1 %** | |