# Peer review of "Reasoning behind step four of the schema for the simultaneous GPR and simple linear regression calibration model"

_Atmospheric Measurement Techniques, 2019_

## Referee Comment (RC1) · Anonymous Referee #1 · 30 May 2019

General Comments:

Overall, the paper presents an interesting approach to in-field low-cost sensor calibration using Gaussian Process models. I have identified several points below which should be addressed before publication.

First, a better description of the algorithm should be supplied. The equations are not matched well with their descriptions in the text, and multiple steps of the process (e.g. the linear regression and Gaussian Process hyperparameter optimization) are described simultaneously. The diagrams of figure 3 are helpful, but not sufficient to clarify the entire process. A complete step-by-step breakdown of an example run of

[Figure]

the algorithm could be provided.

Second, it appears that both the Gaussian process hyperparameter calibration and the linear regression calibration of the low-cost nodes are carried out over an approximately 60-day period, using all data collected during this period. This would seem to preclude the use of your methods for on-line calibration. You may want to examine how this technique could be used in an on-line fashion, but designating a "current time" within the dataset and only using data collected prior to that time to calibrate the Gaussian Process hyperparameters and linear regression coefficients which are used to correct the data for that time. You could then also examine the effect of time history on your model, analyzing how the performance changes as more or less past data is included in the calibration process. As it currently is, if I am understanding your approach correctly, it can only be applied retroactively to a designated period of time for which all sensor data are available.

Third, when analyzing possible failure modes of sensors to determine if the algorithm can detect these modes, only two modes are considered: linear drift over time and replacement of the sensor signal with random noise. Other common failure modes should also be examined. These should include a "random walk" baseline drift (rather than simple linear drift), flatlining of the sensor (either at zero or at a non-zero value), and noisy corruption of a true signal (i.e. adding a random noise to the original signal, rather than completely replacing the true signal with random noise).

Finally, while the body of the paper presents a good discussion of the limitations of the proposed approach (mainly its need for spatial homogeneity in the true concentrations to be fully effective), this discussion is missing from the abstract. I believe that this observation is an important result of this paper and should be highlighted in the abstract as well.

Specific Comments:

Page 1, Lines 25-29: This is a very long and complex sentence; consider splitting in

into several sentences and/or revising how the information is presented. For example: "Simulations conducted using our algorithm suggest that in addition to dynamic calibration, it can also be adapted to automated monitoring of WLPMSNs. In these simulations, the algorithm was able to differentiate malfunctioning or singular low-cost nodes by identifying aberrant model-generated calibration factors (i.e. slopes close to zero and intercepts close to the global mean of true PM2.5). The algorithm was also able to track the drift of low-cost nodes accurately within 4% error for all the simulation scenarios."

Page 1, Line 27: I am not clear on what is meant by a "singular" node.

Page 2, Line 14: I assume you mean "since the emergence of low-cost AQ sensors" Rather than "since the emergence of calibration-related issues". It might be better to state that.

Page 3, Line 10: These coordinates are likely too precise to denote the city of Delhi generally. It is probably sufficient here to just state "Delhi, India", rather than providing coordinates, unless you are trying to describe a specific location within the city.

Page 3, Line 18: Rather than "drift nodes" I would say "the drift of nodes".

Page 4: Line 15: Use "the" rather than "our".

Page 5, Lines 6-7: It is not clear to me why the GPR model would require data from all stations to operate. If it is interpolating between stations then it should be able to fill in for any missing station data as well.

Page 5, Line 20: The meaning of "with that of after missing data imputation" is not clear.

Page 5, Line 21: I don't know if "imputed" is the correct word to use here.

Page 5, Line 26: Should be "while outliers have scores significantly larger than 1".

Page 6, Line 19 to Page 7, Line 5: This description could be improved. In particular,

it is not clear how the alpha and beta parameters of Equation 3 are determined. The description seems to combine a linear regression and a calibration of the hyperparameters of the Gaussian Process. These two steps should be described separately.

Page 6, Lines 23-24: The process of "standardization" is not clear to me. If this is done separately for each node, wouldn't this eliminate any systematic differences between measurement locations? If this step is only done to the data which are to be used for calibrating the model hyperparameters, then that should be stated. Even so, it is not clear that this is an appropriate step; for example, two node may be systematically higher than other locations, and so should have a mutual correlation, while if the means are subtracted, the data from the nodes would no longer be correlated (in other words, two variables can be made similar in a GP model either by giving them a high mutual correlation or by giving them a smaller prior variance and the same prior mean).

Equation 4: What is Gamma?

Page 7, line 7: What does the bold-face Theta denote? Are these the hyperparameters of the GP model as described in Equation 2?

Page 7, Lines 10-11: It is not clear what it means to re-calibrate a node based on its posterior mean. I am assuming this involves adjusting the alpha and beta parameters, but this is not clear.

Page 7, Line 29: What criteria are used for convergence?

Page 7, Line 31: This relates to a previous comment, I believe, but it should be described how the predictions are transformed back into the original PM scale.

Page 9, Lines 10-12: This sentence can be better written as "...the reference node mapping accuracy follows a pattern, with relatively high quality prediction for those nodes whose means are close to the global mean (e.g., global mean $\pm$ SD as highlighted with shading in Table 2) and relatively poor prediction for those nodes whose means differ substantially from the global mean (particularly on the lower end)".

Page 9, Line 21: It is unclear what the "scale of 10" refers to.

Page 12, Line 4: It is unclear what "quality drift estimation" is.

Page 12, Line 11: This should be "Questions which remain unsolved".

Page 13, Lines 24-27: The end of this sentence may be incomplete.

Page 13, Line 31: Again, it is not clear what is meant by a singular node.

---

## Referee Comment (RC2) · Anonymous Referee #2 · 7 Jun 2019

General Comments

This manuscript presents a methodology for performing a dynamical calibration for a network of low-cost sensors (LCS) in a polluted urban area. The work is based on taking 10 Plantower PMS7003 sensors in Delhi alongside 22 reference sites and using a combination of linear regression and kriging to attempt to calibrate the individual LCS without the need for any previous calibration. While this would be a huge boost to the field of LCS (requiring much less work to build out a network), the author's make it clear this method is not really a "calibration" method at all, but works better as a tool for the quality assurance and quality control of large networks. The main takeaway from

this manuscript (to me) was that with a network of 10 LCS, you can somewhat reliably (~30% error) predict the 24h average PM2.5 concentration in a major urban area. I don't intend for these comments to sound harsh – I think this is a very interesting and important result that others should know about – it is very hard to calibrate sensors in this way due to the spatial and temporal variation for certain pollutants (NO2, PM) while it may work for others that are more regional in nature (O3)...

Major and minor comments are outlined below.

Major Comments

1. I am unconvinced by the authors claims that they reduce the uncertainty in the spatial interpolation of reference networks – this needs a better description and error analysis to show the 2% improvement is statistically significant and/or applicable across different parts of the network in both space and time. 2. The authors interpolate data for both reference stations and LCS without any evidence this is a valid assumption. If this is going to stand, the authors should spend some time convincing the readers this is an appropriate methodology. The authors could quite easily show this by taking periods where the data are complete and comparing the linear interpolation results to the known concentrations. There is also no mention of the length of time covered by the temporal interpolations – are they just interpolating an hour? 12 hours? 3. The paper covers 24h averaged data – why not 1h data? Does the ability of the calibration model significantly decay? The authors note in the introduction that one of the key advantages of LCS is that they provide high temporal availability. From their results, we can only conclude the 30% accuracy applies to 24h measurements which are less resolved than most real-time reference monitors. 4. The author's base a large chunk of their model on the linear regression for calibrating LCS despite the fact most recent literature on the topic suggests this in invalid – relative humidity causes the growth of hygroscopic aerosols through water uptake which causes the linear regression approach to fall apart at humidities > ~50% depending on the kappa value of the aerosol. See work by Birmingham or CMU. This should be discussed at some point since linearity is a

major assumption in your model as I understand it. 5. Delhi has extremely complex air quality (as noted by the authors), but there was very little discussion on how these specific complexities contribute to the ability or inability of the model to perform. This is especially important for LCS as they don't suffer from random error, but error caused specifically by their inability to account for changes in aerosol composition and the underlying particle size distribution. There is a paper out by Gani et al (2019) that contains data in Delhi during this time period I think would be useful to this discussion.

Minor Comments (format is Page No., Line No.)

P. 1, L. 1: I don't think it's necessary to create unneeded acronyms – simply call them "sensor networks"

P. 2, L. 14: Are LCS really suffering from calibration issues? Or fundamental issues associated with using light scattering to determine the mass of particles?

P. 8, L. 13: The word "global" could be switched for something more specific. Maybe "Delhi-wide" or something more descriptive.

P. 10, L. 14: I disagree that IITD qualifies to be a background site when it is close to major roadways – see Gani et al (2019). They show a huge amount of local influence, especially during certain periods throughout the winter months.

---

## Referee Comment (RC3) · Anonymous Referee #3 · 25 Jun 2019

General Comments

This manuscript presents findings from a deployment of 10 low-cost PM monitors (based on the Plantower light-scattering sensor) in Delhi, a dense, polluted, urban environment – and methods used to calibrate the network leveraging 22 reference sites. The calibration method described relies on a blended approach using both kriging (Gaussian process regression) and linear regression. These types of novel approaches are of increasing importance given the emphasis on lower-cost sensing systems globally. These systems are often calibrated by collocation with reference monitors, which can be time-consuming and expensive. Findings from the study indicate some success with this new calibration method, though dealing with the perceived (and likely real) heterogeneity in emissions and sources across the vast, varied Delhi landscape proved challenging.

Major Comments 1/ Data are interpolated for both monitor types (LCS and reference). Why not perform analyses to validate your interpolation? For instance, by removing data of similar size to what is missing from non-missing periods and applying the same interpolation? How much consecutive data is interpolated? An hour here or there, or larger chunks of time?

2/ Speaking of interpolation and missingness: Are data missing in any specific pattern? That is, are areas that are typically reading higher levels of pollution more likely to have missing data? Do missing data occur most often on certain days (weekdays vs weekends)? Is missingness associated with ambient temperature, time of day, etc? Are certain monitors more prone to missingness?

3/ Relatedly, QAQC procedures for reference monitors are not described. While this data can be hard to obtain from the relevant Indian agencies, it is important to more strongly highlight this as a potential shortcoming or to find out more data on how and how often reference monitors are maintained and calibrated.

4/ Is any correction – of raw signal or for temperature and/or humidity – performed by the LCS platform? Are any filters applied at the LCS station or in the cloud? Describe more fully.

5/ Can you provide and compare data from the India Meteorology Department for average temp and RH across the period you performed measurements and for the 59 days of data you used? Are they statistically distinct?

Minor Comments P1, L15 – insert comma after "sites is questionable" P1, L19 – insert comma after Delhi P1, L20 – rephrase – perhaps "available for 59 days. . ." If you elect to keep the word "valid", describe what makes the data valid P2, L15 – add "with" between

[Figure]

"follow-up" and "routine" P3, L18 – whole sentence is very long, but specifically, for item 3), rephrase "auto-detect the faulty and auto-correct drift nodes" to (perhaps) "auto-detect faulty and auto-correct nodes with drift" P4, L15 – replace "our" with "the" P5, L12-13 – describe the API, or remove mention of it P5, L16 – add " 's " to location P6 – describe more fully the "standardization" that occurs P8, L14 – rephrase "Spatially, the global average..." – is this the average across all LCS and reference monitors? Or? And if it is the average across all, then does "spatially" apply? P9, L3 – insert comma after "decent"; consider rephrasing (what does decent mean in this context?) P9, L8-13 – While I understand that GPR would have done better absent local sources, is that realistic for these types of urban environments in places like India or China? Or even in the US, in places like Queens, Oakland, or Atlanta? Isn't the spatial heterogeneity exactly why many are considering more spatially and temporally resolved monitoring networks? P10, L14 – Neither of these sites are really background sites in the way they are traditionally thought of. P12, L10 – Perhaps rephrase to "The following questions remain:" or somesuch

Figure 2 – consider different shapes and colors (in B&W, the colors are not distinguishable) Figure 3 – a more elaborate caption may help better explain the flow (for instance, a sentence for each step)

---

## Author Comment (AC1) · 9 Jul 2019

**Response to Comments from Reviewer #1 AMT-2019-55**

The authors would like to sincerely thank the reviewer #1 for the careful review of the manuscript, the quick feedback, and the very constructive comments which helped dramatically improve the manuscript. The reviewer's comments are in italics, the summaries of our responses are in plain font, and the changes in the manuscript are in **bold red text**. Page and line
5   numbers refer to the original document. We also appended a marked-up manuscript version to the end of the responses to better show all the changes made from this review.

**Reviewer #1**

**General Comments:**

*First, a better description of the algorithm should be supplied. The equations are not matched well with their descriptions in*
10  *the text, and multiple steps of the process (e.g. the linear regression and Gaussian Process hyperparameter optimization) are described simultaneously. The diagrams of figure 3 are helpful, but not sufficient to clarify the entire process. A complete step-by-step breakdown of an example run of the algorithm could be provided.*
**Response:** We agree with the reviewer on that the algorithm should be better described in Section 2.3. Based on the reviewer's specific comments, we have now provided more details about how the alpha and beta parameters of Equation 3
15  were determined, how the standardization process was implemented, what it meant to re-calibrate a low-cost node based on its conditional mean, what criteria were used for convergence, and how the predictions were transformed back to the original PM scale. Regarding **1) "The equations are not matched well with their descriptions in the text",** we identified problems such as lack of the description of the $y_i$ term in Equation 3, lack of the description of the $\Gamma$ term in Equation 4, discrepancy in the Theta notation between Equations 2 and 5, lack of the description of the Theta term in Equations 2 and 5. We have
20  now corrected these issues. Regarding **2) "multiple steps of the process (e.g. the linear regression and Gaussian Process hyperparameter optimization) are described simultaneously"**, we presume that this comment is connected to the reviewer's specific comment on Page 6, Line 19 to Page 7, Line 5. The linear regression step (called low-cost node initialization, corresponding to step 2 in Fig. 3, described on page 6, lines 19-23) and the training/optimization of the hyperparameters of the GPR model (corresponding to step 3 in Fig. 3, described on page 6, lines 23-29, starting from 'After
25  standardizing') were previously described separately. To better highlight this fact and in order to avoid confusion, we have now added additional details to the low-cost node initialization step, have split the descriptions of the two steps into two separate paragraphs, have re-organized the places of Equations 3-5, and have added additional texts to explain the terms in Equations 3-5. Additionally, we have now placed each critical step under a sub-section (e.g., Sect. 2.3.x) to facilitate reading.

Regarding **3) "The diagrams of figure 3 are helpful, but not sufficient to clarify the entire process",** we have now revised Figure 3 to make it more informative about and more accurately reflect the entire process and we have now expanded the Figure 3 caption to help better carry readers through the algorithm. Regarding 4) **"A complete step-by-step breakdown of an example run of the algorithm could be provided",** we have now added a detailed algorithm block along with a sub-section number next to each critical step to indicate under which sub-section the details of that step can be found. The Section 2.3 has been completely overhauled.

Modified Section 2.3:

[revised manuscript text omitted]

*Second, it appears that both the Gaussian process hyperparameter calibration and the linear regression calibration of the low-cost nodes are carried out over an approximately 60-day period, using all data collected during this period. This would seem to preclude the use of your methods for on-line calibration. You may want to examine how this technique could be used in an on-line fashion, but designating a "current time" within the dataset and only using data collected prior to that time to calibrate the Gaussian Process hyperparameters and linear regression coefficients which are used to correct the data for that time. You could then also examine the effect of time history on your model, analyzing how the performance changes as more or less past data is included in the calibration process. As it currently is, if I am understanding your approach correctly, it can only be applied retroactively to a designated period of time for which all sensor data are available.*

**Response:** We really appreciate the reviewer's insightful suggestions and have examined both the possibility of using our method for online calibration and the effect of time history on our model. We will answer the reviewer's second question first and then circle back to the first question. Regarding **the effect of time history on our model**, we analyzed how the model performance changed when an increment of 2 days' data were included in the model. The model performance was based on the accuracy of model prediction on the 22 reference nodes (within the time periods of the data included) using leave-one-out CV, as described in Sect. 3.2.1. We observed a surprisingly consistent ~30 % error rate and ~3–4 % standard error of the mean (SEM) regardless of how many 2-day increments were used as the training window size. The small effect of training window size on the model performance hints **that using our method for online calibration/prediction is feasible**. We assessed the performance of using simple linear regression calibration factors and GPR hyperparameters that were optimized from one week to calibrate the 10 low-cost nodes and predict each of the 22 reference nodes in the next week. For example, the first/second/third/… week data were used as training data to build GPR models and simple linear regression models. These simple linear regression models were then used to calibrate the low-cost nodes in the second/third/fourth/… week, followed by GPR models to predict each of the 22 reference nodes in that week. The performance was still measured by the accuracy of model prediction on the 22 reference nodes using leave-one-out CV, as described in Sect. 3.2.1. We found similarly stable 26–34 % online calibration error rates and ~3–7 % SEMs throughout the weeks. We have now added two sub-sections (i.e., **Sect. 3.2.3 GPR model performance as a function of training window size** and **Sect. 3.2.4 GPR model dynamic calibration performance**, respectively) to address the time history and online calibration questions, respectively. A figure showing the model performance as a function of training window size for Sect. 3.2.3 was added to the main manuscript as Figure 8. Another figure showing the GPR model dynamic calibration performance for each successive week (from weeks 2 to 8) was added to the supplement as Figure S4. We have also updated the abstract, introduction, and conclusions to include the new results. Finally, all the figure numbers have been changed accordingly.

Added Section 3.2.3:

**3.2.3 GPR model performance as a function of training window size**

**So far, the optimization of both GPR model hyperparameters and the linear regression calibration factors for the low-cost nodes has been carried out over the entire sampling period using all 59 valid daily-averaged data points. It is of critical importance to examine the effect of time history on the algorithm, by analyzing how sensitive the model performance is to training window size. We tracked the model performance change when an increment of 2 days' data were included in the model training. The model performance was measured by the mean accuracy of model prediction on the 22 reference nodes (within the time period of the training window) using leave-one-out CV, as described in Sect. 3.2.1. Figure 8 illustrates that, throughout the 59 days, the error rate and the standard error of the mean (SEM) remained surprisingly consistent at ~30 % and ~3–4 %, respectively, regardless of how many 2-day**

**increments were used as the training window size. The little influence of training window size on the GPR model performance is possibly a positive side effect of the algorithm's time-invariant mean assumption, strong spatial smoothing effect, and the additional averaging of the error rates of the 22 reference nodes. The markedly low requirement of our algorithm for training data is powerful in that it enables the GPR model hyperparameters and**

5    **the linear regression calibration factors to always be nearly most updated in the field. This helps realize the algorithm's full potential for automatically surveilling large-scale networks by detecting malfunctioning low-cost nodes within a network (see Sect. 3.3.1) and tracking the drift of low-cost nodes (see Sect. 3.3.2) with as little latency as possible.**

10    Added Figure 8:

[Figure]

**Figure 8: The mean percent error rate of GPR model prediction on the 22 reference nodes using leave-one-out CV (see Sect. 3.2.1) as a function of training window size in an increment of 2 days. The error bars represent the standard error of the mean (SEM) of the GPR prediction errors of the 22 reference nodes.**

Added Section 3.2.4:

**3.2.4 GPR model dynamic calibration performance**

**The stationary model performance in response to the increase of training data hints that using our method for dynamic calibration/prediction is feasible. We assessed the algorithm's 1 week-ahead prediction performance, by**

using simple linear regression calibration factors and GPR hyperparameters that were optimized from one week to calibrate the 10 low-cost nodes and predict each of the 22 reference nodes, respectively, in the next week. For example, the first/second/third/… week data were used as training data to build GPR models and simple linear regression models. These simple linear regression models were then used to calibrate the low-cost nodes in the second/third/fourth/… week, followed by the GPR models to predict each of the 22 reference nodes in that week. The performance was still measured by the mean accuracy of model prediction on the 22 reference nodes using leave-one-out CV, as described in Sect. 3.2.1. We found similarly stable 26–34 % dynamic calibration error rates and ~3–7 % SEMs throughout the weeks (see Figure S4).

Added Figure S4:

[Figure]

Figure S4: The 1 week-ahead prediction error of the GPR models (which were pre-trained on the current week's data) as a function of the week being predicted. The error bars represent the standard error of the mean (SEM) of the GPR prediction errors of the 22 reference nodes.

Added text for the Abstract on Page 1, line 25:

We further demonstrated that our algorithm performance is insensitive to training window size as the mean prediction error rate and the standard error of the mean (SEM) for the 22 reference stations remained consistent at ~30 % and ~3–4 % when an increment of 2 days' data were included in the model training. The markedly low

**requirement of our algorithm for training data enables the models to always be nearly most updated in the field, thus realizing the algorithm's full potential for dynamically surveilling large-scale WLPMSNs by detecting malfunctioning low-cost nodes and tracking the drift with little latency. Our algorithm presented similarly stable 26–34 % mean prediction errors and ~3–7 % SEMs over the sampling period when pre-trained on the current week's data and predicting 1 week ahead, therefore suitable for online calibration.**

**3) examining the sensitivity of our algorithm to the training data size and the feasibility of it for dynamic calibration;**

**We showed that our algorithm performance is insensitive to training window size as the mean prediction error rate and the standard error of the mean (SEM) for the 22 reference stations remained consistent at ~30 % and ~3–4 % when an increment of 2 days' data were included in the model training. The markedly low requirement of our algorithm for training data enables the models to always be nearly most updated in the field, thus realizing the algorithm's full potential for dynamically surveilling large-scale WLPMSNs by detecting malfunctioning low-cost nodes and tracking the drift with little latency. Our algorithm presented similarly stable 26–34 % mean prediction errors and ~3–7 % SEMs over the sampling period when pre-trained on the current week's data and predicting 1 week ahead, therefore suitable for dynamic calibration.**

*Third, when analyzing possible failure modes of sensors to determine if the algorithm can detect these modes, only two modes are considered: linear drift over time and replacement of the sensor signal with random noise. Other common failure modes should also be examined. These should include a "random walk" baseline drift (rather than simple linear drift), flatlining of the sensor (either at zero or at a non-zero value), and noisy corruption of a true signal (i.e. adding a random noise to the original signal, rather than completely replacing the true signal with random noise).*

**Response:** We thank the reviewer for his/her expertise in the possible failure modes of sensors and his/her scientific rigor. As we demonstrated in our response to the reviewer's second general comment, the performance of our algorithm is insensitive to the training data size. And we believe that models with a similar prediction accuracy should have a similar failure mode detection power. For example, if the prediction accuracy of the model trained on 59 days' data is virtually the same as the accuracy of the model trained on 2 days' data (as shown previously), and if the model trained on 59 days is able to detect the simulated drift, then so should the model trained on 2 days. Then if we reasonably assume that the drift rate remains roughly unchanged within a 2-day window (as drift is believed to occur on time periods much longer than 2 days), then the drift mode (linear or random), which only dictates how the drift rate jumps (usually smoothly as well) between any adjacent discrete 2-day windows, does not matter that much anymore. All that matters is to track that one fixed drift rate reasonably well within those 2 days, which is virtually the same as what we already did and demonstrated with the entire 59

days' data in Sect. 3.3.2. Therefore, we do not believe that **the mode of drift** is a major issue. Regarding **flatlining**, we thank the reviewer for mentioning and defining this type of failure mode. Flatlining is in fact the most commonly seen failure mode of our PM sensors in Delhi. The raw signals of such malfunctioning PM sensors were observed to flatline at the upper end of the sensor output values (typically thousands of µg m$^{-3}$). The very distinct signals of these flatlining low-cost

5   PM nodes make it rather easy to separate them from the rest of the nodes and filter them out at the early pre-processing stage before analyses without having to resort to our algorithm. Regarding **noisy corruption of a true signal**, this particular failure mode is commonly seen in low-cost electrochemical sensors (such as ozone and nitrogen dioxide) based on redox reactions, but rarely seen in low-cost PM sensors that employ a light-scattering approach. Therefore, we consider the noisy corruption failure mode not applicable to or out of the scope of our current paper whose main subject is low-cost PM sensors.

10   We have now added the discussion about why we do not need our algorithm to detect flatlining and why the mode of drift will not affect our simulation results to Sect. 3.3.1 and Sect. 3.3.2, respectively.

Added text on Page 11, line 8:

"**It is worth mentioning that flatlining is another commonly seen failure mode of our low-cost PM sensors in Delhi. The raw signals of such malfunctioning PM sensors were observed to flatline at the upper end of the sensor output**

15   **values (typically thousands of µg m$^{-3}$). The very distinct signals of these flatlining low-cost PM nodes, however, make it rather easy to separate them from the rest of the nodes and filter them out at the early pre-processing stage before analyses, therefore without having to resort to our algorithm.**"

Modified text on Page 12, lines 6-9:

20   "~~We can rebuild a model such as every week using a rolling window (to keep the number of observations for model construction roughly unchanged) to assess the drifts in the model space over time. After that, the true calibration factors obtained from the initial collocation with reference instruments prior to deployment can be adjusted accordingly based on the model-estimated drifts. This procedure allows for real-time drift corrections to low-cost node measurements.~~**

**It should be noted that the mode of drift (linear or random drift) will not significantly affect our simulation results.**

25   **As we demonstrated in Sect. 3.2.3, the performance of our algorithm is insensitive to the training data size. And we believe that models with a similar prediction accuracy should have a similar drift detection power. For example, if the prediction accuracy of the model trained on 59 days' data is virtually the same as the accuracy of the model trained on 2 days' data, and if the model trained on 59 days is able to detect the simulated drift, then so should the model trained on 2 days. Then if we reasonably assume that the drift rate remains roughly unchanged within a 2-day**

30   **window, then the drift mode (linear or random), which only dictates how the drift rate jumps (usually smoothly as well) between any adjacent discrete 2-day windows, does not matter anymore. All that matters is to track that one fixed drift rate reasonably well within those 2 days, which is virtually the same as what we already did with the entire 59 days' data.**"

*Finally, while the body of the paper presents a good discussion of the limitations of the proposed approach (mainly its need for spatial homogeneity in the true concentrations to be fully effective), this discussion is missing from the abstract. I believe that this observation is an important result of this paper and should be highlighted in the abstract as well.*

**Response:** We did attempt to convey this message on Page 1, lines 23-25; however, the message might be somewhat subtle. We have now more prominently discussed the limitations of the proposed approach.

Modified text on Page 1, lines 23-25:

"**Of the 22 reference stations, high-quality predictions were observed for those stations whose PM$_{2.5}$ means were close to the Delhi-wide mean (i.e., 138 ± 31 μg m$^{-3}$) and relatively poor predictions for those nodes whose means differed substantially from the Delhi-wide mean (particularly on the lower end). We also observed washed-out local variability in PM$_{2.5}$ across the 10 low-cost sites** after calibration **using our approach**, **which stands in marked contrast to the true wide variability across the reference sites. These observations revealed that our proposed technique (and more generally the geostatistical technique) requires high spatial homogeneity in the pollutant concentrations to be fully effective.**"

**Specific Comments:**

*Page 1, Lines 25-29: This is a very long and complex sentence; consider splitting in into several sentences and/or revising how the information is presented. For exam- ple: "Simulations conducted using our algorithm suggest that in addition to dynamic calibration, it can also be adapted to automated monitoring of WLPMSNs. In these simulations, the algorithm was able to differentiate malfunctioning or singular low-cost nodes by identifying aberrant model-generated calibration factors (i.e. slopes close to zero and intercepts close to the global mean of true PM2.5). The algorithm was also able to track the drift of low-cost nodes accurately within 4% error for all the simulation scenarios."*

**Response:** Thank you, we have made the suggested revision to this sentence.

Modified text on Page 1, lines 25-29:

"**Simulations conducted using our algorithm suggest that in addition to dynamic calibration, the algorithm can also be adapted for automated monitoring of large-scale WLPMSNs. In these simulations, the algorithm was able to differentiate malfunctioning low-cost nodes (due to either hardware failure or under heavy influence of local sources) within a network by identifying aberrant model-generated calibration factors (i.e., slopes close to zero and intercepts close to the Delhi-wide mean of true PM$_{2.5}$). The algorithm was also able to track the drift of low-cost nodes accurately within 4 % error for all the simulation scenarios.**"

*Page 1, Line 27: I am not clear on what is meant by a "singular" node.*

**Response:** "Singular" node means an anomalous/abnormal node that reports signals that are spatially uncorrelated with other normal nodes within the network, due to under heavy influence of local sources. This echoes Section 3.3.1. The word

"singular" was meant to differentiate the situation from sensor failure ("malfunctioning"). Given the confusion this word gives, we have removed "singular" throughout the manuscript. Instead, we have clarified that "malfunctioning" corresponds to two situations throughout the manuscript (i.e., sensor hardware failure and sensors under heavy influence of local sources).

Modified text on Page 1, lines 26-28:

"In these simulations, the algorithm was able to differentiate **malfunctioning low-cost nodes (due to either hardware failure or under heavy influence of local sources)** within a network by identifying aberrant model-generated calibration factors (i.e., slopes close to zero and intercepts close to the global mean of true $PM_{2.5}$)."

Modified text on Page 11, lines 2-4:

"These two observations indicate that the GPR model enables automated and streamlined process of instantly spotting any malfunctioning low-cost nodes **(due to either hardware failure or under heavy influence of local sources)** within a large-scale sensor network."

Modified text on Page 13, lines 30-32:

"Simulations proved our algorithm's capability of differentiating malfunctioning low-cost nodes **(due to either hardware failure or under heavy influence of local sources)** within a network and of tracking the drift of low-cost nodes accurately with less than 4 % errors for all the simulation scenarios."

*Page 2, Line 14: I assume you mean "since the emergence of low-cost AQ sensors" Rather than "since the emergence of calibration-related issues". It might be better to state that.*

**Response:** Thank you for pointing this out, we have revised the sentence accordingly.

Modified text on Page 2, line 14:

"On the down side, researchers have been plagued by calibration-related issues since **the emergence of low-cost AQ sensors.**"

*Page 3, Line 10: These coordinates are likely too precise to denote the city of Delhi generally. It is probably sufficient here to just state "Delhi, India", rather than providing coordinates, unless you are trying to describe a specific location within the city.*

**Response:** Thank you, we have removed the coordinates.

Modified text on Page 3, line 10:

"…collocation calibration by leveraging all available reference monitors across an area (e.g., **Delhi, India**)."

*Page 3, Line 18: Rather than "drift nodes" I would say "the drift of nodes".*

**Response:** Thank you, we have made the suggested change to wording.

Modified text on Page 3, line 18:

"…auto-detect the faulty and auto-correct **the drift of nodes** within a network via computational simulation, …"

5    *Page 4: Line 15: Use "the" rather than "our".*

**Response:** Thank you, we have made the suggested change

Modified text on Page 4, line 15:

"…KairosDB as **the** primary fast scalable time series database built on Apache Cassandra, …"

10    *Page 5, Lines 6-7: It is not clear to me why the GPR model would require data from all stations to operate. If it is interpolating between stations then it should be able to fill in for any missing station data as well.*

**Response:** The reviewer is correct. Mathematically, the GPR model would only require data from at least one reference monitoring station to operate. In this paper, the choice of attempting to interpolate all the stations' missing data first was made based on some practical reasons, specifically the speed of the algorithm/program. Theoretically, relying on GPR model

15    to fill in for any missing station data is 59 (the number of daily-averaged data points) times slower. This theoretical upper bound is 59 because the algorithm will have to loop through each of the 59 days if each day's missing reference and low-cost nodes are different. And this process is relatively computationally expensive because it involves many matrix inversions. In reality, the algorithm with interpolating all the stations' missing data first takes ~10 mins to run 22 times (a complete leave-one-out process) while the algorithm without any interpolation takes ~200 mins to run 22 times. If a complete leave-one-out

20    process takes 200 min to run, it will be nearly impossible to implement the simulation experiments shown in Section 3.3. We have now clarified our motivation for requiring data from all the stations to operate the GPR model in this paper.

Modified text on Page 5, lines 6–10:

"**While mathematically the GPR model can operate without requiring data from all the stations to be non-missing on each day by relying on the GPR model to fill in each day's missing station data, we practically required** concurrent

25    measurements of all the stations **in this paper to drastically increase the speed of the algorithm (~10 mins to run a complete 22-fold leave-one-out CV, up to ~20 times faster) by avoiding the expensive computational cost of excessive amount of matrix inversion that can be incurred otherwise.** We linearly interpolated **the** $PM_{2.5}$ values for the hours with missing measurements for each station, after which we averaged the hourly data to daily resolution as the model inputs."

30    *Page 5, Line 20: The meaning of "with that of after missing data imputation" is not clear.*

**Response:** "with that of after missing data imputation" means "with 1 h $PM_{2.5}$'s completeness after missing data imputation". The whole sentence "The comparison of initial 1 h $PM_{2.5}$'s completeness with that of after missing data imputation for both reference and low-cost nodes is detailed in Table 1" means "The comparison between the initial 1 h $PM_{2.5}$'s completeness

and the 1 h PM$_{2.5}$'s completeness after missing data imputation for both reference and low-cost nodes is detailed in Table 1". We have now revised the sentence to make its meaning clearer.

Modified text on Page 5, lines 20–21:

"**The comparison of 1 h PM$_{2.5}$'s completeness before and after** missing data imputation for both reference and low-cost nodes is detailed in Table 1 and…"

*Page 5, Line 21: I don't know if "imputed" is the correct word to use here.*

**Response:** Imputation just means replacing missing values with estimated values based on available information. Therefore, the word "imputed" seems reasonable to me.

Text remains unmodified.

*Page 5, Line 29: Should be "while outliers have scores significantly larger than 1".*

**Response:** Thank you, we have revised the sentence accordingly.

Modified text on Page 5, line 29:

"Normal observations tend to have LOF scores near 1 while outliers **have scores** significantly larger than 1."

*Page 6, Line 19 to Page 7, Line 5: This description could be improved. In particular, it is not clear how the alpha and beta parameters of Equation 3 are determined. The description seems to combine a linear regression and a calibration of the hyperparam- eters of the Gaussian Process. These two steps should be described separately.*

**Response:** We agree with the reviewer on that "how the alpha and beta parameters of Equation 3 are determined" should be more clearly described. The $\alpha_i$ and $\beta_i$ parameters of Equation 3 were determined by fitting a simple linear regression model to all available pairs of daily PM$_{2.5}$ mass concentrations from the uncalibrated low-cost node $i$ (independent variable) and its closest reference node (dependent variable). The $\alpha_i$ and $\beta_i$ parameters are the slope and intercept of the fitted simple linear regression calibration model, respectively. As shown in Equation 3, the $\alpha_i$ and $\beta_i$ calibration factors were then used to calibrate each low-cost node $i$ to its closest reference node to bridge disagreements between low-cost and reference node measurements which led to a more consistent spatial interpolation and a faster convergence during model optimization. Therefore, "the linear regression step (called low-cost node initialization, corresponding to step 2 in Fig. 3, described on page 6, lines 19-23) and the training/optimization of the hyperparameters of the GPR model (corresponding to step 3 in Fig. 3, described on page 6, lines 23-29, starting from 'After standardizing')" were previously described separately. To better highlight this fact and in order to avoid confusion, we have now added additional details to the low-cost node initialization step, have split the descriptions of the two steps into two separate paragraphs, have re-organized the places of Equations 3-5, and have added additional texts to explain the terms in Equations 3-5.

Modified text from Page 6, line 19 to Page 7, line 5:

"What separates our method from standard GP applications is the simultaneous incorporation of calibration for the low-cost nodes using a simple linear regression model into the spatial model. Linear regression has previously been shown to be effective at calibrating PM sensors (Zheng et al., 2018). **Linear regression was first used to initialize low-cost nodes' calibrations (step two in Fig. 3). In this step**, each low-cost node $i$ was **linearly** calibrated **to** its closest reference node using Eq. (3), where the calibration factors $\alpha_i$ (slope) and $\beta_i$ (intercept) were determined by fitting a simple linear regression model to all available pairs of daily PM$_{2.5}$ mass concentrations from the uncalibrated low-cost node $i$ (independent variable) and its closest reference node (dependent variable). This step aims to bridge disagreements between low-cost and reference node measurements, which can lead to a more consistent spatial interpolation and a faster convergence during the GPR model optimization.

$$r_i = \begin{cases} y_i, & \text{if reference node} \\ \alpha_i \cdot y_i + \beta_i, & \text{if low} - \text{cost node} \end{cases} \tag{3}$$

where $y_i$ **is either a vector of all the daily PM$_{2.5}$ measurements of reference node $i$ or a vector of all the daily raw PM$_{2.5}$ signals of low-cost node $i$; $r_i$ is either a vector of all the daily PM$_{2.5}$ measurements of reference node $i$ or a vector of all the daily calibrated PM$_{2.5}$ measurements of** low-cost node $i$; $\alpha_i$ and $\beta_i$ are the slope and intercept, respectively, **determined from the fitted simple linear regression** calibration equation **with daily PM$_{2.5}$ mass concentrations of the uncalibrated** low-cost node $i$ **as independent variable and PM$_{2.5}$ mass concentrations of low-cost node $i$'s** closest reference node **as dependent variable**.

**In the next step (step three in Fig. 3)**, a GPR model was fit to **each day $t$'s** 31 nodes (i.e., 10 initialized low-cost nodes and 21 reference nodes) as described in Eq. (4). **Prior to the GPR model fitting, all the PM$_{2.5}$ measurements of the 31 nodes over 59 valid days used for GPR model hyperparameters training were standardized. The standardization was performed by first concatenating all these training PM$_{2.5}$ measurements (from the 31 nodes over 59 days), then** subtracting **their** mean $\mu_{training}$ and **dividing them by their standard deviation** $s_{training}$ (i.e., transforming **all** the **training** PM$_{2.5}$ measurements to have a zero mean and unit variance). **After the standardization of training samples,** the GPR was trained to maximize the log marginal likelihood over all 59 days **using Eq. 5 and** using an L-BFGS-B optimizer (Byrd et al., 1994). To avoid bad local minima, several random hyperparameter initializations were tried and the initialization **that resulted in** the **largest** log marginal likelihood **after optimization** was chosen **(in this paper, $\Theta = [\sigma_s^2, l, \sigma_n^2]$ was initialized to [0.1, 50, 0.01]).**"

$$r_t | \Gamma \sim N(\mu, \Sigma) \tag{4}$$

where $t$ **ranges from 1 (inclusive) to 59 (inclusive); $r_t \in \mathbb{R}^{31}$ is a vector** of all 31 nodes' PM$_{2.5}$ measurements **(calibrated if low-cost nodes)** on day $t$; $\Gamma = \{x_1, \dots, x_{31}\}$ **denotes 31 nodes' locations and $x_i \in \mathbb{R}^2$ is a vector of the latitude and longitude of node $i$; $\mu \in \mathbb{R}^{31}$ represents the mean function (assumed to be 0 in this study) and $\Sigma \in \mathbb{R}^{31 \times 31}$ with $\Sigma_{ij} = K(x_i, x_j; \Theta)$ represents the covariance function/kernel function.**

$$\arg\max_{\Theta} L(\Theta) = \arg\max_{\Theta} \sum_{t=1}^{59} \log p(\boldsymbol{r}_t | \Theta) = \arg\max_{\Theta} \left(-0.5 \cdot 59 \cdot \log|\boldsymbol{\Sigma}_\theta| - 0.5 \sum_{t=1}^{59} \boldsymbol{r}_t^T \boldsymbol{\Sigma}_\theta^{-1} \boldsymbol{r}_t\right) \qquad (5)$$

where $\Theta \in \mathbb{R}^3$ is a vector of **the GPR hyperparameters** $\sigma_s^2$, $l$, and $\sigma_n^2$."

*Page 6, Lines 23-24: The process of "standardization" is not clear to me. If this is done separately for each node, wouldn't*
5 *this eliminate any systematic differences between measurement locations? If this step is only done to the data which are to be used for calibrating the model hyperparameters, then that should be stated. Even so, it is not clear that this is an appropriate step; for example, two node may be systematically higher than other locations, and so should have a mutual correlation, while if the means are subtracted, the data from the nodes would no longer be correlated (in other words, two variables can be made similar in a GP model either by giving them a high mutual correlation or by giving them a smaller prior variance*
10 *and the same prior mean).*

**Response:** We agree with the reviewer on that the standardization process should be more clearly described. First, the standardization was not done separately for each node. The original text ("After standardizing the PM$_{2.5}$ measurements for each node…") did not describe the process accurately. All the PM$_{2.5}$ measurements of the 31 nodes over 59 valid days used for GPR model hyperparameters training were standardized at once. The standardization was performed by first
15 concatenating all these training PM$_{2.5}$ measurements (from the 31 nodes over 59 days), then subtracting their mean $\mu_{training}$ and dividing them by their standard deviation $s_{training}$ (i.e., transforming all the training PM$_{2.5}$ measurements to have a zero mean and unit variance). Therefore, the standardization done in this way will not eliminate any systematic differences between measurement locations. Second, the standardization was only done to the data used for training/optimizing the hyperparameters of the GPR model (i.e., all the PM$_{2.5}$ measurements of the 31 nodes over 59 valid days). The holdout node's
20 PM$_{2.5}$ measurements were never used to calculate the $\mu_{training}$ and $s_{training}$. Third, assuming the mean function $\boldsymbol{\mu} \in \mathbb{R}^{31}$ to be **0** in this study along with standardizing all the training stations' PM$_{2.5}$ measurements to have a zero mean and unit variance is absolutely an appropriate step and will not destroy the correlations. The correlations can be learned from the covariance matrix. Assuming a same mean value for all the stations is one of the common modelling formulations on the GPR model and the simplest one. Alternative modelling formulations include a station-specific mean function (lack of prior
25 information for this project), a time-dependent mean function (computationally expensive), and a combination of both. These relatively complex formulations were not considered for this paper.

Modified text on Page 6, lines 23-26:

"**In the next step (step three in Fig. 3)**, a GPR model was fit to **each day $t$'s** 31 nodes (i.e., 10 initialized low-cost nodes and 21 reference nodes) as described in Eq. (4). **Prior to the GPR model fitting, all the PM$_{2.5}$ measurements of the 31**
30 **nodes over 59 valid days used for GPR model hyperparameters training were standardized. The standardization was performed by first concatenating all these training PM$_{2.5}$ measurements (from the 31 nodes over 59 days), then** subtracting **their** mean $\mu_{training}$ and **dividing them by their standard deviation** $s_{training}$ (i.e., transforming **all** the **training** PM$_{2.5}$ measurements to have a zero mean and unit variance). **It is worth noting that assuming the mean function**

$\mu \in \mathbb{R}^{31}$ **to be 0 along with standardizing all the training PM$_{2.5}$ samples in this study is one of the common modelling formulations on the GPR model and the simplest one. More complex formulations including a station-specific mean function (lack of prior information for this project), a time-dependent mean function (computationally expensive), and a combination of both were not considered for this paper. After the standardization of training samples,** the GPR was trained to maximize the log marginal likelihood over all 59 days **using Eq. 5 and** using an L-BFGS-B optimizer (Byrd et al., 1994)."

*Equation 4: What is Gamma?*

**Response:** Gamma, $\Gamma = \{x_1, \dots, x_{31}\}$, denotes 31 nodes' locations and $x_i \in \mathbb{R}^2$ is a vector of the latitude and longitude of node $i$. This was originally stated on page 6, lines 5-6. We have now also added the description of the $\Gamma$ term to Equation 4.

Modified Equation 4:

"$r_t | \Gamma \sim N(\mu, \Sigma)$ $\qquad\qquad\qquad\qquad\qquad\qquad\qquad\qquad\qquad\qquad\qquad\qquad$ (4)

**where $t$ ranges from 1 (inclusive) to 59 (inclusive); $r_t \in \mathbb{R}^{31}$ is a vector** of all 31 nodes' PM$_{2.5}$ measurements **(calibrated if low-cost nodes)** on day $t$; $\Gamma = \{x_1, \dots, x_{31}\}$ **denotes 31 nodes' locations and $x_i \in \mathbb{R}^2$ is a vector of the latitude and longitude of node $i$; $\mu \in \mathbb{R}^{31}$ represents the mean function (assumed to be 0 in this study) and $\Sigma \in \mathbb{R}^{31 \times 31}$ with $\Sigma_{ij} = K(x_i, x_j; \Theta)$ represents the covariance function/kernel function.**"

*Page 7, line 7: What does the bold-face Theta denote? Are these the hyperparameters of the GP model as described in Equation 2?*

**Response:** The bold-face Theta ($\Theta \in \mathbb{R}^3$) denotes the vector of the GPR hyperparameters $\sigma_s^2$, $l$, and $\sigma_n^2$. This was originally stated in Equation 5. Yes, these are the same hyperparameters of the GPR model as described in Equation 2. Thank you for pointing out the discrepancy in the Theta notation between Equations 2 and 5. We have now changed the Theta notation in both Equations 1 and 2 to bold-face Theta ($\Theta \in \mathbb{R}^3$).

Modified Equation 1:

"$y_t | \Gamma \sim N(\mu, \Sigma)$ $\qquad\qquad\qquad\qquad\qquad\qquad\qquad\qquad\qquad\qquad\qquad\qquad$ (1)

where $\mu \in \mathbb{R}^{31}$ represents the mean function (assumed to be $0$ in this study)**; $\Sigma \in \mathbb{R}^{31 \times 31}$ with $\Sigma_{ij} = K(x_i, x_j; \Theta)$** represents the covariance function/kernel function and **$\Theta$ is a vector of the GPR hyperparameters.**"

Modified Equation 2:

"$K(x_i, x_j; \Theta) = \sigma_s^2 \exp\left(-\dfrac{\|x_i - x_j\|_2^2}{2l^2}\right) + \sigma_n^2 I$  (Rasmussen and Williams, 2006) $\qquad\qquad$ (2)

where $\sigma_s^2$, $l$, and $\sigma_n^2$ are the model hyperparameters (to be optimized) that control the signal magnitude, characteristic length-scale, and noise magnitude, respectively**; $\Theta \in \mathbb{R}^3$ is a vector of the GPR hyperparameters $\sigma_s^2$, $l$, and $\sigma_n^2$.**"

*Page 7, Lines 10-11: It is not clear what it means to re-calibrate a node based on its posterior mean. I am assuming this involves adjusting the alpha and beta parameters, but this is not clear.*

**Response:** We agree with the reviewer on that not enough details were provided to fully clarify what it means to re-calibrate a low-cost node based on its conditional mean. But the reviewer's assumption is correct. "Re-calibrating a low-cost node based on its conditional mean" just means first fitting a simple linear regression model to all 59 pairs of daily PM$_{2.5}$ mass concentrations from the uncalibrated low-cost node $i$ ($y_i$, independent variable) and its conditional mean ($\mu^i_{A|B}$, dependent variable) and then using this newly fitted simple linear regression calibration model to calibrate the low-cost node $i$ again. As the reviewer said, this is essentially adjusting/updating the calibration factors $\alpha_i$ (slope) and $\beta_i$ (intercept) in Equation 3. This step is also the "simple linear regression" step of the entire "simultaneous GPR and simple linear regression" algorithm. We have now added additional descriptions to clarify this process.

Modified text on Page 7, lines 8-11:

"Once the optimum $\Theta$ for the (initial) GPR was found, we used the learned covariance function to find the mean of each low-cost node $i$'s Gaussian Distribution conditional on the remaining 30 nodes within the network (i.e., $\mu^{it}_{A|B}$) on day $t$ as described mathematically in Eq. (6)–(8) and repeatedly did so until all 59 days' $\mu^{it}_{A|B}$ (i.e., $\boldsymbol{\mu^i_{A|B}} \in \mathbb{R}^{59}$) were found and then re-calibrated that low-cost node $i$ based on the $\boldsymbol{\mu^i_{A|B}}$. **The re-calibration was done by first fitting a simple linear regression model to all 59 pairs of daily PM₂.₅ mass concentrations from the uncalibrated low-cost node $i$ ($y_i$, independent variable) and its conditional mean ($\mu^i_{A|B}$, dependent variable) and then using the updated calibration factors (slope $\alpha_i$ and intercept $\beta_i$) obtained from this newly fitted simple linear regression calibration model to calibrate the low-cost node $i$ again (using Eq. 3).**"

*Page 7, Line 29: What criteria are used for convergence?*

**Response:** The criteria used for convergence are the differences in all the GPR hyperparameters between the two adjacent runs below 0.01 (i.e., with $\Delta\sigma_s^2 \leq 0.01, \Delta l \leq 0.01, and\ \Delta\sigma_n^2 \leq 0.01$).

Modified text on Page 7, line 29:

"…until the GPR parameters $\Theta$ converged **with the convergence criteria being the differences in all the GPR hyperparameters between the two adjacent runs below 0.01 (i.e., with $\Delta\sigma_s^2 \leq 0.01, \Delta l \leq 0.01, and\ \Delta\sigma_n^2 \leq 0.01$).**"

*Page 7, Line 31: This relates to a previous comment, I believe, but it should be de- scribed how the predictions are transformed back into the original PM scale.*

**Response:** We agree that how the back transformation of the predictions was done should be more clearly described. Since standardization is just linear transformation, back transformation is relatively simple. The predictions were transformed back

by multiplying the standard deviation $s_{training}$ (the standard deviation of the training PM$_{2.5}$ measurements) and then adding back the mean $\mu_{training}$ (the mean of the training PM$_{2.5}$ measurements).

Modified text on Page 7, line 31:

"…with the standardized predictions being transformed back to the original PM$_{2.5}$ measurement scale at the end. **The back**

5 **transformation was done by multiplying the predictions by the standard deviation $s_{training}$ (the standard deviation of the training PM$_{2.5}$ measurements) and then adding back the mean $\mu_{training}$ (the mean of the training PM$_{2.5}$ measurements).**"

*Page 9, Lines 10-12: This sentence can be better written as ". . .the reference node mapping accuracy follows a pattern, with*

10 *relatively high quality prediction for those nodes whose means are close to the global mean (e.g., global mean ± SD as high-lighted with shading in Table 2) and relatively poor prediction for those nodes whose means differ substantially from the global mean (particularly on the lower end)".*

**Response:** We appreciate the suggestion and have revised the sentence accordingly.

Modified text on Page 9, lines 10-12:

15 ". . .the reference node mapping accuracy follows a pattern**,** with relatively **high-quality** prediction for those nodes whose means **were** close to the **Delhi-wide** mean (e.g., **Delhi-wide** mean ± SD as highlighted with shading in Table 2) **and** relatively poor prediction for **those nodes whose means differed substantially from** the **Delhi-wide** mean (particularly **on** the lower end)."

20 *Page 9, Line 21: It is unclear what the "scale of 10" refers to.*

**Response:** The "scale of 10" refers to the number of the low-cost nodes within the network is 10. We have now clarified this.

Modified text on Page 9, line 21:

"While only a marginal improvement **with 10 low-cost nodes in the network**, …"

25 *Page 12, Line 4: It is unclear what "quality drift estimation" is.*

**Response:** "quality drift estimation" means "high-quality drift estimation".

Modified text on Page 12, line 4:

"The **high-quality** drift estimation has therefore presented another convincing case …"

30 *Page 12, Line 11: This should be "Questions which remain unsolved".*

**Response:** Thank you for pointing this out. We have now corrected it.

Modified text on Page 12, line 11:

"Questions **which** remain unsolved are …"

*Page 13, Lines 24-27: The end of this sentence may be incomplete.*

**Response:** The sentence on Page 13, lines 21-27 is complete but very long and complex, which has caused confusion. We have now split it into several sentences.

Modified text on Page 13, lines 21-27:

"**We closely investigated** into 1) the large model calibration errors (~50 %) at two Atmos regional background sites (3-month mean $PM_{2.5}$: ~72 µg m$^{-3}$) where our E-BAMs were collocated; 2) the similarly large model prediction errors at the comparatively clean Pusa and Sector 62 reference sites; **and** 3) the washed-out local variability in the model calibrated low-cost sites**. These observations** revealed that the performance of our technique (and more generally the geostatistical techniques) can calibrate the low-cost nodes dynamically, but effective only if the degree of urban homogeneity in $PM_{2.5}$ is high. **High urban homogeneity scenarios can be that** the local contributions are as small a fraction of the regional ones as possible or the local contributions are prevalent but of similar magnitudes."

*Page 13, Line 31: Again, it is not clear what is meant by a singular node.*

**Response:** This was addressed previously. "Singular" node means an anomalous/abnormal node that reports signals that are spatially uncorrelated with other normal nodes within the network, due to under heavy influence of local sources. This echoes Section 3.3.1. The word "singular" was meant to differentiate the situation from sensor failure ("malfunctioning"). Given the confusion this word gives, we have removed "singular" throughout the manuscript. Instead, we have clarified that "malfunctioning" corresponds to two situations throughout the manuscript (i.e., sensor hardware failure and sensors under heavy influence of local sources).

Modified text on Page 13, lines 30-32:

[revised manuscript text omitted]

$$r_t | \Gamma \sim N(\mu, \Sigma) \qquad \qquad (4)$$

where $t$ ranges from 1 (inclusive) to 59 (inclusive); $r_t \in \mathbb{R}^{31}$ is a vector of all 31 nodes' PM$_{2.5}$ measurements (calibrated if low-cost nodes) on day $t$; $\Gamma = \{x_1, \ldots, x_{31}\}$ denotes 31 nodes' locations and $x_i \in \mathbb{R}^2$ is a vector of the latitude and longitude of node $i$; $\mu \in \mathbb{R}^{31}$ represents the mean function (assumed to be 0 in this study) and $\Sigma \in \mathbb{R}^{31 \times 31}$ with $\Sigma_{ij} = K(x_i, x_j; \Theta)$ represents the covariance function/kernel function.

$$\arg\max_{\Theta} L(\Theta) = \arg\max_{\Theta} \sum_{t=1}^{59} \log p(r_t|\Theta) = \arg\max_{\Theta}(-0.5 \cdot 59 \cdot \log|\Sigma_\theta| - 0.5 \sum_{t=1}^{59} r_t^T \Sigma_\theta^{-1} r_t) \qquad (5)$$

[revised manuscript text omitted]
. In this study, our MRU and IITD sites are similar to the IITM site from the studies by Tiwari et al. (2012 and 2015), which are all on campus and free from major pollution sources and therefore qualified to be regional background sites. The $PM_{2.5}$ regional background concentration during winter in Delhi was then estimated to be approximately 72 μg m$^{-3}$. The  Delhi-wide mean of the 22 reference sites was 138 μg m$^{-3}$, thus the mean local contribution across Delhi was roughly 66 μg m$^{-3}$. Clearly this ~1:1 regional–to–local ratio did not fully support the technique. Alternatively, prior 
[revised manuscript text omitted]

---

## Author Comment (AC2) · 12 Jul 2019

**Response to Comments from Reviewer #2 AMT-2019-55**

The authors would like to sincerely thank the reviewer #2 for the careful review of the manuscript and the very constructive comments which helped improve the manuscript. The reviewer's comments are in italics, the summaries of our responses are in plain font, and the changes in the manuscript are in **bold red text**. Page and line numbers refer to the original document. We also appended a marked-up manuscript version to the end of the responses to better show all the changes made from this review.

**Reviewer #2**

**Major Comments:**

*1. I am unconvinced by the authors claims that they reduce the uncertainty in the spatial interpolation of reference networks – this needs a better description and error analysis to show the 2% improvement is statistically significant and/or applicable across different parts of the network in both space and time.*

**Response:** We agree with the reviewer on that the results (i.e., p-values) of rigorous statistical tests can be more convincing to readers that modelling with low-cost nodes can decrease the extent of pure interpolation among only reference stations. We used the Wilcoxon rank-sum test, also called Mann-Whitney U test (Wilcoxon, 1945; Mann and Whitney, 1947) to prove that the accuracy improvement is significant **(at least) when the optimum number of reference stations is used.** The Wilcoxon rank-sum test is a non-parametric version of the parametric t-test (involving two independent samples/groups) that requires no specific distribution on the measurements (unlike the parametric t-test that assumes a normal distribution). Specifically in our study, for each number of reference stations, the two independent samples (100 replications per sample) are the 100 replications of the mean of the 24 h percent errors (in predicting all the holdout reference nodes) from the 100 repeated random simulations when modelling with and without the low-cost nodes, respectively. We conducted a one-sided test which has the null hypothesis that our model's mean 24 h prediction percent errors with and without including the low-cost nodes are the same (i.e., $H_0$: with = without) against the alternative that the error with the low-cost nodes is smaller than the error without them (i.e., $H_1$: with < without). The p-values of the Wilcoxon rank-sum tests are superimposed on the original Figure 9 (see the modified Figure 9 below). The level of statistical significance was chosen to be 0.05, which means that the null hypothesis (i.e., $H_0$: with = without) can be rejected in favor of the alternative (i.e., $H_1$: with < without) when p-values are below 0.05. The modified Figure 9 shows that the accuracy improvement is significant (at least) when the number of reference stations is optimum (i.e., 19 or 20). Significant accuracy improvements were also observed for 17 and 18 reference stations that had comparably low prediction errors. We think that it is reasonable/meaningful to view the entire sensor network in Delhi as a whole system and analyze, compare, and report how well this whole system performs rather

than segment it into sub-parts. Additionally, given that our sampling period was relatively short, it makes sense to analyze, compare, and report how well this whole system performs over the entire sampling period. We have now added the description and the results of the Wilcoxon rank-sum tests to the manuscript. We have also slightly modified our original findings based on the statistical test results.

Modified text on Page 13, lines 5-12:

[revised manuscript text omitted]

*2. The authors interpolate data for both reference stations and LCS without any evidence this is a valid assumption. If this is going to stand, the authors should spend some time convincing the readers this is an appropriate methodology. The authors could quite easily show this by taking periods where the data are complete and comparing the linear interpolation results to*
15 *the known concentrations. There is also no mention of the length of time covered by the temporal interpolations – are they just interpolating an hour? 12 hours?*

**Response:** We thank the reviewer for bringing up the validity of interpolation. Mathematically, the GPR model can operate without requiring data from all the stations to be non-missing on each day by relying on only the non-missing stations' covariance information on each day for inference. In this paper, the choice of attempting to interpolate all the stations'
20 missing data first was made based on some practical reasons, one of the most important being the speed of the algorithm/program. Theoretically, relying on only the non-missing stations' covariance information on each day for inference is 59 (the number of daily-averaged data points) times slower. This theoretical upper bound is 59 because the algorithm will have to loop through each of the 59 days if each day's missing reference and low-cost nodes are different. And this process is relatively computationally expensive because it involves an excessive amount of matrix inversions. In
25 reality, with 59 daily-averaged data points, the algorithm with interpolating all the stations' missing data first takes ~10 mins to run 22 times (a complete 22-fold leave-one-out cross-validation) while the algorithm without any interpolation takes ~200 mins to run 22 times. If a complete 22-fold leave-one-out cross-validation takes 200 mins to run, it will be nearly impossible to implement the simulation experiments shown in Section 3.3. We are mentioning these because **we would like to prove that interpolating data for both reference and low-cost nodes is an appropriate methodology for this paper by**
30 **showing that the accuracies of model prediction on the 22 reference nodes with and without interpolation are statistically the same**. The comparison of the model prediction percent errors for the 22 reference stations with and without interpolation is shown in the newly added Table S1 (see below). The percent errors for all the stations are essentially the same with only one exception of station Vasundhara whose error without interpolation is 10 % lower than that with interpolation. The Delhi-wide mean percent errors averaged over the 22 reference stations are also essentially the same (30 %

and 29 % for with and without interpolation, respectively). We used the Wilcoxon signed-rank test (Wilcoxon, 1945) to prove that the two related paired samples (i.e., the percent errors for the 22 reference stations with and without interpolation) are statistically the same. The Wilcoxon signed-rank test is a non-parametric version of the parametric paired t-test (involving two related/matched samples/groups) that requires no specific distribution on the measurements (unlike the parametric paired t-test that assumes a normal distribution). We conducted a two-sided test which has the null hypothesis that the percent errors for the 22 reference stations with and without interpolation are the same (i.e., $H_0$: with = without) against the alternative that they are not the same (i.e., $H_1$: with ≠ without). The p-value of the test is 0.07. The level of statistical significance was chosen to be 0.05, which means that the null hypothesis (i.e., $H_0$: with = without) cannot be rejected when the p-value is 0.07, above 0.05. Therefore, interpolating data for both reference and low-cost nodes is appropriate for this paper because the accuracies of model prediction on the 22 reference nodes with and without interpolation are not distinct based on the Wilcoxon signed-rank test result. Regarding **there is also no mention of the length of time covered by the temporal interpolations**, the periods over which 1 h data were imputed for each site are already illustrated in Fig. S1; we also already specified on Page 5, lines 8-10 that the interpolation was implemented on the 1 h averaged measurements for each station; additionally, the comparison of 1 h $PM_{2.5}$'s percentage completeness with respect to the entire sampling period (i.e., from January 1, 2018 00:00 to March 31, 2018 23:59, Indian Standard Time, IST, in total 90 days, 2160 hours) before and after missing data imputation for both reference and low-cost nodes is already provided in Table 1 (this means that a 10 % increase in the percentage completeness after interpolation is equivalent to ~216 hours of data being interpolated); Given all these pieces of information that have already been provided previously, we believe 1) readers can have a good understanding of how much data were interpolated for each station (also whether the interpolation was done an hour here and there or over a large chunk of time) and 2) curious readers can also easily work out the exact number of hours being interpolated for each station. Again, **we would like to emphasize that the interpolation approach in this paper has little effect on the model's overall prediction accuracy considering all the 22 reference stations and does not affect any of the conclusions in this paper.** We have now added the validation of our interpolation approach to Sect. 3.2.1 of the manuscript, including showing the comparison of accuracies of model prediction on the 22 reference nodes with and without interpolation in the newly added Table S1 and proving that they are not distinct based on the Wilcoxon signed-rank test. We have now also modified Table 1 caption to make it more informative about how to interpret the percentage data completeness such as indicating that a 10 % increase in the percentage completeness after interpolation is equivalent to 216 hours of 1 h data being interpolated.

Modified text on Page 5, lines 6–10:

"**While mathematically the GPR model can operate without requiring data from all the stations to be non-missing on each day by relying on only each day's non-missing stations' covariance information to make inference, we practically required** concurrent measurements of all the stations **in this paper to drastically increase the speed of the algorithm (~10 mins to run a complete 22-fold leave-one-out CV, up to ~20 times faster) by avoiding the expensive**

**computational cost of excessive amount of matrix inversions that can be incurred otherwise.** We **therefore** linearly interpolated **the 1 h** PM$_{2.5}$ values for the hours with missing measurements for each station, after which we averaged the hourly data to daily resolution as the model inputs. **We validate our interpolation approach in Sect. 3.2.1 by showing that the model accuracies with and without interpolation are statistically the same.**"

Added text on Page 9, line 13:

**"In this paper, we interpolated the missing 1 h PM$_{2.5}$ values for all the reference and low-cost stations to fulfil our requirement of concurrent measurements of all the stations. This approach drastically increased the speed of the algorithm (up to ~20 times faster) by avoiding the expensive computational cost of excessive amount of matrix**

10 **inversions that can be incurred from relying on only each day's non-missing stations' covariance information to make inference. Here we prove that the interpolation is an appropriate methodology for this paper by demonstrating that the model prediction percent errors for the 22 reference stations with and without interpolation are statistically the same. The comparison of the errors for each station can be found in Table S1. Table S1 shows that the percent errors for all the stations are essentially the same with only one exception of station Vasundhara whose error without**

15 **interpolation is 10 % lower than that with interpolation. The Delhi-wide mean percent errors with (30 %) and without interpolation (29 %) are also essentially the same. We further used the Wilcoxon signed-rank test (Wilcoxon, 1945) to prove that the two related paired samples (i.e., the percent errors for the 22 reference stations with and without interpolation) are indeed statistically the same. The Wilcoxon signed-rank test is a non-parametric version of the parametric paired t-test (involving two related/matched samples/groups) that requires no specific distribution on**

20 **the measurements (unlike the parametric paired t-test that assumes a normal distribution). We conducted a two-sided test which has the null hypothesis that the percent errors for the 22 reference stations with and without interpolation are the same (i.e., $H_0$: with = without) against the alternative that they are not the same (i.e., $H_1$: with $\neq$ without). The p-value of the test is 0.07. The level of statistical significance was chosen to be 0.05, which means that the null hypothesis (i.e., $H_0$: with = without) cannot be rejected when the p-value is 0.07, above 0.05. Therefore,**

25 **interpolating missing 1 h PM$_{2.5}$ data for both reference and low-cost nodes is appropriate for this paper because the accuracies of model prediction on the 22 reference nodes with and without interpolation are not distinct based on the Wilcoxon signed-rank test result."**

Added Table S1:

**Table S1: Comparison of the GPR model 24 h prediction percent errors for the 22 reference nodes across the 22-fold leave-one-out CV with and without interpolating the missing 1 h PM2.5 values for all the reference and low-cost stations.**

| Reference nodes | Percent error | |
|---|---|---|
| | with interpolation | without interpolation |
| Anand Vihar | 32 % | 31 % |
| Aya Nagar | 38 % | 37 % |
| Burari Cross | 39 % | 38 % |
| CRRI Mathura Road | 21 % | 21 % |
| DTU | 36 % | 35 % |
| Faridabad | 18 % | 17 % |
| IGI Airport Terminal–3 | 32 % | 32 % |
| IHBAS, Dilshad Garden | 41 % | 42 % |
| ITO | 14 % | 12 % |
| Lodhi Road | 41 % | 39 % |
| Mandir Marg | 14 % | 13 % |
| North Campus | 24 % | 24 % |
| NSIT Dawarka | 19 % | 20 % |
| Punjabi Bagh | 20 % | 20 % |
| Pusa | 70 % | 69 % |
| R K Puram | 20 % | 20 % |
| Sector125 Noida | 23 % | 21 % |
| Sector62 Noida | 60 % | 60 % |
| Shadipur | 22 % | 22 % |
| Sirifort | 18 % | 16 % |
| US Embassy | 18 % | 18 % |
| Vasundhara, Ghaziabad | 44 % | 34 % |
| Delhi-wide mean | 30 % | 29 % |
| SD | 14 % | 15 % |

Modified Table 1 caption:

*"*Table 1: Delhi PM sensor network sites along with the 1 h data **percentage** completeness **with respect to the entire sampling period (i.e.,** from January 1, 2018 00:00 to March 31, 2018 23:59, Indian Standard Time, IST; **in total 90 days,**

**2160 hours**) before and after **1 h** missing-data imputation for each individual site. **Note that a 10 % increase in the percentage data completeness after 1 h missing-data imputation is equivalent to ~216 hours of 1 h data being interpolated.**"

5    *3. The paper covers 24h averaged data – why not 1h data? Does the ability of the calibration model significantly decay? The authors note in the introduction that one of the key advantages of LCS is that they provide high temporal availability. From their results, we can only conclude the 30% accuracy applies to 24h measurements which are less resolved than most real-time reference monitors.*

**Response:** For this comment, we will answer the reviewer's last question first and then circle back to the first two.
10    Regarding **the authors note in the introduction that one of the key advantages of LCS is that they provide high temporal availability. From their results, we can only conclude the 30% accuracy applies to 24h measurements which are less resolved than most real-time reference monitors**, the statement "24h measurements which are less resolved than most real-time reference monitors" is not really true. Real-time reference monitors that are certified as the Federal Equivalent Methods (FEMs) by the US Environmental Protection Agency (EPA) are required to provide results comparable
15    to the Federal Reference Methods (FRMs) only for a 24 h but not a 1 h sampling period. Met One Instruments β-attenuation monitors (most of the reference instruments used in Delhi) can even report 15-min averaged measurements, instruments such as Teledyne Model T640s and ThermoScientific Model 5030 SHARP can even report 1-min averaged measurements. But just because these real-time reference monitors can report $PM_{2.5}$ values at more resolved temporal resolutions does not mean these measurements are certified or can be trusted at these temporal resolutions, particularly β-attenuation monitors which
20    are known to be very noisy at finer temporal resolutions. So, to answer **the paper covers 24h averaged data – why not 1h data,** real-time reference monitors that are certified as the FEMs by the US EPA are required to provide results comparable to the FRMs only for a 24 h but not a 1 h sampling period. Our algorithm, which essentially relies on the accuracy of the reference measurements, can thus only calibrate/predict as well as the reference methods measure. Therefore, in this paper, we only reported our algorithm's accuracy on the more reliable 24 h data. Regarding **does the ability of the calibration**
25    **model significantly decay,** the model's mean percent error did increase from 30 % at 24 h to 49 % at 1 h. However, as we mentioned previously, a large majority of the increase in percent error should arise from the uncertainty about the 1 h reference measurements. This 1 h percent error is not a fair representation of our algorithm's true calibration/prediction ability and cannot be used as evidence to suggest that our model's accuracy will truly get significantly worse at an 1 h resolution. We have now added why we reported our algorithm's accuracy on 24 h averaged data only rather than on 1 h data
30    to the manuscript.

Added text on Page 9, line 3:

"**In this paper, we reported our algorithm's accuracy on the 24 h data only rather than on the 1 h data because real-time reference monitors that are certified as the Federal Equivalent Methods (FEMs) by the US Environmental Protection Agency (EPA) are required to provide results comparable to the Federal Reference Methods (FRMs) only**

**for a 24 h but not a 1 h sampling period. Our algorithm, which essentially relies on the accuracy of the reference measurements, can only calibrate/predict as well as the reference methods measure. Therefore, only the percent error based on the reliable 24 h reference measurements is a fair representation of our algorithm's true calibration/prediction ability.**"

*4. The author's base a large chunk of their model on the linear regression for calibrating LCS despite the fact most recent literature on the topic suggests this in invalid – relative humidity causes the growth of hygroscopic aerosols through water uptake which causes the linear regression approach to fall apart at humidities > ~50% depending on the kappa value of the aerosol. See work by Birmingham or CMU. This should be discussed at some point since linearity is a major assumption in*

10   *your model as I understand it.*

**Response:** We thank the reviewer for bringing up the relative humidity (RH) interferences in the $PM_{2.5}$ measurements of nephelometric sensors. We are well aware of and really appreciate the important work published by Birmingham and CMU on this topic. In fact, our previous work (Zheng et al., 2018) also found a major RH influence in Research Triangle Park (RTP), NC, US (1 h RH = 64 ± 22 %) that can explain up to ~30 % of the variance in 1 min to 6 h PMS3003 $PM_{2.5}$

15   measurements. And this previous work also demonstrated that when proper RH corrections are made by empirical non-linear equations after using a more precise reference method (such as T640) to calibrate the sensors, the PMS3003s can measure $PM_{2.5}$ concentrations within ~10 % of ambient values. So, when we attempted to improve the model performance, the very first thing that came to our mind was naturally adjustment for systematic meteorology-induced influences. We attempted RH correction by incorporating an RH term in the linear regression models, where the RH values were the measurements from

20   each corresponding low-cost sensor package's embedded Adafruit DHT22 RH and temperature sensor. However, there was no improvement in the algorithm's accuracy at all. A plausible explanation is regarding the infrequently high RH conditions during the winter months in Delhi and stronger smoothing effects at longer averaging time intervals (i.e., 24 h). Our previous work suggested that the PMS3003 $PM_{2.5}$ measurements exponentially increased only when RH was above ~70%. The Delhi-wide average of the 3-month RH measured by the 10 low-cost sites was found to be 55 ± 15 %. Only 17 % and 6 % of these

25   RH values were greater than 70 % and 80 %, respectively. The infrequently high RH conditions can cause the RH-induced biases insignificant. Additionally, while previously a major RH influence was found in 1 min to 6 h $PM_{2.5}$ measurements in RTP, the influence significantly diminished in 12 h $PM_{2.5}$ measurements and was barely observable in 24 h measurements. Therefore, longer averaging time intervals can average out the RH biases. We agree with the reviewer on that the discussion about our attempt to include RH correction in our algorithm and the possible reasons why no improvements were observed

30   should appear in the manuscript. We have now added a subsection (Sect. 3.2.3) to address these issues.

Added text on Page 10, line 20:

**3.2.3 RH adjustment to the algorithm**

We attempted RH adjustment to the algorithm by incorporating an RH term in the linear regression models, where the RH values were the measurements from each corresponding low-cost sensor package's embedded Adafruit DHT22 RH and temperature sensor. However, there was no improvement in the algorithm's accuracy after RH correction. A plausible explanation is regarding the infrequently high RH conditions during the winter months in Delhi and stronger smoothing effects at longer averaging time intervals (i.e., 24 h). Our previous work (Zheng et al., 2018) suggested that the PMS3003 $PM_{2.5}$ weights exponentially increased only when RH was above ~70%. The Delhi-wide average of the 3-month RH measured by the 10 low-cost sites was found to be 55 ± 15 %. Only 17 % and 6 % of these RH values were greater than 70 % and 80 %, respectively. The infrequently high RH conditions can cause the RH-induced biases insignificant. Additionally, our previous work found that even though major RH influences can be found in 1 min to 6 h $PM_{2.5}$ measurements, the influence significantly diminished in 12 h $PM_{2.5}$ measurements and was barely observable in 24 h measurements. Therefore, longer averaging time intervals can smooth out the RH biases.

*5. Delhi has extremely complex air quality (as noted by the authors), but there was very little discussion on how these specific complexities contribute to the ability or inability of the model to perform. This is especially important for LCS as they don't suffer from random error, but error caused specifically by their inability to account for changes in aerosol composition and the underlying particle size distribution. There is a paper out by Gani et al (2019) that contains data in Delhi during this time period I think would be useful to this discussion.*

**Response:** We appreciate the reviewer suggesting the paper by Gani et al. (2019) and acknowledge their unique and valuable work that provides long-term characterization of the highly time-resolved ambient $PM_1$ composition in Delhi and insight into the role of meteorology in the concentration and composition of $PM_1$ in Delhi. However, first, we do not think that aerosol composition or size distribution can account for large errors in the calibration of low-cost PM sensors. In our current study, two low-cost nodes (i.e., MRU and IITD) were collocated with two E-BAMs throughout the entire study. We fit simple linear regression models between 24 h $PM_{2.5}$ mass concentrations of the two uncalibrated low-cost nodes (independent variable) and their respectively collocated E-BAMs (dependent variable), then used the regression models to calibrate the two low-cost nodes, respectively, and then evaluated the accuracies of the two collocation calibrations based on percent error (defined in Eq. 10 of the manuscript). The two low-cost PM nodes both have an error of ~17 % after being calibrated by collocation with E-BAMs using simple linear regression models (with the $PM_{2.5}$ as the sole regressor) at a 24 h temporal resolution. The manuscript of E-BAM indicates a conservative 10 % error in the 1 h measurements. This is roughly equivalent to a $10 / \sqrt{24} = 2$ % error in the 24 h measurements. This means the upper bound of all sorts of remaining errors including aerosol composition or size distribution biases along with any other possible sources of interferences/errors is at most ~15 %. Therefore, the contribution of aerosol composition or size distribution to total calibration errors is not likely to

be huge. Second, from a practical perspective, to correct for aerosol compositions and size distributions, an additional series of instruments such as those detailed in Section 2.1 of Gani et al. (2019) are needed. To adequately capture the variations of aerosol compositions and size distributions in Delhi, these high-cost instruments will have to be deployed at several locations across Delhi. Dramatically increasing the cost just to gain a few percent of accuracy rates seems to go against the original

5  intention of using low-cost sensing, therefore impractical. Third, while the data provided by Gani et al. (2019) are highly time-resolved, they are not highly space-resolved (measurements taken at one single site, IITD). Even though this single site's speciated $PM_1$ data were very informative about the overall complex air quality issues in Delhi (i.e., influence of both primary emissions and potentially more regional secondary processes), they are far from sufficient for a full discussion about our spatial model's performance ability at all 22 reference stations across the entire Delhi. Overall, given the aforementioned

10  three reasons, we think that the requested discussion is not applicable to or out of the scope of this paper.

Text remains unmodified.

**Minor Comments:**

*P. 1, L. 1: I don't think it's necessary to create unneeded acronyms – simply call them "sensor networks"*

**Response:** We agree with the reviewer on that it is not necessary to create an additional acronym out of "sensor networks".

15  In fact, "sensor networks" were all fully spelled out after abstract in the main manuscript. The only reason that we used "WLPMSNs" to represent "wireless low-cost particulate matter sensor networks" in the abstract is because "wireless low-cost particulate matter sensor networks" has been used multiple times in the abstract and fully spelling it out can take too much space. Therefore, we decided to keep using the acronym "WLPMSNs" in the abstract.

Text remains unmodified.

*P. 2, L. 14: Are LCS really suffering from calibration issues? Or fundamental issues associated with using light scattering to determine the mass of particles?*

**Response:** In the introduction, this "calibration-related issues" term functions as an umbrella term that covers a broad category of relevant issues. Regarding **using light scattering to determine the mass of particles,** calibration factors varying

25  with aerosol optical properties and relative humidity interferences (as mentioned on page 2, lines 17-18) are only some (but not all) of the calibration issues that low-cost PM sensors suffer from. They also suffer severely from degradation and drift, which require **routine recalibration** involving frequent transit of the deployed sensors between the field and the reference sites. Therefore, to state that "since the emergence of low-cost PM sensors, researchers have been plagued by (only) the fundamental issues of them using light scattering to determine the mass of particles" at the start of the 2nd paragraph in the

30  introduction is an oversimplification of the challenges with low-cost PM sensors. Thus, we decided to keep this line unchanged.

Text remains unmodified.

*P. 8, L. 13: The word "global" could be switched for something more specific. Maybe "Delhi-wide" or something more descriptive.*

**Response:** We agree with the reviewer on that "global" can be confusing and should be replaced by more meaningful words. We have now changed all "global mean/average" to "Delhi-wide mean/average". We have also changed all "global trend" to "regional trend".

Modified text on Page 1, line 28:

"…and intercepts close to the **Delhi-wide** mean of true $PM_{2.5}$…"

Modified text on Page 8, line 13:

"Spatially, the **Delhi-wide** average of the 3-month mean $PM_{2.5}$…"

Modified text on Page 8, line 23:

"…model majorly captures a **regional** trend rather than fine-grained local variations."

Modified text on Page 9, line 7:

"…the optimized model's ability to simulate only the **regional** trend well."

Modified text on Page 9, lines 11-12:

"whose means close to the **Delhi-wide** mean (e.g., **Delhi-wide** mean ± SD as highlighted with shading in Table 2) while poor prediction for the means wide of the **Delhi-wide** mean (and particularly in the lower end)."

Modified text on Page 10, line 12:

"…those nodes whose means are close to the **Delhi-wide** mean."

Modified text on Page 10, line 15:

"The **Delhi-wide** mean of the 22 reference sites was 138 μg m$^{-3}$,…"

Modified text on Page 10, line 32:

"…and intercepts close to the **Delhi-wide** mean of true $PM_{2.5}$…"

Modified text on Page 13, line 17:

"…(**Delhi-wide** average of the 3-month mean $PM_{2.5}$…"

Modified text on Page 13, line 28:

"…those nodes whose means are close to the **Delhi-wide** mean."

Modified text on Page 27, line 4:

"…[**Delhi-wide** mean ± SD, i.e., 138 ± 31]…"

Modified text on Page 27, Table 2:

"**Delhi-wide** mean"

*P. 10, L. 14: I disagree that IITD qualifies to be a background site when it is close to major roadways – see Gani et al (2019). They show a huge amount of local influence, especially during certain periods throughout the winter months.*

**Response:** We agree that IITD is not really background in the way it is traditionally thought of. The only reason we called it a background site is because this site had the lowest mean $PM_{2.5}$ concentrations of 71 µg m$^{-3}$ (and significantly lower than the Delhi-wide mean of 138 µg m$^{-3}$) during the sampling period among 24 reference instruments across Delhi. But as shown in Gani et al. (2019), black carbon (BC) had a constantly year-round local influence on the IITD site, potentially due to trucks (and other diesel vehicles) as discussed in their Section 3.5. As discussed in their Section 3.4.1, the many industrial sites in the northwest of Delhi that use HCl for steel pickling were arguably the local source that had the strongest influence on the downwind IITD site during winter. The predominantly northwestern wind during winter transported extreme levels of chloride concentrations to IITD. The single one brief organic episode during winter (discussed in their Section 3.4.2) that was due to bonfires burning during the Lohri Festival should be a minor local source to the IITD site when looking at the whole winter. Therefore, considering the huge amount of local influences on the IITD site, we decided to remove our estimate of Delhi's $PM_{2.5}$ regional–to–local ratio and instead reference the estimate of Delhi's $PM_1$ regional/local contribution given in Gani et al. (2019).

Modified text on Page 10, lines 12-19:

"~~In this study, our MRU and IITD sites are similar to the IITM site from the studies by Tiwari et al. (2012 and 2015), which are all on campus and free from major pollution sources and therefore qualified to be regional background sites. The $PM_{2.5}$ regional background concentration during winter in Delhi was then estimated to be approximately 72 µg m$^{-3}$. The global mean of the 22 reference sites was 138 µg m$^{-3}$, thus the mean local contribution across Delhi was roughly 66 µg m$^{-3}$. Clearly this ~1:1 regional to local ratio did not fully support the technique. Alternatively, prior information about urban $PM_{2.5}$ spatial patterns such as high spatial resolution annual average concentration basemap from air pollution dispersion models can dramatically improve the on the fly calibration performance by correcting for the concentration range specific biases (Schneider et al., 2017).~~ **Gani et al. (2019) estimated that Delhi's local contribution to the composition-based submicron particulate matter ($PM_1$) was ~30 to 50 % during winter and spring months. Clearly the huge amount of local influence in Delhi did not fully support our technique.**"

Added reference on Page 15, line 18:

"**Gani, S., Bhandari, S., Seraj, S., Wang, D. S., Patel, K., Soni, P., Arub, Z., Habib, G., Hildebrandt Ruiz, L., and Apte, J. S.: Submicron aerosol composition in the world's most polluted megacity: the Delhi Aerosol Supersite study, Atmos. Chem. Phys., 19, 6843-6859, https://doi.org/10.5194/acp-19-6843-2019, 2019.**"

**Gaussian Process regression model for dynamically calibrating and surveilling a wireless low-cost particulate matter sensor network in Delhi**

Tongshu Zheng[1], Michael H. Bergin[1], Ronak Sutaria[2], Sachchida N. Tripathi[3], Robert Caldow[4], David E. Carlson[1,5]

[1]Department of Civil and Environmental Engineering, Duke University, Durham, NC 27708, USA
[2]Respirer Living Sciences Pvt. Ltd, 7, Maheshwar Nivas, Tilak Road, Santacruz (W), Mumbai 400054, India
[3]Department of Civil Engineering, Indian Institute of Technology Kanpur, Kanpur, Uttar Pradesh 208016, India
[4]TSI Inc., 500 Cardigan Road, Shoreview, MN 55126, USA
[5]Department of Biostatistics and Bioinformatics, Duke University, Durham, NC 27708, USA

*Correspondence to*: Tongshu Zheng (tongshu.zheng@duke.edu)

**Abstract.** Wireless low-cost particulate matter sensor networks (WLPMSNs) are transforming air quality monitoring by providing PM information at finer spatial and temporal resolutions; however, large-scale WLPMSN calibration and maintenance remain a challenge because the manual labor involved in initial calibration by collocation and routine recalibration is intensive, the transferability of the calibration models determined from initial collocation to new deployment sites is questionable as calibration factors typically vary with urban heterogeneity of operating conditions and aerosol optical properties, and the stability of low-cost sensors can develop drift or degrade over time. This study presents a simultaneous Gaussian Process regression (GPR) and simple linear regression pipeline to calibrate and monitor dense WLPMSNs on the fly by leveraging all available reference monitors across an area without resorting to pre-deployment collocation calibration. We evaluated our method for Delhi where the $PM_{2.5}$ measurements of all 22 regulatory reference and 10 low-cost nodes were available in 59 valid days from January 1, 2018 to March 31, 2018 ($PM_{2.5}$ averaged $138 \pm 31$ µg m$^{-3}$ among 22 reference stations) using a leave-one-out cross-validation (CV) over the 22 reference nodes. We showed that our approach can achieve an overall 30 % prediction error (RMSE: 33 µg m$^{-3}$) at a 24 h scale and is robust as underscored by the small variability in the GPR model parameters and in the model-produced calibration factors for the low-cost nodes among the 22-fold CV. We revealed that the accuracy of our calibrations depends on the degree of homogeneity of PM concentrations, and decreases with increasing local source contributions. As by-products of dynamic calibration, our algorithm can be adapted for automated large-scale WLPMSN monitoring as simulations proved its capability of differentiating malfunctioning or singular low-cost nodes within a network via model-generated calibration factors with the aberrant nodes having 
[revised manuscript text omitted]

30   based on its closest reference node (Eq. 3) to bridge disagreements between low-cost and reference node measurements which led to a more consistent spatial interpolation and a faster convergence during model optimization. After standardizing the PM$_{2.5}$ measurements for each node by subtracting the mean and scaling to unit variance (i.e., transforming the PM$_{2.5}$

measurements to have a zero mean and unit variance), a GPR model was fit to all 31 nodes (i.e., 10 initialized low-cost nodes and 21 reference nodes) as described in Eq. (4) and step three in Fig. 3. Then the GPR was trained to maximize the log marginal likelihood over all 59 days (Eq. 5) using an L-BFGS-B optimizer (Byrd et al., 1994). To avoid bad local minima, several random hyperparameter initializations were tried and the initialization with the best log marginal likelihood was chosen.

$$r_i = \begin{cases} y_i, & \text{if reference node} \\ \alpha_i \cdot y_i + \beta_i, & \text{if low} - \text{cost node} \end{cases} \tag{3}$$

$$r_t | \Gamma \sim N(\mu, \Sigma) \tag{4}$$

where $\alpha_i$ and $\beta_i$ are the slope and intercept, respectively, of the calibration equation for low-cost node $i$ based on its closest reference node; $r_i$ is all the daily PM$_{2.5}$ measurements of either the initially-calibrated low-cost node $i$ or reference node $i$; and $r_t$ is the concatenation of all 31 nodes' PM$_{2.5}$ measurements on day $t$.

$$\arg\max_{\Theta} L(\Theta) = \arg\max_{\Theta} \sum_{t=1}^{59} \log p(r_t|\Theta) = \arg\max_{\Theta} -0.5 \cdot 59 \cdot \log|\Sigma_\theta| - 0.5 \sum_{t=1}^{59} r_t^T \Sigma_\theta^{-1} r_t \tag{5}$$

where $\Theta$ is a vector of $\sigma_s^2$, $l$, and $\sigma_n^2$.

Once the optimum $\Theta$ for the (initial) GPR was found, we used the learned covariance function to find the mean of each low-cost node $i$'s Gaussian Distribution conditional on the remaining 30 nodes within the network (i.e., $\mu_{A|B}^{it}$) on day $t$ as described mathematically in Eq. (6)–(8) and repeatedly did so until all 59 days' $\mu_{A|B}^{it}$ (i.e., $\mu_{A|B}^i$) were found and then re-calibrated that low-cost node $i$ based on the $\mu_{A|B}^i$. This procedure is summarized graphically in Fig. 3 step four and was performed iteratively for all low-cost nodes one at a time. The reasoning behind this step is given in the Supplement. A high-level interpretation of this step is that the target low-cost node is calibrated by being weighted over the remaining nodes within the network and the $\Sigma_{AB}^{it} \Sigma_{BB}^{it\ -1}$ term computes the weights. In contrast to the inverse distance weighting interpolation which will weight the nodes used for calibration equally if they are equally distant from the target node, the GPR will value sparse information more and lower the importance of redundant information (suppose all the nodes are equally distant from the target node) as shown in Fig. S2.

$$p\left(\begin{bmatrix} r_A^{it} \\ r_B^{it} \end{bmatrix}\right) = N\left(\begin{bmatrix} r_A^{it} \\ r_B^{it} \end{bmatrix}; \begin{bmatrix} \mu_A^{it} \\ \mu_B^{it} \end{bmatrix} \begin{bmatrix} \Sigma_{AA}^{it} & \Sigma_{AB}^{it} \\ \Sigma_{BA}^{it} & \Sigma_{BB}^{it} \end{bmatrix}\right) \tag{6}$$

$$r_A^{it} | r_B^{it} \sim N(\mu_{A|B}^{it}, \Sigma_{A|B}^{it}) \tag{7}$$

$$\mu_{A|B}^{it} = \mu_A^{it} + \Sigma_{AB}^{it} \Sigma_{BB}^{it\ -1} (r_B^{it} - \mu_B^{it}) \tag{8}$$

where $r_A^{it}$ and $r_B^{it}$ are the daily PM$_{2.5}$ measurement(s) of the low-cost node $i$ and the remaining 30 nodes on day $t$; $\mu_A^{it}$, $\mu_B^{it}$, and $\mu_{A|B}^{it}$ are the mean (**vector**) of the partitioned Multivariate Gaussian Distribution of the low-cost node $i$, the remaining 30 nodes, and the low-cost node $i$ conditional on the remaining 30 nodes, respectively, on day $t$; and $\Sigma_{AA}^{it}$, $\Sigma_{AB}^{it}$, $\Sigma_{BA}^{it}$, $\Sigma_{BB}^{it}$, and

$\Sigma_{A|B}^{it}$ are the covariance between the low-cost node $i$ and itself, the low-cost node $i$ and the remaining 30 nodes, the remaining 30 nodes and the low-cost node $i$, the remaining 30 nodes and themselves, and the low-cost node $i$ conditional on the remaining 30 nodes and itself, respectively, on day $t$.

5 Iterative optimizations alternated between the GPR covariance function and the low-cost node measurements (Fig. 3 steps five and six) until the GPR parameters $\Theta$ converged. 
[revised manuscript text omitted]
. In this study, our MRU and IITD sites are similar to the IITM site from the studies by Tiwari et al. (2012 and 2015), which are all on campus and free from major pollution sources and therefore qualified to be regional background sites. The PM$_{2.5}$ regional background concentration during winter in Delhi was then estimated to be approximately 72 µg m$^{-3}$. The global mean of the 22 reference sites was 138 µg m$^{-3}$, thus the mean local contribution across Delhi was roughly 66 µg m$^{-3}$. Clearly this ~1:1 regional-to-local ratio did not fully support the technique. Alternatively, prior 
[revised manuscript text omitted]

[Figure]

**Figure S2: Simplified illustration of the relative importance (i.e., importance normalized by the max value) of each node within the network when using GPR to calibrate the target low-cost node and when all the nodes used for calibration are equally distant from the target node.**

[Figure]

**Figure S3: Box plots of the learned optimum Gaussian Process Regression model parameters including the signal variance ($\sigma^2_{sig}$), the characteristic length scale ($l$), and the noise variance ($\sigma^2_{noise}$) from the 22-fold leave-one-out cross-validation. The mean and SD of each parameter are superimposed on the box plots.**

[Figure]

**Figure S4: Gaussian Process Regression model 24 h performance scores (including RMSE and percent error) for predicting the measurements of the 22 holdout reference nodes across the 22-fold leave-one-out cross-validation using the full sensor network, when measurements of all (top left), nine (top center), seven (top right), three (bottom left), one (bottom center), and zero (bottom right) of the low-cost nodes are replaced with random integers bounded by the min and max of the true signals reported by the corresponding low-cost nodes.**

[Figure]

**Figure S5: Gaussian Process Regression model 24 h performance scores (including RMSE and percent error) for predicting the measurements of the 22 holdout reference nodes across the 22-fold leave-one-out cross-validation using the full sensor network, when measurements of two (bottom/1st row), four (2nd row), six (3rd row), eight (4th row), and all ten (top/5th row) of the low-cost nodes developed significant (11 %–99 %, left column), marginal (1 %–10 %, right column), and a balanced mixture of significant and marginal drifts. Note the sensors that drifted, the percentages of drift, and which sensors drifted significantly or marginally are randomly chosen. The results reported under each scenario are based on averages of 10 simulation runs.**

**Table S1: Comparison of the GPR model 24 h prediction percent errors for the 22 reference nodes across the 22-fold leave-one-out CV with and without interpolating the missing 1 h PM$_{2.5}$ values for all the reference and low-cost stations.**

| Reference nodes | Percent error | |
|---|---|---|
| | with interpolation | without interpolation |
| Anand Vihar | 32 % | 31 % |
| Aya Nagar | 38 % | 37 % |
| Burari Cross | 39 % | 38 % |
| CRRI Mathura Road | 21 % | 21 % |
| DTU | 36 % | 35 % |
| Faridabad | 18 % | 17 % |
| IGI Airport Terminal–3 | 32 % | 32 % |
| IHBAS, Dilshad Garden | 41 % | 42 % |
| ITO | 14 % | 12 % |
| Lodhi Road | 41 % | 39 % |
| Mandir Marg | 14 % | 13 % |
| North Campus | 24 % | 24 % |
| NSIT Dawarka | 19 % | 20 % |
| Punjabi Bagh | 20 % | 20 % |
| Pusa | 70 % | 69 % |
| R K Puram | 20 % | 20 % |
| Sector125 Noida | 23 % | 21 % |
| Sector62 Noida | 60 % | 60 % |
| Shadipur | 22 % | 22 % |
| Sirifort | 18 % | 16 % |
| US Embassy | 18 % | 18 % |
| Vasundhara, Ghaziabad | 44 % | 34 % |
| Delhi-wide mean | 30 % | 29 % |
| SD | 14 % | 15 % |

**Table S2: Comparison of pre-determined percentages of drift to those estimated from the Gaussian Process Regression model for intercept and slope, respectively, for each individual low-cost node, assuming eight and four of the low-cost nodes developed various degrees of drift such as significant (11 %–99 %), marginal (1 %–10 %), and a balanced mixture of significant and marginal. Note the sensors that drifted, the percentages of drift, and which sensors drifted significantly or marginally are randomly chosen. The results reported under each scenario are based on averages of 10 simulation runs.**

| Drift category | Low-cost nodes | Eight low-cost nodes drift | | | | Four low-cost nodes drift | | | |
| --- | --- | --- | --- | --- | --- | --- | --- | --- | --- |
| | | Intercept drift (%) | | Slope drift (%) | | Intercept drift (%) | | Slope drift (%) | |
| | | True | Estimated | True | Estimated | True | Estimated | True | Estimated |
| Significant | AIIMS | 55 % | 54 % | 55 % | 55 % | 0 % | -2 % | 0 % | 0 % |
| | Hiran Kudna | 57 % | 43 % | 54 % | 56 % | 47 % | 42 % | 54 % | 54 % |
| | IITD | 68 % | 70 % | 61 % | 61 % | 0 % | -1 % | 0 % | -1 % |
| | IITM | 0 % | -2 % | 0 % | -1 % | 0 % | -2 % | 0 % | -1 % |
| | Kaushambi | 0 % | -1 % | 0 % | -1 % | 0 % | -1 % | 0 % | -1 % |
| | MRU | 45 % | 46 % | 52 % | 51 % | 0 % | -4 % | 0 % | 1 % |
| | Mayur Vihar | 56 % | 59 % | 48 % | 47 % | 42 % | 44 % | 57 % | 56 % |
| | Naraina Vihar | 63 % | 61 % | 57 % | 57 % | 51 % | 51 % | 48 % | 48 % |
| | New Friends Colony | 53 % | 53 % | 57 % | 57 % | 70 % | 71 % | 39 % | 38 % |
| | S.D.A. Park | 55 % | 50 % | 55 % | 56 % | 0 % | -4 % | 0 % | 2 % |
| | **Mean absolute difference** | **3 %** | | **1 %** | | **2 %** | | **1 %** | |
| 50 % significant and 50 % marginal | AIIMS | 0 % | -1 % | 0 % | -1 % | 0 % | -1 % | 0 % | -1 % |
| | Hiran Kudna | 47 % | 40 % | 58 % | 58 % | 0 % | -9 % | 0 % | 3 % |
| | IITD | 57 % | 62 % | 58 % | 57 % | 0 % | 0 % | 0 % | -2 % |
| | IITM | 6 % | 5 % | 6 % | 3 % | 4 % | 3 % | 7 % | 6 % |
| | Kaushambi | 4 % | 4 % | 5 % | 1 % | 0 % | 0 % | 0 % | -2 % |
| | MRU | 47 % | 54 % | 55 % | 53 % | 0 % | -1 % | 0 % | -1 % |
| | Mayur Vihar | 56 % | 62 % | 46 % | 43 % | 44 % | 48 % | 70 % | 68 % |
| | Naraina Vihar | 5 % | 3 % | 4 % | 3 % | 58 % | 56 % | 46 % | 47 % |
| | New Friends Colony | 6 % | 7 % | 6 % | 2 % | 5 % | 6 % | 6 % | 3 % |
| | S.D.A. Park | 0 % | -3 % | 0 % | 1 % | 0 % | -3 % | 0 % | 2 % |
| | **Mean absolute difference** | **3 %** | | **2 %** | | **2 %** | | **2 %** | |
| Marginal | AIIMS | 5 % | 6 % | 4 % | 3 % | 0 % | 0 % | 0 % | -1 % |
| | Hiran Kudna | 6 % | 6 % | 7 % | 6 % | 0 % | 0 % | 0 % | 0 % |
| | IITD | 6 % | 7 % | 6 % | 4 % | 0 % | 1 % | 0 % | -1 % |
| | IITM | 5 % | 5 % | 5 % | 4 % | 0 % | 0 % | 0 % | -1 % |
| | Kaushambi | 5 % | 5 % | 5 % | 4 % | 5 % | 6 % | 7 % | 6 % |
| | MRU | 7 % | 9 % | 4 % | 2 % | 7 % | 8 % | 5 % | 4 % |
| | Mayur Vihar | 0 % | 1 % | 0 % | -1 % | 6 % | 7 % | 4 % | 3 % |
| | Naraina Vihar | 6 % | 7 % | 6 % | 5 % | 0 % | 0 % | 0 % | -1 % |
| | New Friends Colony | 0 % | 1 % | 0 % | -2 % | 0 % | 1 % | 0 % | -1 % |
| | S.D.A. Park | 5 % | 6 % | 4 % | 3 % | 7 % | 7 % | 5 % | 4 % |
| | **Mean absolute difference** | **1 %** | | **1 %** | | **1 %** | | **1 %** | |

---

## Author Comment (AC3) · 17 Jul 2019

**Response to Comments from Reviewer #3 AMT-2019-55**

The authors would like to sincerely thank the reviewer #3 for the careful review of the manuscript and the very constructive comments which helped improve the manuscript. The reviewer's comments are in italics, the summaries of our responses are in plain font, and the changes in the manuscript are in **bold red text**. Page and line numbers refer to the original document. We also appended a marked-up manuscript version to the end of the responses to better show all the changes made from this review.

**Reviewer #3**

**Major Comments:**

*1. Data are interpolated for both monitor types (LCS and reference). Why not perform analyses to validate your interpolation? For instance, by removing data of similar size to what is missing from non-missing periods and applying the same interpolation? How much consecutive data is interpolated? An hour here or there, or larger chunks of time?*

**Response:** We thank the reviewer for bringing up the validity of interpolation. Mathematically, the GPR model can operate without requiring data from all the stations to be non-missing on each day by relying on only the non-missing stations' covariance information on each day for inference. In this paper, the choice of attempting to interpolate all the stations' missing data first was made based on practical reasons. Specifically, interpolating prior to the GPR inference allows matrix inversions to be shared, greatly speeding the algorithm. To elaborate, with 59 daily-averaged data points, the algorithm with interpolating all the stations' missing data first takes ~10 mins to run 22 times (a complete 22-fold leave-one-out cross-validation) while the algorithm without any interpolation takes ~200 mins to run 22 times. If a complete 22-fold leave-one-out cross-validation takes 200 mins to run, it will be unrealistic to implement the simulation experiments shown in Section 3.3. We are mentioning these because **we would like to prove that interpolating data for both reference and low-cost nodes is an appropriate methodology for this paper by showing that the accuracies of model prediction on the 22 reference nodes with and without interpolation are statistically the same**. The comparison of the model prediction percent errors for the 22 reference stations with and without interpolation is shown in the newly added Table S1 (see below). The percent errors for all the stations are essentially the same with only one exception of station Vasundhara whose error without interpolation is 10 % lower than that with interpolation. The Delhi-wide mean percent errors averaged over the 22 reference stations are also essentially the same (30 % and 29 % for with and without interpolation, respectively). We used the Wilcoxon signed-rank test (Wilcoxon, 1945) to prove that the two related paired samples (i.e., the percent errors for the 22 reference stations with and without interpolation) are statistically the same. The Wilcoxon signed-rank test is a non-parametric version of the parametric paired t-test (involving two related/matched samples/groups) that requires no specific

distribution on the measurements (unlike the parametric paired t-test that assumes a normal distribution). We conducted a two-sided test which has the null hypothesis that the percent errors for the 22 reference stations with and without interpolation are the same (i.e., $H_0$: with = without) against the alternative that they are not the same (i.e., $H_1$: with $\neq$ without). The p-value of the test is 0.07. The level of statistical significance was chosen to be 0.05, which means that the null hypothesis (i.e., $H_0$: with = without) cannot be rejected when the p-value is 0.07, above 0.05. Therefore, interpolating data for both reference and low-cost nodes is appropriate for this paper because the accuracies of model prediction on the 22 reference nodes with and without interpolation are not distinct based on the Wilcoxon signed-rank test result. Regarding **how much consecutive data is interpolated and an hour here or there, or larger chunks of time**, all the periods over which 1 h data were interpolated for each site are already illustrated **in Fig. S1** (also mentioned on Page 5, lines 21-22); we also already specified on Page 5, lines 8-10 that the interpolation was implemented on the 1 h averaged measurements for each station; additionally, the comparison of 1 h $PM_{2.5}$'s percentage completeness with respect to the entire sampling period (i.e., from January 1, 2018 00:00 to March 31, 2018 23:59, Indian Standard Time, IST, in total 90 days, 2160 hours) before and after missing data imputation for both reference and low-cost nodes is already provided in Table 1 (this means that a 10 % increase in the percentage completeness after interpolation is equivalent to a total of ~216 hours of data being interpolated); We believe these pieces of information are sufficient for readers to have a good understanding of how much data (the exact number of hours) was interpolated for each station and whether the interpolation for each station was done an hour here and there or over a large chunk of time. Again, **we would like to emphasize that the interpolation approach in this paper has little effect on the model's overall prediction accuracy considering all the 22 reference stations and does not affect any of the conclusions in this paper.** We have now added the validation of our interpolation approach to Sect. 3.2.1 of the manuscript, including showing the comparison of accuracies of model prediction on the 22 reference nodes with and without interpolation in the newly added Table S1 and proving that they are not distinct based on the Wilcoxon signed-rank test. We have now also modified Table 1 caption to make it more informative about how to interpret the percentage data completeness such as indicating that a 10 % increase in the percentage completeness after interpolation is equivalent to 216 hours of 1 h data being interpolated.

Modified text on Page 5, lines 6–10:

"**While mathematically the GPR model can operate without requiring data from all the stations to be non-missing on each day by relying on only each day's non-missing stations' covariance information to make inference, we practically required** concurrent measurements of all the stations **in this paper to drastically increase the speed of the algorithm (~10 mins to run a complete 22-fold leave-one-out CV, up to ~20 times faster) by avoiding the expensive computational cost of excessive amount of matrix inversions that can be incurred otherwise.** We **therefore** linearly interpolated **the 1 h** $PM_{2.5}$ values for the hours with missing measurements for each station, after which we averaged the hourly data to daily resolution as the model inputs. **We validate our interpolation approach in Sect. 3.2.1 by showing that the model accuracies with and without interpolation are statistically the same.**"

Added text on Page 9, line 13:

"**In this paper, we interpolated the missing 1 h $PM_{2.5}$ values for all the reference and low-cost stations to fulfil our requirement of concurrent measurements of all the stations. This approach drastically increased the speed of the**

5   **algorithm (up to ~20 times faster) by avoiding the expensive computational cost of excessive amount of matrix inversions that can be incurred from relying on only each day's non-missing stations' covariance information to make inference. Here we prove that the interpolation is an appropriate methodology for this paper by demonstrating that the model prediction percent errors for the 22 reference stations with and without interpolation are statistically the same. The comparison of the errors for each station can be found in Table S1. Table S1 shows that the percent errors**

10   **for all the stations are essentially the same with only one exception of station Vasundhara whose error without interpolation is 10 % lower than that with interpolation. The Delhi-wide mean percent errors with (30 %) and without interpolation (29 %) are also essentially the same. We further used the Wilcoxon signed-rank test (Wilcoxon, 1945) to prove that the two related paired samples (i.e., the percent errors for the 22 reference stations with and without interpolation) are indeed statistically the same. The Wilcoxon signed-rank test is a non-parametric version of**

15   **the parametric paired t-test (involving two related/matched samples/groups) that requires no specific distribution on the measurements (unlike the parametric paired t-test that assumes a normal distribution). We conducted a two-sided test which has the null hypothesis that the percent errors for the 22 reference stations with and without interpolation are the same (i.e., $H_0$: with = without) against the alternative that they are not the same (i.e., $H_1$: with $\neq$ without). The p-value of the test is 0.07. The level of statistical significance was chosen to be 0.05, which means that**

20   **the null hypothesis (i.e., $H_0$: with = without) cannot be rejected when the p-value is 0.07, above 0.05. Therefore, interpolating missing 1 h $PM_{2.5}$ data for both reference and low-cost nodes is appropriate for this paper because the accuracies of model prediction on the 22 reference nodes with and without interpolation are not distinct based on the Wilcoxon signed-rank test result.**"

Added Table S1:

**Table S1: Comparison of the GPR model 24 h prediction percent errors for the 22 reference nodes across the 22-fold leave-one-out CV with and without interpolating the missing 1 h PM2.5 values for all the reference and low-cost stations.**

| Reference nodes | Percent error | |
|---|---|---|
| | with interpolation | without interpolation |
| Anand Vihar | 32 % | 31 % |
| Aya Nagar | 38 % | 37 % |
| Burari Cross | 39 % | 38 % |
| CRRI Mathura Road | 21 % | 21 % |
| DTU | 36 % | 35 % |
| Faridabad | 18 % | 17 % |
| IGI Airport Terminal–3 | 32 % | 32 % |
| IHBAS, Dilshad Garden | 41 % | 42 % |
| ITO | 14 % | 12 % |
| Lodhi Road | 41 % | 39 % |
| Mandir Marg | 14 % | 13 % |
| North Campus | 24 % | 24 % |
| NSIT Dawarka | 19 % | 20 % |
| Punjabi Bagh | 20 % | 20 % |
| Pusa | 70 % | 69 % |
| R K Puram | 20 % | 20 % |
| Sector125 Noida | 23 % | 21 % |
| Sector62 Noida | 60 % | 60 % |
| Shadipur | 22 % | 22 % |
| Sirifort | 18 % | 16 % |
| US Embassy | 18 % | 18 % |
| Vasundhara, Ghaziabad | 44 % | 34 % |
| Delhi-wide mean | 30 % | 29 % |
| SD | 14 % | 15 % |

5   Modified Table 1 caption:

*"*Table 1: Delhi PM sensor network sites along with the 1 h data **percentage** completeness **with respect to the entire sampling period** (**i.e.,** from January 1, 2018 00:00 to March 31, 2018 23:59, Indian Standard Time, IST; **in total 90 days, 2160 hours**) before and after **1 h** missing-data imputation for each individual site. **Note that a 10 % increase in the**

**percentage data completeness after 1 h missing-data imputation is equivalent to ~216 hours of 1 h data being interpolated.**"

*2. Speaking of interpolation and missingness: Are data missing in any specific pattern? That is, are areas that are typically reading higher levels of pollution more likely to have missing data? Do missing data occur most often on certain days (weekdays vs weekends)? Is missingness associated with ambient temperature, time of day, etc? Are certain monitors more prone to missingness?*

**Response:** We thank the reviewer for his/her scientific rigor by bringing up the informative missingness. All the periods over which 1 h data were interpolated for each site are already illustrated in Fig. S1 (also mentioned on Page 5, lines 21-22). We believe that there is no obvious pattern in the data missingness. The low-cost sensors generally had a higher percentage of data missingness than the reference instruments. This is expected as the low-cost sensors should be overall less robust to ambient conditions than the costly reference instruments that are well-built and thoroughly-tested. Certain low-cost sensors (e.g., S.D.A. Park, Naraina Vihar, and Hiran Kudna) had higher fractions of data missingness than the rest of the low-cost sensors. This is not surprising either given that the quality of these low-cost packages' electronic circuitry can vary drastically from one package to another, especially considering that the 10 low-cost nodes in this study were the very first generation/prototype of the "Atmos" devices (designed and built on our own) when we first started to build our network and we were still trying to figure out how to improve the electronic circuitry's quality and stability. We are leaning toward not classifying these two trends as patterns in the data missingness. We have now clarified that there is no obvious pattern in the data missingness in both the manuscript and Fig. S1.

Added text on Page 5, line 22:

"… in Fig. S1. **There is no obvious pattern in the data missingness.**"

Modified Figure S1 caption:

"Figure S1: Periods over which 1 h data were available for each individual site before and after missing-data imputation and a total of 59 24 h aggregated observations common to all the nodes in the network used for the on-the-fly calibration feasibility test. **The top 10 sites (i.e., from S.D.A. Park to AIIMS) are the low-cost sites and the remaining sites (i.e., from Vasundhara to Anand Vihar) are the reference sites. Note that there is no obvious pattern in the data missingness.**"

*3. Relatedly, QAQC procedures for reference monitors are not described. While this data can be hard to obtain from the relevant Indian agencies, it is important to more strongly highlight this as a potential shortcoming or to find out more data on how and how often reference monitors are maintained and calibrated.*

**Response:** Unlike the U.S. Embassy, the relevant Indian agencies did not provide any QA/QC (quality assurance/quality control) remark in any of their regulatory monitoring stations' datasets. This was mentioned in the manuscript on Page 5,

lines 4-6. We were unable to find such information including the QA/QC procedures (e.g., how and how often reference monitors are maintained and calibrated) anywhere else either. Due to lack of relevant QA/QC information (such as error flags) to exclude any measurement, all of the hourly $PM_{2.5}$ concentrations of the 21 monitoring stations operated by the Indian agencies were assumed to be correct. We have now more strongly highlighted this as a potential shortcoming for this study.

Modified text on Page 5, lines 5-6:

"however, the same procedure was not applied to the remaining 21 Indian government monitoring stations  **because neither the relevant Indian agencies provided QA/QC remarks or error flags in any of their regulatory monitoring stations' datasets nor can we obtain the QA/QC procedures (e.g., how and how often reference monitors are maintained and calibrated) for these reference monitors. Due to lack of relevant QA/QC information to exclude any measurement, all of the hourly $PM_{2.5}$ concentrations of the 21 monitoring stations operated by the Indian agencies were assumed to be correct. We would like to highlight this as a potential shortcoming of using the measurements from the Indian government monitoring stations.**"

*4. Is any correction – of raw signal or for temperature and/or humidity – performed by the LCS platform? Are any filters applied at the LCS station or in the cloud? Describe more fully.*

**Response:** No correction or filter of any kind was applied to the raw signals of the low-cost nodes over the cloud platform before we downloaded the data. We have now clarified this in the manuscript.

Added text on Page 5, line 13:

"…were downloaded using our custom-designed Application Program Interface (API). **No correction or filter of any kind was applied to the raw signals of the low-cost nodes over the cloud platform before we downloaded the data.**"

*5. Can you provide and compare data from the India Meteorology Department for aver- age temp and RH across the period you performed measurements and for the 59 days of data you used? Are they statistically distinct?*

**Response:** We retrieved the 30 min temperature and relative humidity (RH) data from the station at the Indira Gandhi International (IGI) Airport. We used the Wilcoxon rank-sum test, also called Mann-Whitney U test (Wilcoxon, 1945; Mann and Whitney, 1947) to evaluate if the daily-averaged temperature and RH measurements from the IGI Airport for the entire sampling period (i.e., from January 1 to March 31, 2018, 90 days) were statistically the same as those for the 59 days over which our algorithm was analyzed in this study. The Wilcoxon rank-sum test is a non-parametric version of the parametric t-test (involving two independent samples/groups) that requires no specific distribution on the measurements (unlike the parametric t-test that assumes a normal distribution). We did not use a paired test here because the two groups had different sample sizes (i.e., 59 and 90, respectively). We conducted a two-sided test which has the null hypotheses that the daily-averaged temperature and RH measurements for the 90 days (19 ± 5 °C, 59 ± 14 %) and the 59 days (20 ± 5 °C, 59 ± 12 %) were the same (i.e., $H_0$: Temperature$_{59\ days}$ = Temperature$_{90\ days}$ / RH$_{59\ days}$ = RH$_{90\ days}$) against the alternatives that they were

not the same (i.e., $H_1$: Temperature$_{59\ days}$ ≠ Temperature$_{90\ days}$ / RH$_{59\ days}$ ≠ RH$_{90\ days}$). The p-values for the RH and temperature comparisons are 0.59 and 0.28, respectively. The level of statistical significance was chosen to be 0.05, which means that the null hypotheses (i.e., $H_0$: Temperature$_{59\ days}$ = Temperature$_{90\ days}$ / RH$_{59\ days}$ = RH$_{90\ days}$) cannot be rejected when the p-values are both above 0.05. Therefore, the daily-averaged temperature and RH measurements from the IGI Airport for the entire sampling period and for the 59 days were not statistically distinct. Based on the reviewer 2's comments, we have also added the discussion about our attempt to include RH correction in our algorithm and the possible reasons why no improvements were observed in the manuscript under a new subsection Sect. 3.2.3. The results of RH and temperature measurement comparisons between the 90 days and the 59 days are also placed under this new subsection Sect. 3.2.3.

Added text on Page 10, line 20:

**3.2.3 RH adjustment to the algorithm**

We attempted RH adjustment to the algorithm by incorporating an RH term in the linear regression models, where the RH values were the measurements from each corresponding low-cost sensor package's embedded Adafruit DHT22 RH and temperature sensor. However, there was no improvement in the algorithm's accuracy after RH correction. A plausible explanation is regarding the infrequently high RH conditions during the winter months in Delhi and stronger smoothing effects at longer averaging time intervals (i.e., 24 h). Our previous work (Zheng et al., 2018) suggested that the PMS3003 PM$_{2.5}$ weights exponentially increased only when RH was above ~70%. The Delhi-wide average of the 3-month RH measured by the 10 low-cost sites was found to be 55 ± 15 %. Only 17 % and 6 % of these RH values were greater than 70 % and 80 %, respectively. The infrequently high RH conditions can cause the RH-induced biases insignificant. Additionally, our previous work found that even though major RH influences can be found in 1 min to 6 h PM$_{2.5}$ measurements, the influence significantly diminished in 12 h PM$_{2.5}$ measurements and was barely observable in 24 h measurements. Therefore, longer averaging time intervals can smooth out the RH biases.

Additionally, while our algorithm was analyzed over the 59 available days in this study, the daily-averaged temperature and RH measurements for the entire sampling period (i.e., from January 1 to March 31, 2018, 90 days) were statistically the same as those for the 59 days. To support this statement, we conducted the Wilcoxon rank-sum test, also called Mann-Whitney U test (Wilcoxon, 1945; Mann and Whitney, 1947) on the daily-averaged temperature and RH measurements from the Indira Gandhi International (IGI) Airport. The Wilcoxon rank-sum test is a non-parametric version of the parametric t-test (involving two independent samples/groups) that requires no specific distribution on the measurements (unlike the parametric t-test that assumes a normal distribution). We did not use a paired test here because the two groups had different sample sizes (i.e., 59 and 90, respectively). We conducted a two-sided test which has the null hypotheses that the daily-averaged temperature and RH measurements for the 90 days

(19 ± 5 °C, 59 ± 14 %) and the 59 days (20 ± 5 °C, 59 ± 12 %) were the same (i.e., $H_0$: Temperature$_{59\ days}$ = Temperature$_{90\ days}$ / RH$_{59\ days}$ = RH$_{90\ days}$) against the alternatives that they were not the same (i.e., $H_1$: Temperature$_{59\ days}$ ≠ Temperature$_{90\ days}$ / RH$_{59\ days}$ ≠ RH$_{90\ days}$). The p-values for the temperature and RH comparisons are 0.28 and 0.59, respectively. The level of statistical significance was chosen to be 0.05, which means that the null hypotheses (i.e., $H_0$: Temperature$_{59\ days}$ = Temperature$_{90\ days}$ / RH$_{59\ days}$ = RH$_{90\ days}$) cannot be rejected when the p-values are both above 0.05. Therefore, the daily-averaged temperature and RH measurements from the IGI Airport for the entire sampling period and for the 59 days were not statistically distinct.

**Minor Comments:**

*P1, L15 – insert comma after "sites is questionable"*

10 **Response:** Thanks, we have made the suggested change.

Modified text on Page 1, line 15:

"…sites is questionable**,** as calibration factors typically vary with…"

*P1, L19 – insert comma after Delhi*

15 **Response:** Thanks, we have made the suggested change.

Modified text on Page 1, line 19:

"We evaluated our method for Delhi**,** where the PM$_{2.5}$ measurements…"

*P1, L20 – rephrase – perhaps "available for 59 days. . ." If you elect to keep the word "valid", describe what makes the*

20 *data valid*

**Response:** Thanks, we have made the suggested change.

Modified text on Page 1, line 20:

"available **for** 59 days from January 1, 2018 to March 31, 2018…"

25 *P2, L15 – add "with" between "follow-up" and "routine"*

**Response:** Thanks, we have made the suggested change.

Modified text on Page 2, line 15:

"…initial calibration by collocation with reference analyzers before field deployment and follow-up **with** routine recalibration."

*P3, L18 – whole sentence is very long, but specifically, for item 3), rephrase "auto-detect the faulty and auto-correct drift nodes" to (perhaps) "auto- detect faulty and auto-correct nodes with drift"*

**Response:** Thanks, we have rephrased item 3).

Modified text on Page 3, line 18:

"…auto-detect faulty **nodes** and auto-correct **the drift of nodes** within a network via computational simulation, …"

5    *P4, L15 – replace "our" with "the"*

**Response:** Thanks, we have made the suggested change.

Modified text on Page 4, line 15:

"…KairosDB as **the** primary fast scalable time series database built on Apache Cassandra, …"

10    *P5, L12-13 – describe the API, or remove mention of it s*

**Response:** Thanks, we have removed mention of the API.

Modified text on Page 5, lines 12-13:

"Hourly uncalibrated $PM_{2.5}$ measurements from 10 Atmos low-cost nodes across Delhi between January 1, 2018 and March 31, 2018 were downloaded **from our low-cost sensor cloud platform**."

*P5, L16 – add " 's " to location*

**Response:** Thanks, we have made the suggested change.

Modified text on Page 5, line 16:

"…, the **locations** physical accessibility, …"

*P6 – describe more fully the "standardization" that occurs*

**Response:** We agree with the reviewer on that the standardization process should be more fully described. All the $PM_{2.5}$ measurements of the 31 nodes over 59 available days used for GPR model hyperparameters training were standardized at once. The standardization was performed by first concatenating all these training $PM_{2.5}$ measurements (from the 31 nodes

25    over 59 days), then subtracting their mean $\mu_{training}$ and dividing them by their standard deviation $s_{training}$ (i.e., transforming all the training $PM_{2.5}$ measurements to have a zero mean and unit variance). The standardization was only done to the data used for training/optimizing the hyperparameters of the GPR model (i.e., all the $PM_{2.5}$ measurements of the 31 nodes over 59 valid days). The holdout node's $PM_{2.5}$ measurements were never used to calculate the $\mu_{training}$ and $s_{training}$.

Modified text on Page 6, lines 23-27:

30    "**In the next step (step three in Fig. 3)**, a GPR model was fit to **each day $t$'s** 31 nodes (i.e., 10 initialized low-cost nodes and 21 reference nodes) as described in Eq. (4). **Prior to the GPR model fitting, all the $PM_{2.5}$ measurements of the 31 nodes over 59 valid days used for GPR model hyperparameters training were standardized. The standardization was performed by first concatenating all these training $PM_{2.5}$ measurements (from the 31 nodes over 59 days), then**

subtracting **their** mean $\mu_{training}$ and **dividing them by their standard deviation** $s_{training}$ (i.e., transforming **all** the **training** PM$_{2.5}$ measurements to have a zero mean and unit variance). **It is worth noting that assuming the mean function $\mu \in \mathbb{R}^{31}$ to be 0 along with standardizing all the training PM$_{2.5}$ samples in this study is one of the common modelling formulations on the GPR model and the simplest one. More complex formulations including a station-specific mean function (lack of prior information for this project), a time-dependent mean function (computationally expensive), and a combination of both were not considered for this paper. After the standardization of training samples,** the GPR was trained to maximize the log marginal likelihood over all 59 days **using Eq. 5 and** using an L-BFGS-B optimizer (Byrd et al., 1994)."

*P8, L14 – rephrase "Spatially, the global average. . ." – is this the average across all LCS and reference monitors? Or? And if it is the average across all, then does "spatially" apply?*

**Response:** The average is across only the 22 reference stations not including any low-cost sensor (as mentioned on Page 8, line 13). We have now removed "spatially".

Modified text on Page 8, lines 13-14:

" **The** global average of the 3-month mean PM$_{2.5}$ **across** the 22 reference stations was found to be $138 \pm 31$ µg m$^{-3}$."

*P9, L3 – insert comma after "decent"; consider rephrasing (what does decent mean in this context?)*

**Response:** Thanks, we have replaced "decent" with "reasonably accurate".

Modified text on Page 9, line 3:

"Although the technique is **reasonably accurate,** especially considering the minimal amount of field work involved, …"

*P9, L8-13 – While I understand that GPR would have done better absent local sources, is that realistic for these types of urban environments in places like India or China? Or even in the US, in places like Queens, Oakland, or Atlanta? Isn't the spatial heterogeneity exactly why many are considering more spatially and temporally resolved monitoring networks?*

**Response:** We are only confident that in order to use GPR for calibration, the low-cost sensors cannot be placed at sites where local sources are dominant. Even for urban environments that are dominated by regional sources, it is still not hard to find places that are under heavy influence of local sources. Even for urban environments where local sources are prevalent, it is still realistic to find some background stations with careful siting. If low-cost sensors must be placed at sites that are under strong local impact for some specific monitoring purposes (e.g., nodes that measure roadside pollutants or restaurant food-cooking emissions within neighborhoods), then calibration by collocation with reference instruments first might be the only solution as we believe that hardly any model has the ability to account for fine-grained local contributions and accurately calibrate such sensors. The simulation results in this study seem to suggest that GPR is more robust to local source disturbance (i.e., spatial heterogeneity) when detecting faulty nodes and the drift of nodes. This finding indicates that when

facing strong spatial heterogeneity, even though it may be challenging for GPR to calibrate low-cost sensors well, GPR holds great promise as a useful algorithm for large-scale sensor network management.

Text remains unmodified.

5    *P10, L14 – Neither of these sites are really background sites in the way they are traditionally thought of.*

**Response:** We agree that neither of these sites are really background sites in the way they are traditionally thought of. The only reason we called them background sites is because the two sites had the lowest mean $PM_{2.5}$ concentrations of ~71 and 73 μg m$^{-3}$ (and significantly lower than the Delhi-wide mean of 138 μg m$^{-3}$) during the sampling period among 24 reference instruments across Delhi. But as shown in Gani et al. (2019), black carbon (BC) had a constantly year-round local influence

10   on the IITD site, potentially due to trucks (and other diesel vehicles) as discussed in their Section 3.5. As discussed in their Section 3.4.1, the many industrial sites in the northwest of Delhi that use HCl for steel pickling were arguably the local source that had the strongest influence on the downwind IITD site during winter. The predominantly northwestern wind during winter transported extreme levels of chloride concentrations to IITD. The single one brief organic episode during winter (discussed in their Section 3.4.2) that was due to bonfires burning during the Lohri Festival should be a minor local

15   source to the IITD site when looking at the whole winter. Therefore, considering the huge amount of local influences on the IITD site and possibly on the MRU site as well, we decided to remove our estimate of Delhi's $PM_{2.5}$ regional–to–local ratio based on the assumption that these two sites are background sites and instead reference the estimate of Delhi's $PM_1$ regional/local contribution given in Gani et al. (2019).

Modified text on Page 10, lines 12-19:

20   "~~In this study, our MRU and IITD sites are similar to the IITM site from the studies by Tiwari et al. (2012 and 2015), which are all on campus and free from major pollution sources and therefore qualified to be regional background sites. The $PM_{2.5}$ regional background concentration during winter in Delhi was then estimated to be approximately 72 μg m$^{-3}$. The global mean of the 22 reference sites was 138 μg m$^{-3}$, thus the mean local contribution across Delhi was roughly 66 μg m$^{-3}$. Clearly this ~1:1 regional-to-local ratio did not fully support the technique. Alternatively, prior information about urban $PM_{2.5}$

25   spatial patterns such as high-spatial-resolution annual average concentration basemap from air pollution dispersion models can dramatically improve the on the fly calibration performance by correcting for the concentration range specific biases (Schneider et al., 2017).~~ **Gani et al. (2019) estimated that Delhi's local contribution to the composition-based submicron particulate matter ($PM_1$) was ~30 to 50 % during winter and spring months. Clearly the huge amount of local influence in Delhi did not fully support our technique.**"

Added reference on Page 15, line 18:

**"Gani, S., Bhandari, S., Seraj, S., Wang, D. S., Patel, K., Soni, P., Arub, Z., Habib, G., Hildebrandt Ruiz, L., and Apte, J. S.: Submicron aerosol composition in the world's most polluted megacity: the Delhi Aerosol Supersite study, Atmos. Chem. Phys., 19, 6843-6859, https://doi.org/10.5194/acp-19-6843-2019, 2019.**"

*P12, L10 – Perhaps rephrase to "The following questions remain:" or somesuch*

**Response:** Thank you for pointing out this grammatical error. We have now corrected it.

Modified text on Page 12, line 11:

5    "Questions **which** remain unsolved are …"

*Figure 2 – consider different shapes and colors (in B&W, the colors are not distinguish- able)*

**Response:** We thank the reviewer for pointing this out. We have now changed the reference stations' icon to triangle and the reference stations' text font to italic, so that readers can differentiate between reference and low-cost nodes even when the

10   manuscript is printed in black and white.

Modified Figure 2:

[Figure]

Figure 2: Locations of the 22 reference nodes (**triangle icons with *italic text***) and 10 low-cost nodes (**circle icons**) that form the Delhi PM sensor network.

Modified text on Page 4, lines 30-31:

"Figure 2 visualizes the spatial distribution of these 22 reference monitors (**triangle icons with *italic text***) and …"

Modified text on Page 5, lines 13-14:

"Figure 2 shows the sampling locations of these 10 low-cost nodes as **circle icons** and …"

*Figure 3 – a more elaborate caption may help better explain the flow (for instance, a sentence for each step)*

**Response:** We have now expanded the Figure 3 caption to help better carry readers through the algorithm and we have now revised Figure 3 to make it more informative about and more accurately reflect the entire process.

Modified Figure 3:

[revised manuscript text omitted]

15 for our model evaluation.

**2.3 Simultaneous GPR and simple linear regression calibration model**

Figure 3 shows the overall schema for the simultaneous GPR and simple linear regression dynamic calibration model. Under the context of not knowing beforehand the true calibration factors for the low-cost nodes, a leave-one-out CV approach (i.e., holding one of the 22 reference nodes out of modelling each run for model predictive performance evaluation) was adopted

20 as a surrogate to estimate our proposed model accuracy of calibrating the low-cost nodes. For each of the 22-fold CV, 31 node locations (denoted $\Gamma = \{x_1, \ldots, x_{31}\}$) were available, where $x_i$ is the latitude and longitude of node $i$. Let $y_{it}$ represent the daily PM$_{2.5}$ measurement of node $i$ on day $t$ and $y_t \in \mathbb{R}^{31}$ denote the concatenation of the daily PM$_{2.5}$ measurements recorded by the 31 nodes on day $t$. Given a finite number of node locations, a Gaussian Process (GP) becomes a Multivariate Gaussian Distribution over the nodes in the form of:

25 $$y_t | \Gamma \sim N(\mu, \Sigma) \tag{1}$$

where $\mu \in \mathbb{R}^{31}$ represents the mean function (assumed to be $\mathbf{0}$ in this study) and $\Sigma \in \mathbb{R}^{31 \times 31}$ with $\Sigma_{ij} = K(x_i, x_j; \theta)$ represents the covariance function/kernel function.

For simplicity's sake, the kernel function was set to a squared exponential (SE) covariance term to capture the spatially-correlated signals coupled with another component to constrain the independent noise:

30 $$K(x_i, x_j; \theta) = \sigma_s^2 \, exp\left(-\frac{\|x_i - x_j\|_2^2}{2l^2}\right) + \sigma_n^2 I \quad \text{(Rasmussen and Williams, 2006)} \tag{2}$$

where $\sigma_s^2$, $l$, and $\sigma_n^2$ are the model hyperparameters (to be optimized) that control the signal magnitude, characteristic length-scale, and noise magnitude, respectively.

What separates our method from standard GP applications is the simultaneous incorporation of calibration for the low-cost nodes using a simple linear regression model into the spatial model. Linear regression has previously been shown to be effective at calibrating PM sensors (Zheng et al., 2018). Initially (step two in Fig. 3), each low-cost node was calibrated based on its closest reference node (Eq. 3) to bridge disagreements between low-cost and reference node measurements which led to a more consistent spatial interpolation and a faster convergence during model optimization. In the next step (step three in Fig. 3) , a GPR model was fit to each day $t$'s  31 nodes (i.e., 10 initialized low-cost nodes and 21 reference nodes) as described in Eq. (4).  Prior to the GPR model fitting, all the PM₂.₅ measurements of the 31 nodes over 59 valid days used for GPR model hyperparameters training were standardized. The standardization was performed by first concatenating all these training PM₂.₅ measurements (from the 31 nodes over 59 days), then subtracting their mean $\mu_{training}$ and dividing them by their standard deviation $s_{training}$ (i.e., transforming all the training PM₂.₅ measurements to have a zero mean and unit variance). It is worth noting that assuming the mean function $\mu \in \mathbb{R}^{31}$ to be 0 along with standardizing all the training PM₂.₅ samples in this study is one of the common modelling formulations on the GPR model and the simplest one. More complex formulations including a station-specific mean function (lack of prior information for this project), a time-dependent mean function (computationally expensive), and a combination of both were not considered for this paper. After the standardization of training samples,  the GPR was trained to maximize the log marginal likelihood over all 59 days  Eq. 5 and using an L-BFGS-B optimizer (Byrd et al., 1994). To avoid bad local minima, several random hyperparameter initializations were tried and the initialization with the best log marginal likelihood was chosen.

$$r_i = \begin{cases} y_i, & \text{if reference node} \\ \alpha_i \cdot y_i + \beta_i, & \text{if low} - \text{cost node} \end{cases} \tag{3}$$

$$r_t | \Gamma \sim N(\mu, \Sigma) \tag{4}$$

where $\alpha_i$ and $\beta_i$ are the slope and intercept, respectively, of the calibration equation for low-cost node $i$ based on its closest reference node; $r_i$ is all the daily PM₂.₅ measurements of either the initially-calibrated low-cost node $i$ or reference node $i$; and $r_t$ is the concatenation of all 31 nodes' PM₂.₅ measurements on day $t$.

$$\arg\max_{\Theta} L(\Theta) = \arg\max_{\Theta} \sum_{t=1}^{59} \log p(r_t | \Theta) = \arg\max_{\Theta} -0.5 \cdot 59 \cdot \log|\Sigma_\theta| - 0.5 \sum_{t=1}^{59} r_t^T \Sigma_\theta^{-1} r_t \tag{5}$$

where $\Theta$ is a vector of $\sigma_s^2$, $l$, and $\sigma_n^2$.

Once the optimum $\Theta$ for the (initial) GPR was found, we used the learned covariance function to find the mean of each low-cost node $i$'s Gaussian Distribution conditional on the remaining 30 nodes within the network (i.e., $\mu_{A|B}^{it}$) on day $t$ as described mathematically in Eq. (6)–(8) and repeatedly did so until all 59 days' $\mu_{A|B}^{it}$ (i.e., $\mu_{A|B}^i$) were found and then recalibrated that low-cost node $i$ based on the $\boldsymbol{\mu}_{A|B}^i$. This procedure is summarized graphically in Fig. 3 step four and was performed iteratively for all low-cost nodes one at a time. The reasoning behind this step is given in the Supplement. A high-level interpretation of this step is that the target low-cost node is calibrated by being weighted over the remaining nodes within the network and the $\boldsymbol{\Sigma}_{AB}^{it}\boldsymbol{\Sigma}_{BB}^{it\ -1}$ term computes the weights. In contrast to the inverse distance weighting interpolation

5 which will weight the nodes used for calibration equally if they are equally distant from the target node, the GPR will value sparse information more and lower the importance of redundant information (suppose all the nodes are equally distant from the target node) as shown in Fig. S2.

$$p\left(\begin{bmatrix} r_A^{it} \\ \boldsymbol{r}_B^{it} \end{bmatrix}\right) = N\left(\begin{bmatrix} r_A^{it} \\ \boldsymbol{r}_B^{it} \end{bmatrix}; \begin{bmatrix} \mu_A^{it} \\ \boldsymbol{\mu}_B^{it} \end{bmatrix} \begin{bmatrix} \Sigma_{AA}^{it} & \boldsymbol{\Sigma}_{AB}^{it} \\ \boldsymbol{\Sigma}_{BA}^{it} & \boldsymbol{\Sigma}_{BB}^{it} \end{bmatrix}\right) \tag{6}$$

$$r_A^{it}\,|\,\boldsymbol{r}_B^{it} \sim N\big(\mu_{A|B}^{it}, \Sigma_{A|B}^{it}\big) \tag{7}$$

10 $\quad \mu_{A|B}^{it} = \mu_A^{it} + \boldsymbol{\Sigma}_{AB}^{it}\boldsymbol{\Sigma}_{BB}^{it\ -1}(\boldsymbol{r}_B^{it} - \boldsymbol{\mu}_B^{it}) \tag{8}$

where $r_A^{it}$ and $\boldsymbol{r}_B^{it}$ are the daily PM2.5 measurement(s) of the low-cost node $i$ and the remaining 30 nodes on day $t$; $\mu_A^{it}$, $\boldsymbol{\mu}_B^{it}$, and $\mu_{A|B}^{it}$ are the mean (**vector**) of the partitioned Multivariate Gaussian Distribution of the low-cost node $i$, the remaining 30 nodes, and the low-cost node $i$ conditional on the remaining 30 nodes, respectively, on day $t$; and $\Sigma_{AA}^{it}$, $\boldsymbol{\Sigma}_{AB}^{it}$, $\boldsymbol{\Sigma}_{BA}^{it}$, $\boldsymbol{\Sigma}_{BB}^{it}$, and $\Sigma_{A|B}^{it}$ are the covariance between the low-cost node $i$ and itself, the low-cost node $i$ and the remaining 30 nodes, the remaining

15 30 nodes and the low-cost node $i$, the remaining 30 nodes and themselves, and the low-cost node $i$ conditional on the remaining 30 nodes and itself, respectively, on day $t$.

Iterative optimizations alternated between the GPR covariance function and the low-cost node measurements (Fig. 3 steps five and six) until the GPR parameters $\boldsymbol{\Theta}$ converged. 
[revised manuscript text omitted]
. We can rebuild a model such as every week using a rolling window (to keep the number of observations for model construction roughly unchanged) to assess the drifts in the model space over time. After that, the true calibration factors obtained from the initial collocation with reference instruments prior to deployment can be adjusted accordingly based on the model-estimated drifts. This procedure allows for real-time drift corrections to low-cost node measurements.

**3.3.3 Optimal number of reference nodes**

Questions which remain unsolved are 1) what the optimum or minimum number of reference instruments is to sustain this technique and 2) if the inclusion of low-cost nodes can effectively assist in lowering the technique's calibration/mapping inaccuracy. It is interesting to note that optimizing the model's calibration accuracy can not only directly fulfill the fundamental calibration task, but also help better the sensor network monitoring capability as an added bonus. To address these two outstanding issues, we randomly sampled subsets of all the 22 reference nodes within the network in increments of one node (i.e. from 1 to 21 nodes) and implemented our algorithm with and without incorporating the low-cost nodes, before finally computing the mean percent errors in predicting all the holdout reference nodes. To get the performance scores as close to truth as possible but without incurring excessive computational cost in the meantime, the sampling was repeated 100 times for each subset size. The calibration error in this section was defined as the mean percent errors in predicting all the holdout reference nodes further averaged over 100 simulation runs for each subset size.

Figure 9 describes the 24 h calibration percent error rate of the model as a function of the number of reference stations used for modelling with and without involving the low-cost nodes. The error rates generally decrease as the number of reference instruments increases (full network: from ~40 % with 1 node to ~29 % with 21 nodes; network excluding low-cost nodes: from ~43 % to ~30 %) but are somewhat locally variable and most pronounced when five, seven, and eight reference nodes are simulated. These bumps might simply be the result of five, seven, and eight reference nodes being relatively non-ideal (with regard to their neighboring numbers) for the technique, although the possibility of non-convergence due to the limited 100 simulation runs for each scenario cannot be ruled out. The 19 or 20 nodes emerge as the optimum numbers of reference nodes with the lowest errors of close to 28 %, while 17 to 21 nodes all yield comparably low inaccuracies (all below 30 %). The pattern discovered in our research shares certain similarities with Schneider et al. (2017) who studied the relationship between the accuracy of using colocation-calibrated low-cost nodes to map urban AQ and the number of simulated low-cost nodes for their urban-scale air pollution dispersion model and kriging-fueled data fusion technique in Oslo, Norway. Both studies indicate that at least roughly 20 nodes are essential to start producing acceptable degree of accuracy. Unlike Schneider et al. (2017) who further expanded the scope to 150 nodes by generating new synthetic stations from their established model and showed a "the more, the merrier" trend up to 50 stations, we restricted ourselves to only realistic data to investigate the relationship since we suspect that stations created from our model with approximately 30 % errors might introduce large noise which could misrepresent the true pattern. We agree with Schneider et al. (2017) that such relationships are location-specific and cannot be blindly transferred to other study sites. At last, including low-cost nodes in the model building can most of the time reduce the model's errors notably when more than nine reference nodes are sampled (i.e., when the number of simulated reference nodes is favorable for carrying out the technique). And for the comparatively ideal 17–20 nodes, we even observed approximately non-overlapping 95 % confidence intervals, suggesting significantly lower errors are yielded when low-cost nodes are incorporated. 
[revised manuscript text omitted]

[Figure]

**Figure S2: Simplified illustration of the relative importance (i.e., importance normalized by the max value) of each node within the network when using GPR to calibrate the target low-cost node and when all the nodes used for calibration are equally distant from the target node.**

[Figure]

**Figure S3: Box plots of the learned optimum Gaussian Process Regression model parameters including the signal variance ($\sigma_{sig}^2$), the characteristic length scale ($l$), and the noise variance ($\sigma_{noise}^2$) from the 22-fold leave-one-out cross-validation. The mean and SD of each parameter are superimposed on the box plots.**

[Figure]

**Figure S4: Gaussian Process Regression model 24 h performance scores (including RMSE and percent error) for predicting the measurements of the 22 holdout reference nodes across the 22-fold leave-one-out cross-validation using the full sensor network, when measurements of all (top left), nine (top center), seven (top right), three (bottom left), one (bottom center), and zero (bottom right) of the low-cost nodes are replaced with random integers bounded by the min and max of the true signals reported by the corresponding low-cost nodes.**

[Figure]

**Figure S5: Gaussian Process Regression model 24 h performance scores (including RMSE and percent error) for predicting the measurements of the 22 holdout reference nodes across the 22-fold leave-one-out cross-validation using the full sensor network, when measurements of two (bottom/1st row), four (2nd row), six (3rd row), eight (4th row), and all ten (top/5th row) of the low-cost nodes developed significant (11 %–99 %, left column), marginal (1 %–10 %, right column), and a balanced mixture of significant and marginal drifts. Note the sensors that drifted, the percentages of drift, and which sensors drifted significantly or marginally are randomly chosen. The results reported under each scenario are based on averages of 10 simulation runs.**

**Table S1: Comparison of the GPR model 24 h prediction percent errors for the 22 reference nodes across the 22-fold leave-one-out CV with and without interpolating the missing 1 h PM$_{2.5}$ values for all the reference and low-cost stations.**

| Reference nodes | Percent error | |
|---|---|---|
| | with interpolation | without interpolation |
| Anand Vihar | 32 % | 31 % |
| Aya Nagar | 38 % | 37 % |
| Burari Cross | 39 % | 38 % |
| CRRI Mathura Road | 21 % | 21 % |
| DTU | 36 % | 35 % |
| Faridabad | 18 % | 17 % |
| IGI Airport Terminal–3 | 32 % | 32 % |
| IHBAS, Dilshad Garden | 41 % | 42 % |
| ITO | 14 % | 12 % |
| Lodhi Road | 41 % | 39 % |
| Mandir Marg | 14 % | 13 % |
| North Campus | 24 % | 24 % |
| NSIT Dawarka | 19 % | 20 % |
| Punjabi Bagh | 20 % | 20 % |
| Pusa | 70 % | 69 % |
| R K Puram | 20 % | 20 % |
| Sector125 Noida | 23 % | 21 % |
| Sector62 Noida | 60 % | 60 % |
| Shadipur | 22 % | 22 % |
| Sirifort | 18 % | 16 % |
| US Embassy | 18 % | 18 % |
| Vasundhara, Ghaziabad | 44 % | 34 % |
| Delhi-wide mean | 30 % | 29 % |
| SD | 14 % | 15 % |

**Table S2: Comparison of pre-determined percentages of drift to those estimated from the Gaussian Process Regression model for intercept and slope, respectively, for each individual low-cost node, assuming eight and four of the low-cost nodes developed various degrees of drift such as significant (11 %–99 %), marginal (1 %–10 %), and a balanced mixture of significant and marginal. Note the sensors that drifted, the percentages of drift, and which sensors drifted significantly or marginally are randomly chosen. The results reported under each scenario are based on averages of 10 simulation runs.**

| Drift category | Low-cost nodes | Eight low-cost nodes drift | | | | Four low-cost nodes drift | | | |
| --- | --- | --- | --- | --- | --- | --- | --- | --- | --- |
| | | Intercept drift (%) | | Slope drift (%) | | Intercept drift (%) | | Slope drift (%) | |
| | | True | Estimated | True | Estimated | True | Estimated | True | Estimated |
| Significant | AIIMS | 55 % | 54 % | 55 % | 55 % | 0 % | -2 % | 0 % | 0 % |
| | Hiran Kudna | 57 % | 43 % | 54 % | 56 % | 47 % | 42 % | 54 % | 54 % |
| | IITD | 68 % | 70 % | 61 % | 61 % | 0 % | -1 % | 0 % | -1 % |
| | IITM | 0 % | -2 % | 0 % | -1 % | 0 % | -2 % | 0 % | -1 % |
| | Kaushambi | 0 % | -1 % | 0 % | -1 % | 0 % | -1 % | 0 % | -1 % |
| | MRU | 45 % | 46 % | 52 % | 51 % | 0 % | -4 % | 0 % | 1 % |
| | Mayur Vihar | 56 % | 59 % | 48 % | 47 % | 42 % | 44 % | 57 % | 56 % |
| | Naraina Vihar | 63 % | 61 % | 57 % | 57 % | 51 % | 51 % | 48 % | 48 % |
| | New Friends Colony | 53 % | 53 % | 57 % | 57 % | 70 % | 71 % | 39 % | 38 % |
| | S.D.A. Park | 55 % | 50 % | 55 % | 56 % | 0 % | -4 % | 0 % | 2 % |
| | **Mean absolute difference** | **3 %** | | **1 %** | | **2 %** | | **1 %** | |
| 50 % significant and 50 % marginal | AIIMS | 0 % | -1 % | 0 % | -1 % | 0 % | -1 % | 0 % | -1 % |
| | Hiran Kudna | 47 % | 40 % | 58 % | 58 % | 0 % | -9 % | 0 % | 3 % |
| | IITD | 57 % | 62 % | 58 % | 57 % | 0 % | 0 % | 0 % | -2 % |
| | IITM | 6 % | 5 % | 6 % | 3 % | 4 % | 3 % | 7 % | 6 % |
| | Kaushambi | 4 % | 4 % | 5 % | 1 % | 0 % | 0 % | 0 % | -2 % |
| | MRU | 47 % | 54 % | 55 % | 53 % | 0 % | -1 % | 0 % | -1 % |
| | Mayur Vihar | 56 % | 62 % | 46 % | 43 % | 44 % | 48 % | 70 % | 68 % |
| | Naraina Vihar | 5 % | 3 % | 4 % | 3 % | 58 % | 56 % | 46 % | 47 % |
| | New Friends Colony | 6 % | 7 % | 6 % | 2 % | 5 % | 6 % | 6 % | 3 % |
| | S.D.A. Park | 0 % | -3 % | 0 % | 1 % | 0 % | -3 % | 0 % | 2 % |
| | **Mean absolute difference** | **3 %** | | **2 %** | | **2 %** | | **2 %** | |
| Marginal | AIIMS | 5 % | 6 % | 4 % | 3 % | 0 % | 0 % | 0 % | -1 % |
| | Hiran Kudna | 6 % | 6 % | 7 % | 6 % | 0 % | 0 % | 0 % | 0 % |
| | IITD | 6 % | 7 % | 6 % | 4 % | 0 % | 1 % | 0 % | -1 % |
| | IITM | 5 % | 5 % | 5 % | 4 % | 0 % | 0 % | 0 % | -1 % |
| | Kaushambi | 5 % | 5 % | 5 % | 4 % | 5 % | 6 % | 7 % | 6 % |
| | MRU | 7 % | 9 % | 4 % | 2 % | 7 % | 8 % | 5 % | 4 % |
| | Mayur Vihar | 0 % | 1 % | 0 % | -1 % | 6 % | 7 % | 4 % | 3 % |
| | Naraina Vihar | 6 % | 7 % | 6 % | 5 % | 0 % | 0 % | 0 % | -1 % |
| | New Friends Colony | 0 % | 1 % | 0 % | -2 % | 0 % | 1 % | 0 % | -1 % |
| | S.D.A. Park | 5 % | 6 % | 4 % | 3 % | 7 % | 7 % | 5 % | 4 % |
| | **Mean absolute difference** | **1 %** | | **1 %** | | **1 %** | | **1 %** | |

---

## Author Response (AR1)

**Response to Comments from Reviewer #1 AMT-2019-55**

The authors would like to sincerely thank the reviewer #1 for the careful review of the manuscript, the quick feedback, and the very constructive comments which helped dramatically improve the manuscript. The reviewer's comments are in italics, the summaries of our responses are in plain font, and the changes in the manuscript are in **bold red text**. Page and line numbers refer to the original document. We also appended a marked-up manuscript version to the end of the responses to better show all the changes made from this review.

**Reviewer #1**

5

**General Comments:**

First, a better description of the algorithm should be supplied. The equations are not matched well with their descriptions in

10 the text, and multiple steps of the process (e.g. the linear regression and Gaussian Process hyperparameter optimization) are described simultaneously. The diagrams of figure 3 are helpful, but not sufficient to clarify the entire process. A complete step-by-step breakdown of an example run of the algorithm could be provided.

**Response:** We agree with the reviewer on that the algorithm should be better described in Section 2.3. Based on the reviewer's specific comments, we have now provided more details about how the alpha and beta parameters of Equation 3

- 15 were determined, how the standardization process was implemented, what it meant to re-calibrate a low-cost node based on its conditional mean, what criteria were used for convergence, and how the predictions were transformed back to the original PM scale. Regarding 1) "The equations are not matched well with their descriptions in the text", we identified problems such as lack of the description of the  $y_i$  term in Equation 3, lack of the description of the  $\Gamma$  term in Equation 4, discrepancy in the Theta notation between Equations 2 and 5, lack of the description of the Theta term in Equations 2 and 5. We have
- 20 now corrected these issues. Regarding 2) "multiple steps of the process (e.g. the linear regression and Gaussian Process hyperparameter optimization) are described simultaneously", we presume that this comment is connected to the reviewer's specific comment on Page 6, Line 19 to Page 7, Line 5. The linear regression step (called low-cost node initialization, corresponding to step 2 in Fig. 3, described on page 6, lines 19-23) and the training/optimization of the hyperparameters of the GPR model (corresponding to step 3 in Fig. 3, described on page 6, lines 23-29, starting from 'After
- 25 standardizing') were previously described separately. To better highlight this fact and in order to avoid confusion, we have now added additional details to the low-cost node initialization step, have split the descriptions of the two steps into two separate paragraphs, have re-organized the places of Equations 3-5, and have added additional texts to explain the terms in Equations 3-5. Additionally, we have now placed each critical step under a sub-section (e.g., Sect. 2.3.x) to facilitate reading.

Regarding 3) "The diagrams of figure 3 are helpful, but not sufficient to clarify the entire process", we have now revised Figure 3 to make it more informative about and more accurately reflect the entire process and we have now expanded the Figure 3 caption to help better carry readers through the algorithm. Regarding 4) "A complete step-by-step breakdown of an example run of the algorithm could be provided", we have now added a detailed algorithm block along with a sub-

5 section number next to each critical step to indicate under which sub-section the details of that step can be found. The Section 2.3 has been completely overhauled.

Modified Section 2.3:

"2.3 Simultaneous GPR and simple linear regression calibration model

The simultaneous GPR and simple linear regression calibration algorithm is introduced here as Algorithm 1. The 10 critical steps of the algorithm are linked to sub-sections under which the respective details can be found. Complementing Algorithm 1, a flow diagram illustrating the algorithm is given in Figure 3.

**Algorithm 1: Algorithm of simultaneous GPR and simple linear regression**

for each reference node (denote:  $Ref_k$ ) in the network do

15 leave Refk out as test sample (see Sect. 2.3.1 for details)

for each low-cost node (denote: Low-costi) in the network **do**

find Low-costi's closest reference node (denote: Refi) (Sect. 2.3.2)

fit a simple linear regression model between Refi and Low-costi's PM2.5:  $Ref_i = \alpha_i \cdot Low - cost_i + \beta_i$  (Sect. 2.3.2) initialize the simple linear regression calibration factors to  $\alpha_i$  (slope) and  $\beta_i$  (intercept) for Low-costi (Sect. 2.3.2)

20 initialize the calibration of Low-costi using  $\alpha_i$  and  $\beta_i$  (Sect. 2.3.2)

**end for**

initialize GPR hyperparameters  $\Theta = [\sigma_s^2, l, \sigma_n^2]$  to [0.1, 50, 0.01] (Sect. 2.3.3)

standardize the 10 calibrated low-cost and 21 reference nodes at once (Sect. 2.3.3)

while convergence criteria not met do

25

update/optimize GPR hyperparameters **O** using the 31 standardized training nodes (Sect. 2.3.3 and .5)

for each low-cost node (denote:  $Low-cost_i$ ) in the network do

for each day (denote: t) of the 59 days do

calculate Low-costi's mean conditional on the remaining 30 nodes on day t (denote  $\mu_{A|B}^{it}$ ) (Sect. 2.3.4 and .5)

end for

30 fit a linear regression between  $\mu_{A|B}^i \in \mathbb{R}^{59}$  and Low-costi:  $\mu_{A|B}^i = \alpha_i \cdot Low - cost_i + \beta_i$  (Sect. 2.3.4 and .5) update calibration factors  $\alpha_i$  and  $\beta_i$  for Low-costi (Sect. 2.3.4 and .5) update the calibration of Low-costi using  $\alpha_i$  and  $\beta_i$  (Sect. 2.3.4 and .5)

**end for**

check convergence criteria (Sect. 2.3.5)

**end while**

use the final GPR model to predict on  $Ref_k$  (Sect. 2.3.6)

transform the prediction back to original PM2.5 scale (Sect. 2.3.6)

calculate RMSE and percent error (Sect. 2.3.6)

**end for**

5

**2.3.1 Leave one reference node out**

Because the true calibration factors for the low-cost nodes are not know beforehand, a leave-one-out CV approach (i.e.,

- 10 holding one of the 22 reference nodes out of modelling each run for model predictive performance evaluation) was adopted as a surrogate to estimate our proposed model accuracy of calibrating the low-cost nodes. For each of the 22-fold CV, 31 node locations (denoted  $\Gamma = \{x_1, ..., x_{31}\}$ ) were available, where  $x_i$  is the latitude and longitude of node *i*. Let  $y_{it}$  represent the daily PM2.5 measurement of node *i* on day *t* and  $y_t \in \mathbb{R}^{31}$  denote the concatenation of the daily PM2.5 measurements recorded by the 31 nodes on day *t*. Given a finite number of node locations, a Gaussian Process (GP) becomes a Multivariate
- 15 Gaussian Distribution over the nodes in the form of:

$$\mathbf{y}_t | \boldsymbol{\Gamma} \sim N(\boldsymbol{\mu}, \boldsymbol{\Sigma}) \tag{1}$$

where  $\mu \in \mathbb{R}^{31}$  represents the mean function (assumed to be **0** in this study);  $\Sigma \in \mathbb{R}^{31 \times 31}$  with  $\Sigma_{ij} = K(x_i, x_j; \Theta)$  represents the covariance function/kernel function and  $\Theta$  is a vector of the GPR hyperparameters."

20 For simplicity's sake, the kernel function was set to a squared exponential (SE) covariance term to capture the spatiallycorrelated signals coupled with another component to constrain the independent noise:

$$"K(\mathbf{x}_i, \mathbf{x}_j; \mathbf{\Theta}) = \sigma_s^2 \exp\left(-\frac{\|\mathbf{x}_i - \mathbf{x}_j\|_2^2}{2l^2}\right) + \sigma_n^2 \mathbf{I} \text{ (Rasmussen and Williams, 2006)}$$
(2)

where  $\sigma_s^2$ , l, and  $\sigma_n^2$  are the model hyperparameters (to be optimized) that control the signal magnitude, characteristic lengthscale, and noise magnitude, respectively;  $\Theta \in \mathbb{R}^3$  is a vector of the GPR hyperparameters  $\sigma_s^2$ , l, and  $\sigma_n^2$ ."

**25 2.3.2 Initialize low-cost nodes' (simple linear regression) calibrations**

What separates our method from standard GP applications is the simultaneous incorporation of calibration for the low-cost nodes using a simple linear regression model into the spatial model. Linear regression has previously been shown to be effective at calibrating PM sensors (Zheng et al., 2018). Linear regression was first used to initialize low-cost nodes' calibrations (step two in Fig. 3). In this step, each low-cost node *i* was linearly calibrated to its closest reference node

30 using Eq. (3), where the calibration factors  $\alpha_i$  (slope) and  $\beta_i$  (intercept) were determined by fitting a simple linear

regression model to all available pairs of daily PM2.5 mass concentrations from the uncalibrated low-cost node *i* (independent variable) and its closest reference node (dependent variable). This step aims to bridge disagreements between low-cost and reference node measurements, which can lead to a more consistent spatial interpolation and a faster convergence during the GPR model optimization.

5
$$\mathbf{r}_i = \begin{cases} \mathbf{y}_i, & \text{if reference node} \\ \alpha_i \cdot \mathbf{y}_i + \beta_i, & \text{if low - cost node} \end{cases}$$
 (3)

where  $y_i$  is either a vector of all the daily PM2.5 measurements of reference node *i* or a vector of all the daily raw PM2.5 signals of low-cost node *i*;  $r_i$  is either a vector of all the daily PM2.5 measurements of reference node *i* or a vector of all the daily calibrated PM2.5 measurements of low-cost node *i*;  $\alpha_i$  and  $\beta_i$  are the slope and intercept, respectively, determined from the fitted simple linear regression calibration equation with daily PM2.5 mass concentrations of the

10 uncalibrated low-cost node *i* as independent variable and PM2.5 mass concentrations of low-cost node *i*'s closest reference node as dependent variable.

**2.3.3 Optimize GPR model (hyperparameters)**

In the next step (step three in Fig. 3), a GPR model was fit to each day *t*'s 31 nodes (i.e., 10 initialized low-cost nodes and 21 reference nodes) as described in Eq. (4). Prior to the GPR model fitting, all the PM2.5 measurements of the 31 nodes

- 15 over 59 valid days used for GPR model hyperparameters training were standardized. The standardization was performed by first concatenating all these training PM2.5 measurements (from the 31 nodes over 59 days), then subtracting their mean  $\mu_{training}$  and dividing them by their standard deviation  $s_{training}$  (i.e., transforming all the training PM2.5 measurements to have a zero mean and unit variance). It is worth noting that assuming the mean function  $\mu \in \mathbb{R}^{31}$  to be 0 along with standardizing all the training PM2.5 samples in this study is one of the common modelling
- 20 formulations on the GPR model and the simplest one. More complex formulations including a station-specific mean function (lack of prior information for this project), a time-dependent mean function (computationally expensive), and a combination of both were not considered for this paper. After the standardization of training samples, the GPR was trained to maximize the log marginal likelihood over all 59 days using Eq. 5 and using an L-BFGS-B optimizer (Byrd et al., 1994). To avoid bad local minima, several random hyperparameter initializations were tried and the initialization that
- 25 resulted in the largest log marginal likelihood after optimization was chosen (in this paper,  $\Theta = [\sigma_s^2, l, \sigma_n^2]$  was initialized to [0.1, 50, 0.01])."

$$\boldsymbol{r}_t | \boldsymbol{\Gamma} \sim N(\boldsymbol{\mu}, \boldsymbol{\Sigma})$$

where *t* ranges from 1 (inclusive) to 59 (inclusive);  $r_t \in \mathbb{R}^{31}$  is a vector of all 31 nodes' PM2.5 measurements (calibrated if low-cost nodes) on day t;  $\Gamma = \{x_1, ..., x_{31}\}$  denotes 31 nodes' locations and  $x_i \in \mathbb{R}^2$  is a vector of the latitude and longitude of node *i*:  $\mu \in \mathbb{R}^{31}$  represents the mean function (assumed to be 0 in this study) and  $\Sigma \in \mathbb{R}^{31 \times 31}$  with  $\Sigma_{\mu} =$

(4)

30 longitude of node *i*;  $\mu \in \mathbb{R}^{31}$  represents the mean function (assumed to be 0 in this study) and  $\Sigma \in \mathbb{R}^{31 \times 31}$  with  $\Sigma_{ij} = K(x_i, x_j; \Theta)$  represents the covariance function/kernel function.

 $\arg\max_{\boldsymbol{\Theta}} L(\boldsymbol{\Theta}) = \arg\max_{\boldsymbol{\Theta}} \sum_{t=1}^{59} \log p(\boldsymbol{r}_t | \boldsymbol{\Theta}) = \arg\max_{\boldsymbol{\Theta}} \left( -0.5 \cdot 59 \cdot \log |\boldsymbol{\Sigma}_{\theta}| - 0.5 \sum_{t=1}^{59} \boldsymbol{r}_t^T \boldsymbol{\Sigma}_{\theta}^{-1} \boldsymbol{r}_t \right)$ (5) where  $\boldsymbol{\Theta} \in \mathbb{R}^3$  is a vector of the GPR hyperparameters  $\sigma_s^2$ , l, and  $\sigma_n^2$ ."

**2.3.4 Update low-cost nodes' (simple linear regression) calibrations based on their conditional means**

- 5 Once the optimum  $\Theta$  for the (initial) GPR was found, we used the learned covariance function to find the mean of each lowcost node *i*'s Gaussian Distribution conditional on the remaining 30 nodes within the network (i.e.,  $\mu_{A|B}^{it}$ ) on day *t* as described mathematically in Eq. (6)–(8) and repeatedly did so until all 59 days'  $\mu_{A|B}^{it}$  (i.e.,  $\mu_{A|B}^{i} \in \mathbb{R}^{59}$ ) were found and then re-calibrated that low-cost node *i* based on the  $\mu_{A|B}^{i}$ . The re-calibration was done by first fitting a simple linear regression model to all 59 pairs of daily PM2.5 mass concentrations from the uncalibrated low-cost node *i* (*yi*,
- 10 independent variable) and its conditional mean  $(\mu_{A|B}^{i})$ , dependent variable) and then using the updated calibration factors (slope  $\alpha_i$  and intercept  $\beta_i$ ) obtained from this newly fitted simple linear regression calibration model to calibrate the low-cost node *i* again (using Eq. 3). This procedure is summarized graphically in Fig. 3 step four and was performed iteratively for all low-cost nodes one at a time. The reasoning behind this step is given in the Supplement. A highlevel interpretation of this step is that the target low-cost node is calibrated by being weighted over the remaining nodes
- 15 within the network and the  $\Sigma_{AB}^{it} \Sigma_{BB}^{it}^{-1}$  term computes the weights. In contrast to the inverse distance weighting interpolation which will weight the nodes used for calibration equally if they are equally distant from the target node, the GPR will value sparse information more and lower the importance of redundant information (suppose all the nodes are equally distant from the target node) as shown in Fig. S2.

$$p\left(\begin{bmatrix} r_A^{it} \\ r_B^{it} \end{bmatrix}\right) = N\left(\begin{bmatrix} r_A^{it} \\ r_B^{it} \end{bmatrix}; \begin{bmatrix} \mu_A^{it} \\ \mu_B^{it} \end{bmatrix} \begin{bmatrix} \Sigma_{AA}^{it} & \Sigma_{AB}^{it} \\ \Sigma_{BA}^{it} & \Sigma_{BB}^{it} \end{bmatrix}\right)$$
(6)

$$\quad r_A^{it} \left| \boldsymbol{r}_B^{it} \sim N\left( \boldsymbol{\mu}_{A|B}^{it}, \boldsymbol{\Sigma}_{A|B}^{it} \right) \right. \tag{7}$$

$$\mu_{A|B}^{it} = \mu_A^{it} + \Sigma_{AB}^{it} \Sigma_{BB}^{it}^{-1} (r_B^{it} - \mu_B^{it})$$
(8)

where  $r_A^{it}$  and  $r_B^{it}$  are the daily PM2.5 measurement(s) of the low-cost node *i* and the remaining 30 nodes on day *t*;  $\mu_A^{it}$ ,  $\mu_B^{it}$ , and  $\mu_{A|B}^{it}$  are the mean (vector) of the partitioned Multivariate Gaussian Distribution of the low-cost node *i*, the remaining 30 nodes, and the low-cost node *i* conditional on the remaining 30 nodes, respectively, on day *t*; and  $\Sigma_{AA}^{it}$ ,  $\Sigma_{BA}^{it}$ ,  $\Sigma_{BB}^{it}$ , and

25  $\sum_{A|B}^{it}$  are the covariance between the low-cost node *i* and itself, the low-cost node *i* and the remaining 30 nodes, the remaining 30 nodes and the low-cost node *i*, the remaining 30 nodes and themselves, and the low-cost node *i* conditional on the remaining 30 nodes and itself, respectively, on day *t*.

**2.3.5 Optimize alternately and iteratively and converge**

Iterative optimizations alternated between the GPR hyperparameters and the low-cost node calibrations using the approaches described in Sect. 2.3.3 and 2.3.4, respectively (Fig. 3 steps five and six, respectively), until the GPR parameters  $\Theta$  converged with the convergence criteria being the differences in all the GPR hyperparameters between the two adjacent runs below 0.01 (i.e., with  $\Delta \sigma_s^2 \leq 0.01$ ,  $\Delta l \leq 0.01$ , and  $\Delta \sigma_n^2 \leq 0.01$ ).

**2.3.6 Predict on the holdout reference node and calculate accuracy metrics**

The final GPR was used to predict the 59-day PM2.5 measurements of the holdout reference node (Fig. 3 step seven) following the Cholesky decomposition algorithm (Rasmussen and Williams, 2006) with the standardized predictions being transformed back to the original PM2.5 measurement scale at the end. **The back transformation was done by multiplying**

10 the predictions by the standard deviation  $s_{training}$  (the standard deviation of the training PM2.5 measurements) and then adding back the mean  $\mu_{training}$  (the mean of the training PM2.5 measurements). Metrics including root mean square errors (RMSE, Eq. 9) and percent errors defined as RMSE normalized by the average of the true measurements of the holdout reference node in this study (Eq. 10) were calculated for each fold and further averaged over all 22 folds to assess the accuracy and sensitivity of our simultaneous GPR and simple linear regression calibration model.

15 RMSE =
$$\sqrt{\frac{1}{59}} \| \boldsymbol{y}_i - \hat{\boldsymbol{y}}_i \|_2^2$$
 (9)

(10)"

where  $y_i$  and  $\hat{y}_i$  are the true and model predicted 59 daily PM2.5 measurements of the holdout reference node *i*.

Percent error =  $\frac{\text{RMSE}}{\text{avg. holdout reference PM}_{2.5} \text{ conc.}}$

20 Modified Figure 3:

5